# CodecSep: Prompt-Driven Universal Sound Separation on Neural Audio Codec Latents

**Adhiraj Banerjee**                                                    *adhirajbanerjee35@gmail.com*
*Department of Electrical Engineering*
*Indian Institute of Technology, Kanpur*

**Vipul Arora**                                                              *vipular@iitk.ac.in*
*Department of Electrical Engineering*
*Indian Institute of Technology, Kanpur*

**Reviewed on OpenReview:** *https://openreview.net/forum?id=r63GX9hKhC*

## Abstract

Text-guided sound separation enables flexible audio editing and assistive applications, but existing open-domain systems such as AudioSep remain too compute-intensive for low-latency edge or codec-mediated deployment. Neural audio codec (NAC)-based separators such as CodecFormer and SDCodec are more efficient, but they are largely restricted to fixed-class or fixed-stem separation.

We introduce **CodecSep**, a *text-guided universal sound separation* framework that operates directly in neural audio codec latent space. CodecSep combines a frozen DAC backbone with a lightweight Transformer *masker* conditioned by CLAP-derived FiLM parameters, enabling open-vocabulary source extraction while preserving the efficiency advantages of codec-native representations. To our knowledge, this is the first prompt-driven universal sound separation system built directly on NAC latents.

Across **dnr-v2** and five additional open-domain benchmarks under matched training and prompting protocols, CodecSep consistently improves over AudioSep in separation fidelity (**SI-SDR**) while remaining competitive in perceptual quality (**ViSQOL**), and also shows gains in human **MOS–LQS**. Further analyses show that finer-grained semantic supervision improves separation more consistently than coarse prompting, and that *explicit masking* is more effective than decoder-style latent generation for codec-domain source separation. Qualitative and diagnostic analyses further support the central design premise: modern NAC latents preserve meaningful *source-dependent structure*, and the learned masks exploit this structure primarily through *channel-wise modulation*, indicating that source extraction can be performed through masking alone without explicit latent generation.

From a systems perspective, CodecSep also provides a concrete *deployment path* for codec-mediated audio processing. In deployment-typical *code-stream* settings, where the edge device transmits audio as NAC codes generated by the same codec backbone used by the separator, the server can map the received codes to codec embeddings through codebook lookup and perform separation directly in codec space, avoiding a separate decode–separate–re-encode cycle. In this regime, CodecSep requires only **1.35 GMACs** end-to-end—about **54×** less compute than AudioSep in the same codec-mediated pipeline (and about **25×** lower separator-only compute)—while also reducing latency and memory footprint substantially and remaining fully compatible with *codes in: codes out* operation. More broadly, this codes-in / codes-out formulation provides a concrete blueprint for *codec-native downstream audio processing*, suggesting that tasks such as enhancement, denoising, dereverberation, and prompt-guided audio editing can be designed to operate directly on NAC representations rather than repeatedly decoding to waveform and re-encoding after each processing stage.

# 1 Introduction

We propose CodecSep, a text-conditioned universal sound separation (USS) framework that marries the interpretability of prompt-driven extraction with the efficiency of neural audio codecs (NACs). To our knowledge, CodecSep is the first system to bridge NACs with USS: it conditions a transformer *masker* on CLAP text embeddings Wu et al. (2023) via Feature-wise Linear Modulation (FiLM) Perez et al. (2018), and performs separation directly in the codec encoder latent space. This design introduces semantic control while retaining the compact computational footprint of codec representations, making prompt-driven separation particularly well-suited to low-latency, codec-mediated deployment, including edge and potentially on-device settings.

Flexible, real-time separation on bandwidth- or compute-constrained platforms remains challenging. Classic models disentangle sources from complex mixtures Vincent et al. (2018) but are often domain-specific (e.g., speech/music) and heavy. Recent text-guided systems like AudioSep Liu et al. (2024) extend encoder–masker–decoder designs (e.g., Conv-TasNet-style Luo & Mesgarani (2019)) by injecting semantics from BERT/CLAP through FiLM layers Devlin et al. (2019); Wu et al. (2023); Perez et al. (2018). However, spectrogram/waveform-domain separators trained with SI-SDR-style losses Luo & Mesgarani (2019); Le Roux et al. (2019) are compute-intensive and sensitive to compression artifacts, often pushing inference to the cloud.

NACs such as SoundStream, Encodec, and DAC Zeghidour et al. (2022); Défossez et al. (2022); Kumar et al. (2023) compress audio to discrete tokens with Residual Vector Quantization (RVQ), providing compact, perceptually aligned latents useful for generation and conditioned synthesis Borsos et al. (2023); Wang et al. (2023; 2024); Du et al. (2024). Prior codec–separation hybrids (CodecFormer Yip et al. (2024b), SDCodec Bie et al. (2024)) are lightweight and high-fidelity but target fixed stems (e.g., speech separation or speech vs. music vs. SFX); extending them to open-domain, prompt-conditioned USS is non-trivial (cf. §2, para. 3).

CodecSep adopts a frozen DAC encoder–decoder backbone and inserts a FiLM-conditioned transformer masker that predicts a soft mask over codec latents. CLAP-derived text embeddings Wu et al. (2023) are mapped to per-layer FiLM parameters, modulating the masker's intermediate activations to align the selected latent subspace with the query semantics. Operating on compact codec features cuts memory traffic and MACs compared to spectrogram-domain pipelines, while preserving the codec's inductive biases (periodicity, timbre, transients). In doing so, CodecSep delivers interpretable, prompt-guided separation with markedly lower compute without sacrificing separation fidelity. Crucially, conditioning via text embeddings enables open-vocabulary operation of NAC-based separation.

We evaluate CodecSep along seven complementary axes: (i) *in-domain text-guided separation* on dnr-v2 Petermann et al. (2022); (ii) *cross-domain generalization* on five open-domain benchmarks—AudioCaps Kim et al. (2019), ESC-50 Piczak, Clotho-v2 Drossos et al. (2020), AudioSet-eval Gemmeke et al. (2017), and VGGSound Chen et al. (2020a); (iii) *prompt granularity*, comparing fixed-stem baselines, generic three-stem prompting, and fine-grained compositional SFX prompting; (iv) *robustness to prompt paraphrasing*; (v) *architectural analysis*, contrasting decoder-style latent generation with explicit masking in codec latent space; (vi) *qualitative and diagnostic analysis* of the learned latent structure, including source-dependent latent organization, mask behavior, and oracle/reconstruction studies; and (vii) *deployment-oriented efficiency benchmarking* in terms of compute, latency, and memory footprint.

We compare primarily against the state-of-the-art text-guided baseline, AudioSep Liu et al. (2024), under matched training data and prompt protocols. Across benchmarks, CodecSep consistently improves over AudioSep in **SI-SDR** while remaining competitive in **ViSQOL**, and it degrades more gracefully under prompt paraphrasing. The results further show that finer-grained semantic supervision improves separation more consistently than coarse prompting, and that explicit masking is more effective than decoder-style latent generation for codec-domain source separation. Importantly, the qualitative latent-space analysis provides direct support for the central design premise: the codec latent space preserves meaningful *source-dependent structure*, and the learned masks exploit this structure primarily through *channel-wise modulation*, providing qualitative evidence that modern NAC representations are sufficiently organized to support source extraction through masking alone, without explicit latent generation or re-encoding. In deployment-typical *code-stream*

settings, where audio is already exchanged as codec bitstreams, CodecSep requires only **1.35 GMACs** end-to-end—about 54× **less compute** than AudioSep in the same regime (and about 25× **lower** separator-only compute)—while remaining fully compatible with bitstream interfaces.

Our main contributions are **fourfold**.

**First**, we introduce **CodecSep**, a *text-guided universal sound separation* framework that operates *directly* in neural audio codec latent space using a FiLM-conditioned Transformer *masker*. To our knowledge, this is the first prompt-driven USS system built on neural audio codec representations, combining open-vocabulary semantic control with a codec-native separation pipeline.

**Second**, we show through extensive evaluation that codec-latent masking is an effective formulation for universal sound separation. We study CodecSep across: (i) *in-domain text-guided separation* on dnr-v2 Petermann et al. (2022); (ii) *cross-domain generalization* on five open-domain benchmarks—AudioCaps Kim et al. (2019), ESC-50 Piczak, Clotho-v2 Drossos et al. (2020), AudioSet-eval Gemmeke et al. (2017), and VGGSound Chen et al. (2020a); (iii) *prompt granularity*, comparing fixed-stem baselines, generic three-stem prompting, and fine-grained compositional SFX prompting; and (iv) *robustness to prompt paraphrasing*. Under matched training data and prompt protocols, CodecSep consistently improves over the state-of-the-art text-guided baseline **AudioSep** Liu et al. (2024) in **SI-SDR** while remaining competitive in **ViSQOL**, and it degrades more gracefully under paraphrased prompts.

**Third**, we provide both *architectural* and *qualitative* evidence for the underlying codec-latent separation mechanism. Architecturally, we show that **explicit masking** is more effective than decoder-style latent generation for source separation in codec space. Qualitatively, our latent-space analysis shows that the frozen codec representation preserves meaningful *source-dependent structure*, while the learned masks are predominantly *channel-wise*, indicating that CodecSep separates sources mainly through source-conditioned latent reweighting rather than strongly time-localized gating. Together with the oracle and reconstruction diagnostics, these results support the view that modern neural audio codec latents are sufficiently organized to support source extraction through masking alone, without explicit latent generation or re-encoding.

**Fourth**, we show that this modeling choice yields a substantial *systems advantage* in codec-mediated deployment. In an edge–server workflow, the edge device may already transmit audio as *neural audio codec (NAC) codes* rather than as raw waveform samples. Under the assumption that the edge device uses the *same codec backbone and embedding interface* on which the masker is trained, CodecSep can operate directly in that codec domain. Concretely, the server converts the received codec codes to codec embeddings through *codebook lookup*, applies the separator in latent space to estimate source-specific representations, and can then either decode them to audio or re-quantize them back into codec codes for downstream transmission. This avoids the separate decode–separate–re-encode cycle required by conventional waveform- or spectrogram-domain separators and provides a practical *codes in: codes out* deployment pathway. In this regime, CodecSep requires only **1.35 GMACs** end-to-end—about 54× **less compute** than AudioSep in the same setting (and about 25× **lower** separator-only compute)—while also reducing latency and memory footprint substantially. More broadly, this deployment pathway is useful for efficient codec-mediated downstream applications and provides a concrete blueprint for how audio processing modules can operate directly on NAC representations, rather than repeatedly decoding to waveform and re-encoding after each downstream task.

## 2 Related Work

Classical sound separation systems frequently adopt an encoder–masker–decoder design in which an encoder produces STFT-like latents, a masker predicts source-specific masks, and a decoder reconstructs waveforms. Representative models include DPTNet Chen et al. (2020b), SepFormer Subakan et al. (2021), and TDANet Li et al. (2023), the last introducing a top-down attention scheme that blends global and local attention to capture multi-scale acoustic structure. Beyond masking pipelines, several works generate waveforms directly in the time domain (Wave-UNet Stoller et al. (2018), Demucs Défossez et al. (2019); Défossez et al. (2021)) or operate fully in the complex STFT domain with joint magnitude–phase modeling (MM-DenseLSTM Takahashi et al. (2018), DCCRN Hu et al. (2020), Spleeter Hennequin et al. (2020)), underscoring the breadth of design choices.

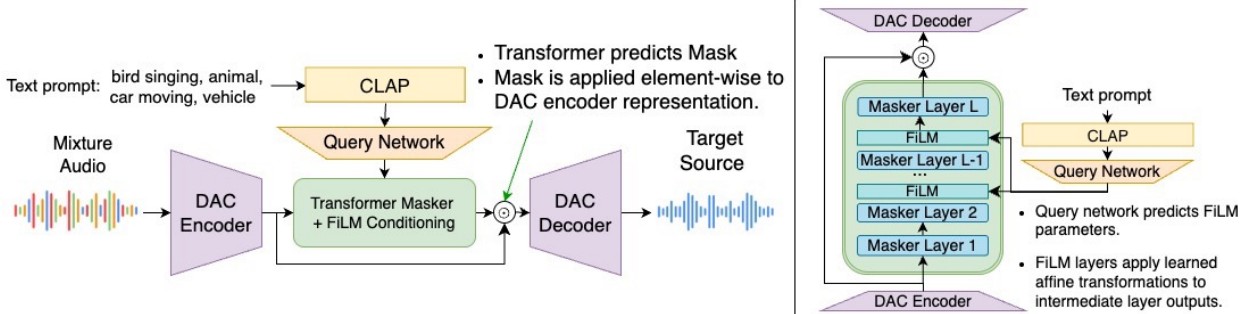

Figure 1: An overview of CodecSep. (Left) The full pipeline for text-guided USS. (Right) The integration of text conditioning into intermediate layers of transformer masker via FiLM layers.

Moving from domain-specific separation to *universal* sound separation (USS), supervised systems typically rely on Permutation Invariant Training (PIT) Yu et al. (2017); Kavalerov et al. (2019), while unsupervised methods such as MixIT Wisdom et al. (2020) learn directly from mixtures. Both paradigms assume a fixed maximum number of sources and output all estimates indiscriminately, requiring a post-hoc identification step (cf. Appendix A for detailed failure modes). A recent PIT-trained USS model is Sudo rm-rf Tzinis et al. (2022a), a parameter-efficient time-domain separator that is based on ConvTasNet-style encoder–masker–decoder architecture with adaptive encoder/decoder modules. It downsamples input waveform to STFT-like latents before separation and is PIT-trained to separate mixtures with up to four sources. Query-Guided Sound Separation (QSS) addresses this limitation of PIT or MixIT models by conditioning extraction on external queries—visual cues, audio, class labels, or text. Text queries are compact, expressive, and capture high-level semantics without requiring additional reference signals. AudioSep Liu et al. (2024) follows this direction by injecting BERT Devlin et al. (2019) or CLAP Wu et al. (2023) embeddings via FiLM Perez et al. (2018) at intermediate masker layers to steer separation toward the query source. BiModalSS Mahmud et al. (2024) extends AudioSep with attention-based conditioning and more efficient training strategies. FiLM-conditioned variants of Sudo rm-rf Tzinis et al. (2022b; 2023) have also been explored for class-guided separation using one-hot or multi-hot conditioning vectors; however, such label-based conditioning does not extend naturally to open-domain text queries and cannot handle unseen classes.

Neural audio codecs (NACs) have recently been integrated into separation pipelines to improve efficiency. CodecFormer separates directly in DAC Kumar et al. (2023) latent space with a transformer trained using negative SI-SDR, and CodecFormer-EL Yip et al. (2024a) adds an embedding-level objective to align separator outputs with encoder latents. SDCodec Bie et al. (2024) embeds separation inside the codec by assigning dedicated RVQ branches to speech, music, and SFX and summing their codes to form mixture representations. However, neither design trivially extends to open-domain, prompt-conditioned USS: SDCodec's hardwired, per-stem RVQ branches do not scale to open vocabularies (adding branches explodes parameters, while a mixture-invariant reformulation collapses into "three generic codecs" with no stem-specific RVQ specialization). MixIT-style training of CodecFormer still presupposes a maximum number of sources, conflicting with universal separation.

## 3 Method

CodecSep adapts the 16 kHz DAC Kumar et al. (2023) codec backbone for text-driven universal sound separation (USS). We use a transformer masker that estimates soft masks over codec latents and inject text conditioning via Feature-wise Linear Modulation (FiLM) Perez et al. (2018) using CLAP text embeddings Wu et al. (2023). FiLM is applied to intermediate transformer activations so the query semantics steer separation. Figure 1 (Left) shows the overall text-guided pipeline; Figure 1 (Right) highlights the FiLM-conditioned masker. Compared to STFT-domain AudioSep, operating in compact codec latents yields markedly lower compute and is amenable to edge deployment.

## 3.1 Task Formulation

We consider a mono mixture as $x(t) = \sum_{s \in \mathcal{S}} y_s(t)$, where $x(t)$ is the observed waveform, $\mathcal{S}$ is the (unbounded) set of source classes/instances present, and $y_s(t)$ is the waveform of source $s$. Given a natural-language query $\tau$ (e.g., "dog barking", "speech and music"), the goal is to recover the waveform of the source consistent with the query.

In spectrogram-domain text-guided separation systems such as AudioSep, this takes the form

$$x(t) \xrightarrow[\text{STFT}]{} X \in \mathbb{C}^{F \times T_{spec}} \xrightarrow{Spec(X, e_\tau)} \tilde{Y}_s = |\hat{M}_s| \odot |X| \exp\left(\angle X + \angle \hat{M}_s\right) \xrightarrow{\text{ISTFT}} \tilde{y}_s(t), \tag{1}$$

where $Spec(\cdot, e_\tau)$ is a FiLM-conditioned masker that predicts a magnitude mask $|\hat{M}_s| \in [0,1]^{F \times T_{spec}}$ ($F$: frequency bins, $T_{spec}$: spectrogram frames) and a phase residual $\angle \hat{M}_s$ given the complex STFT $X$ of audio $x(t)$ and text-embedding $e_\tau$.

In CodecSep, the task is posed directly in the NAC latent domain:

$$x(t) \xrightarrow[\text{DAC}]{Enc(\cdot)} Z \in \mathbb{R}^{d \times T} \xrightarrow{Mask(Z, e_\tau)} \tilde{Z}_s = M_s \odot Z \xrightarrow[\text{DAC}]{Dec(\cdot)} \tilde{y}_s(t), \tag{2}$$

with frozen DAC backbone $Enc(\cdot), Dec(\cdot)$ and a FiLM-conditioned transformer masker $Mask(\cdot, e_\tau)$ that estimates mask $M_s \in [0,1]^{d \times T}$ applied element-wise to DAC latent $Z$.

Thus, for an input clip producing $T$ latent frames, the separator receives $Z \in \mathbb{R}^{d \times T}$, predicts a same-shape mask $M_s \in [0,1]^{d \times T}$, forms the masked latent $\tilde{Z}_s \in \mathbb{R}^{d \times T}$, and decodes it to the waveform estimate $\tilde{y}_s(t)$. In the main system, the separator therefore acts as a selection mechanism over codec latents rather than a source generator.

## 3.2 Model Architecture

### 3.2.1 Descript Audio Codec (DAC) Backbone

We use DAC Kumar et al. (2023) as encoder–decoder. Following Encodec/SoundStream, DAC uses fully-convolutional encoder/decoder with the periodic Snake activation $x + \sin^2 x$ (replacing LeakyReLU) to bias periodic audio modeling. Residual vector quantization (RVQ) compresses encoder outputs with factorized codes and $\ell_2$-normalized codebooks. For a 1 s audio $x(t)$ at $F_s{=}24\,\text{kHz}$ compressed to $R{=}6000\,\text{bps}$, the encoder $Enc(\cdot)$ downsamples by $M{=}320$ to $T{=}F_s/M{=}75$ frames of latents $Z{=}[z_t \in \mathbb{R}^d]_{t=1}^T$ ($d$: channel width) with $r{=}R/T{=}80$ bits/frame. RVQ allocates $r_i{=}r/N_q{=}10$ bits across $N_q{=}8$ codebooks (size $2^{10}{=}1024$). Given $Z$, $Quant(\cdot)$ yields discrete codes $A{=}[a_t \in [1024]^8]$, which map to embeddings $e_t{=}\sum_{i=1}^8 e_t^i$; $Dec(\cdot)$ upsamples $E{=}[e_t]$ back to waveform $y(t)$.

### 3.2.2 Why NAC latents vs. spectrograms

Operating on NAC latents $Z$ slashes dimensionality while preserving perceptual factors. For 1 s audio at 32 kHz, complex STFT with $N{=}1024$ and hop size $M{=}320$ samples has $T_{\text{spec}} \approx 100$ frames and $F = 2 \times 1024$ (Re+Im) scalars per frame, so $F \cdot T_{spec} \approx 204{,}800$. A 16 kHz DAC with width $d{=}64$ and the same $M$ yields $T \approx 50$ and $d \cdot T{=}64 \times 50{=}3{,}200$ ($\sim 64\times$ smaller), shrinking $Q/K/V$ and MLP sizes and easing self-attention. Similarly, for 32 kHz NACs like EnCodec, $T \approx 50$ with $d{=}128$, so attention/MLPs still operate on $\sim 32\times$ smaller latents than complex STFTs. Crucially, $Enc(\cdot)$ organizes $Z$ on a discriminative, perceptually aligned manifold, making selection (masking) easier than representation learning from raw $X$. In spectrogram systems $Spec(.)$, the separator must first learn a high-level latent from $X$ (via CNN/UNet) and then separate, coupling abstraction and masking and inflating parameters/compute. Waveform separators $Wave(.)$ such as Sudo rm-rf Tzinis et al. (2022a;b; 2023) likewise downsample the waveform into STFT-like intermediate latents via 1D convolutions before encoding, resulting in latents with similar dimensionality and thus inheriting the same challenges as spectrogram systems.

### 3.2.3 FiLM-conditioned Transformer Masker

**Leveraging the codec prior.** Because the DAC codec induces a strong semantic prior in its latent space via residual vector quantization (RVQ) and perceptual/adversarial training, we mask the codec latents rather than generate sources from scratch as in CodecFormer. RVQ creates a coarse-to-fine hierarchy in $Z=Enc(x) \in \mathbb{R}^{d \times T}$: early stages capture coarse structure (e.g., low-frequency content, timbre), while later stages refine residual detail (e.g., high-frequency components, transients) Wang et al. (2023). We exploit this organization with a FiLM-conditioned transformer masker that predicts a soft mask $M_s \in [0,1]^{d \times T}$ and applies it element-wise, yielding source latent estimate $\tilde{Z}_s = M_s \odot Z$.

**Masking, not generating.** In contrast to learning a generator $Gen(Z, e_\tau) : Z \to \tilde{Z}_s$ as in CodecFormer, learning a *mask* $Mask(Z, e_\tau) : Z \to M_s$ on the compact, semantically organized codec manifold both exploits the codec prior more effectively and yields a more stable optimization that converges faster. Moreover, masking in the denoised, low-dimensional codec space is fundamentally easier than masking in the high-dimensional, noisy spectrogram domain. This selection-centric design (i) constrains learning to modulation of existing latent content, (ii) avoids hallucination and reduces leakage because no new signals are synthesized, and (iii) preserves long-horizon structure (periodicity, timbre, transients) already organized by the codec, yielding stable, low-artifact separations.

**Architecture and dimensional interface.** Concretely, passing $x(t)$ through the frozen DAC encoder yields codec latents $Z \in \mathbb{R}^{d \times T}$ ($d$: codec channel width, $T$: latent frames). The masker $Mask(\cdot)$ operates in this latent space to predict an element-wise mask that selects the target source.

Since the codec latent dimensionality $d$ can differ from the transformer width, we introduce lightweight channel projections to interface between the two. Specifically, the codec latents are first mapped to the transformer width $d_t$ using a pointwise convolution,

$$Z' = \text{Conv}(Z), \quad Z' \in \mathbb{R}^{d_t \times T}, \tag{3}$$

after which all transformer operations are performed in this space.

We adopt a CodecFormer-style transformer with $L$=16 layers, width $d_t$=256, and Snake activations. Given the natural-language query $\tau$, we compute a CLAP text embedding $e_\tau \in \mathbb{R}^{d_t}$. A lightweight query network $query(.)$—implemented as a single linear layer—maps $e_\tau$ to per-layer FiLM parameters $(\gamma^l, \beta^l) \in \mathbb{R}^{d_t}$ for $l \in \{2, \ldots, L-1\}$, applied channel-wise to intermediate activations $H^l \in \mathbb{R}^{d_t \times T}$:

$$\tilde{H}^l = \text{FiLM}(H^l; \gamma^l, \beta^l) = \gamma^l \odot H^l + \beta^l. \tag{4}$$

Thus, FiLM is injected at the intermediate transformer layers $l = 2, \ldots, L-1$, while the first and final transformer layers remain unmodulated.

The final transformer output $H^L$ is then passed through a convolutional mask head consisting of a 1D convolution followed by a pointwise ($1 \times 1$) convolution. Together, these layers map the transformer output from width $d_t$ back to the codec latent dimensionality $d$ and produce the prompt-conditioned mask $M_s \in [0,1]^{d \times T}$. The mask is then applied element-wise to the codec latents, yielding $\tilde{Z}_s = M_s \odot Z$. Finally, the waveform estimate is obtained with the frozen codec decoder as $\tilde{y}_s = Dec(\tilde{Z}_s)$, bypassing RVQ lookup.

Temporal resolution is preserved throughout, and all projections are channel-wise operations. This design explicitly handles the dimensionality mismatch between codec latents and transformer representations while maintaining a direct, shape-aligned masking operation in the original codec latent space.

**Why FiLM inside the masker.** Placing FiLM in the masker (rather than in $Enc/Dec$) directly targets the selection step while preserving the codec manifold. We further adopt a post-LN FiLM design: modulation is applied after the transformer sublayer activations, rather than through the normalization itself. In our setting, this is desirable because the conditioning behaves as an explicit channel-wise feature gate, allowing the masker to suppress interference and emphasize target-relevant latent components without perturbing the backbone normalization statistics. This design introduces minimal overhead (two vectors per layer) and maintains a non-iterative, single-pass conditioning mechanism, enabling low-latency inference in both edge and server settings.

**Why Post-LN FiLM instead of AdaLN.** We apply FiLM after the transformer sublayer rather than conditioning the normalization itself (AdaLN-style), following prior text-guided separation work such as AudioSep. Our motivation is task-specific. CodecSep performs *masking-based source selection*, not generative synthesis, so the conditioning mechanism must act as directly as possible on the features that determine whether a latent component is preserved or suppressed. In this setting, Post-LN FiLM is well aligned with the separation objective because it modulates the hidden activations *after* they have been computed, through channel-wise affine conditioning. Operationally, this makes the text signal behave like an explicit feature gate: for a given query, it can directly attenuate channels carrying interfering content and emphasize channels carrying target-relevant structure, while leaving the backbone normalization statistics unchanged.

By contrast, AdaLN injects conditioning into the normalization transform itself. This is highly effective in generative architectures such as DiT Peebles & Xie (2023), where conditioning is intended to steer representation formation in a broad and distributed manner. However, for source separation—especially in our lightweight low-GMAC regime—the goal is not broad conditional generation but sharp source-selective suppression. In that regime, distributing conditioning through the normalization pathway can lead to more diffuse modulation of the representation, whereas Post-LN FiLM provides a simpler and more targeted mechanism for source-conditioned masking. We therefore adopt Post-LN FiLM as the conditioning scheme that is more naturally matched to explicit latent selection in codec-space separation.

This interpretation is also consistent with our qualitative analysis in § 4.1.4, where the learned masks appear predominantly channel-wise rather than strongly time-localized, which is consistent with the view that separation is achieved mainly through FiLM-conditioned feature reweighting within the masker.

**FiLM parameterization and training stability.** Following AudioSep, we adopt a simplified FiLM parameterization where only the bias term is learned, while the scale is fixed. Concretely, we set $\gamma^l = \mathbf{1}$ for all layers and learn only $\beta^l$, so that FiLM reduces to an additive modulation:

$$\tilde{H}^l = H^l + \beta^l. \tag{5}$$

The FiLM layers are implemented as linear projections from the text embedding, with weights initialized using Xavier uniform initialization Glorot & Bengio (2010) and biases initialized to zero. This ensures that $\beta^l = \mathbf{0}$ at initialization, so the conditioned transformer initially behaves identically to an unmodulated masker.

This design improves training stability by avoiding uncontrolled multiplicative scaling of hidden activations, which can be particularly problematic in Post-LN architectures. Moreover, in the context of masking-based separation, additive modulation is sufficient to introduce source-selective cues while preserving the structure of the codec latent manifold. Fixing $\gamma^l$ therefore provides a stable and effective conditioning mechanism without the risk of prematurely amplifying or suppressing latent channels.

### 3.3 Training Objective

We supervise on mixtures with prompts spanning *speech*, *music*, and diverse (possibly compositional) *SFX*. Besides per-source reconstruction, we encourage mixture consistency by decoding the summed latent estimates $\tilde{x} = g(\sum_s \tilde{Z}_s)$. The loss maximizes SI-SDR Luo & Mesgarani (2019); Le Roux et al. (2019) for both sources and mixture:

$$\mathcal{L} = -\sum_s \text{SI-SDR}(y_s, \tilde{y}_s) - \text{SI-SDR}(x, \tilde{x}). \tag{6}$$

During training, DAC and the CLAP text encoder are frozen; we update only the FiLM-conditioned masker $Mask(\cdot)$ and the query network $query(\cdot)$.

For training and analysis, we operate on *continuous* latents $Z = Enc(x) \in \mathbb{R}^{d \times T}$: (i) gradients flow cleanly through $Mask(\cdot, e_\tau)$ and $Dec(\cdot)$ with a frozen codec (no straight-through estimators), yielding stable convergence; (ii) RVQ pretraining *regularizes* $Z$ so pitch, timbre, onsets/transients, and textures are hierarchically organized, providing a richer, more disentangled signal for FiLM; and (iii) $Z$ avoids run-to-run variance from codebook utilization (e.g., late RVQ sensitivity, bitrate truncation), reducing the need for special regularizers.

### 3.4 Deployment Discussion

For deployments with compressed bitstreams, we reconstruct embeddings by codebook lookup and use the same masker:

$$A = [a_t \in [1024]^{N_q} \mid t \in [T]], \qquad e_t = \sum_{i=1}^{N_q} \text{lookup}(a_t^{(i)}), \tag{7}$$

$$E = [e_t]_{t=1}^{T} \approx Z, \qquad \tilde{E}_s = M_s \odot E, \qquad \tilde{y}_s(t) = Dec(\tilde{E}_s). \tag{8}$$

When a codes-out interface is desired, we re-quantize masked embeddings and optionally decode:

$$\hat{A}_s = Quant(\tilde{E}_s), \qquad \hat{E}_s = \text{lookup}(\hat{A}_s), \qquad \tilde{y}_s(t) = Dec(\hat{E}_s). \tag{9}$$

By design, $E \approx Z$ at the operating bitrate and $Dec(E)$ already yields high-fidelity reconstructions; because our separator is a masker (selective modulation) rather than a generator, swapping $Z \to E$ preserves the semantics needed for separation with no architectural change. While we report results on $Z$ to isolate separator performance and maintain stable optimization, we also evaluate the bitstream path by feeding reconstructed embeddings $E$ (codes-in) to the same trained masker without any fine-tuning; performance remains competitive relative to the $Z$ path. The residual gap can be narrowed with light fine-tuning the masker on $E$ or optimizing an embedding-consistency loss (cf. CodecFormer-EL) in place of, or alongside, SI-SDR:

$$\mathcal{L}_{\text{emb}} = \sum_s \|\tilde{E}_s - Z_s\|_1. \tag{10}$$

In deployment, the variant simply replaces the masker input with $E$ and optionally re-quantizes for codes-out as,

$$x(t) \xrightarrow[\text{On Edge}]{Quant(Enc(\cdot))} A \xrightarrow[\text{Codes In}]{lookup(A)} E \approx Z \xrightarrow[\text{On Server}]{Mask(E, e_\tau)} \tilde{E}_s = M_s \odot E \xrightarrow[\text{Codes Out}]{Quant(\tilde{E}_s)} \hat{A}_s. \tag{11}$$

In realistic pipelines, edge devices already run a codec and transmit code streams rather than raw audio. Traditional spectrogram-based $Spec(.)$ and waveform-based $Wave(.)$ separators, however, operate on the audio stream: they first convert audio to STFT or STFT-like representations (often via 1D convolutions), and then must *decode $\to$ separate on $X$ $\to$ re-encode*, incurring additional latency and energy cost. In contrast, CodecSep performs *masking directly in the codec domain* and can output code streams without any decode–re-encode cycle.

Concretely, with codec costs $C_{Enc}, C_{Dec}$, spectrogram or audio-stream separator (AudioSep) cost $C_{Spec}$, and CodecSep masker cost $C_{Mask}$:

$$\text{Compute Cost for Code-stream input:} \quad \text{AudioSep} = C_{Dec} + C_{Spec} + C_{Enc}, \quad \text{CodecSep} = C_{Mask}. \tag{12}$$

We treat the codebook lookup $C_{lookup}$ and quantization $C_{Quant}$ costs as negligible ($\approx 0$) and omit the CLAP text-encoder cost since it is shared across all models. Figure 2 illustrates a typical edge–server deployment and compares compute requirements for conventional audio-stream separators (audio in $\to$ codes out) versus CodecSep's code-stream separator (codes in $\to$ codes out). As shown, the CodecSep masker operates on $Z/E$ with small $(d, T)$ where $|Z| \ll |X|$, dramatically reducing attention and MLP activations and enabling tighter batching and lower memory bandwidth. Interface compatibility is immediate when only codes $A$ are available: perform a lookup to obtain $E$, apply the FiLM-conditioned masker, and optionally re-quantize to produce $\hat{A}_s$. CodecSep thus eliminates redundant decode/re-encode cycles in server workflows yielding low-latency, high-fidelity separation at scale. See Appendix B for a full discussion covering all of the aforementioned rationale in §3.

More broadly, this codes-in / codes-out pathway is useful beyond the specific separation setting studied here. It illustrates a general *codec-native* deployment pattern for audio processing, in which downstream models

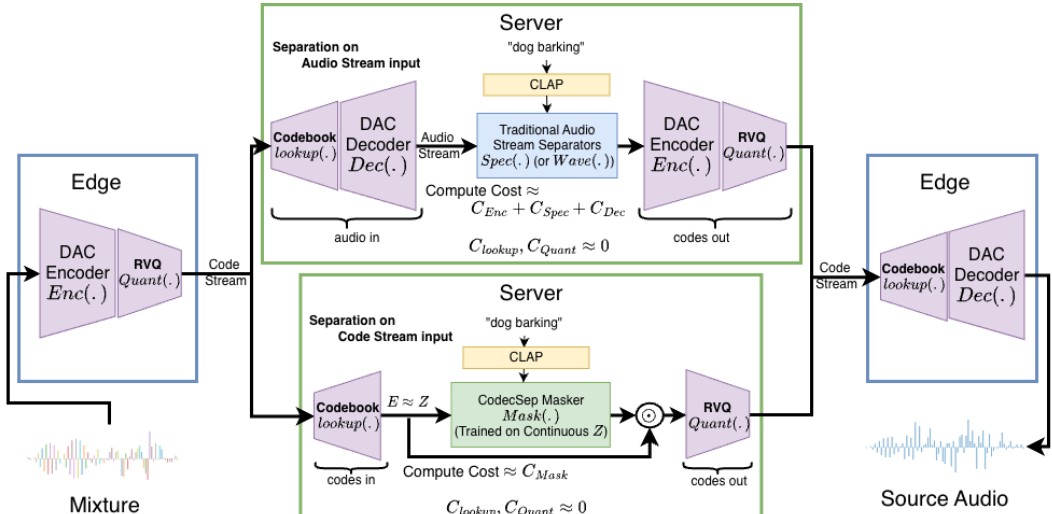

Figure 2: Typical edge–server deployment comparing compute requirements of conventional audio-stream separators (audio in → codes out) versus CodecSep discrete inference (codes in → codes out).

operate directly on neural codec representations rather than repeatedly decoding to waveform, processing in audio or spectrogram space, and re-encoding. This perspective is attractive in realistic edge–server systems because it preserves representational continuity across compression, transmission, and inference while avoiding redundant latency, memory traffic, and energy cost. The same principle could be relevant to other deployment-oriented audio tasks such as target speaker extraction, speech enhancement, denoising, dereverberation, or prompt-guided audio editing, where one may wish to conditionally modulate or refine an existing compressed representation without leaving codec space. In this sense, CodecSep provides not only an efficient separator, but also a concrete example of how neural audio codecs can serve as a shared representational backbone for broader codec-aware audio processing pipelines.

## 4    Experiments

**Datasets.**   We evaluate across a controlled multi-stem corpus and multiple open-domain benchmarks. For in-domain experiments, we adapt Divide and Remaster v2 (dnr-v2) Petermann et al. (2022) from fixed-label three-stem separation to universal, prompt-driven separation by replacing source labels with natural-language queries. Speech and music are queried with broad category prompts (e.g., "speech", "music"), while SFX stems are queried using long-form, compositional prompts ($\geq 2$ overlapping sources) synthesized from FSD50K's hierarchical annotations Fonseca et al. (2022), combining fine-grained classes with parent categories (e.g., "dog barking, *Animal*, engine rumbling, *motor vehicle*"). To assess cross-domain generalization, we form three-source mixtures on AudioCaps Kim et al. (2019) (used for both training and testing in our open-domain setting) and construct test-only three-source mixtures from ESC-50 Piczak, Clotho-v2 Drossos et al. (2020), AudioSet-eval Gemmeke et al. (2017), and VGGSound Chen et al. (2020a). Dataset construction details, clip durations, split statistics, and segmentation rules are provided in the Appendix C. For dnr-v2, we report per-source scores for speech, music, and SFX. For the additional open-domain benchmarks, however, evaluation is event-conditioned rather than stem-conditioned, and many labels do not map cleanly onto the speech/music/SFX taxonomy. We therefore report dataset-level averages there, which better reflect the open-domain extraction setting; a finer semantic grouping could be explored in future analysis.

**Evaluation.**   We compare CodecSep against representative spectrogram-, waveform- and codec-domain baselines—TDANet Li et al. (2023); Pons et al. (2024), Sudo rm-rf Tzinis et al. (2022a), CodecFormer Yip et al. (2024b), SDCodec Bie et al. (2024), and the text-guided audio stream separators AudioSep Liu et al. (2024), BiModalSS Mahmud et al. (2024) and Sudo rm-rf + FiLM Tzinis et al. (2022b; 2023). We report objective signal fidelity via scale-invariant signal-to-distortion ratio (SI-SDR) Luo & Mesgarani (2019); Le Roux

et al. (2019) and perceptual quality via ViSQOL Chinen et al. (2020), which measures spectro–temporal similarity between the estimate $\tilde{x}$ and reference $x$ and maps it to a 1–5 MOS-LQO scale. Following prior work (e.g., SDCodec Bie et al. (2024)), we use ViSQOL as a proxy MOS score for perceptual listening quality and complement it with a human MOS-LQS study comparing real-world outputs from CodecSep (trained on dnr-v2) and the publicly released AudioSep. To quantify efficiency, we report multiply–accumulate operations (MACs), inference time, memory footprint using `torchinfo`[1] under matched input durations (2 s) and batching (batch size 2) on a 24GB NVIDIA A30 GPU. Details on evaluation workflow for each benchmark are deferred to Appendix D.

**Training.** Unless otherwise stated, the DAC codec Kumar et al. (2023) and CLAP text encoder Wu et al. (2023) remain frozen; we train the FiLM-conditioned transformer masker and the lightweight query network end-to-end with an Adam optimizer Kingma & Ba (2017) and a plateau-based learning-rate schedule Mukherjee et al. (2019). We produce two variants of CodecSep, trained separately on dnr-v2 and on AudioCaps, to study distributional effects; we denote them with the suffixes "+dnr-v2" and "+AudioCaps". For fair comparison, the 3-stem versions of TDANet, Sudo rm-rf, and CodecFormer are re-trained from scratch on dnr-v2 using our setup. AudioSep is evaluated both as the publicly released checkpoint and when re-trained under matched protocols. We similarly re-train Sudo rm-rf+FiLM under the same matched settings. Pretrained checkpoints of BiModalSS and SDCodec are used as released by the authors. To reflect realistic deployments where signals traverse compression pipelines, inputs to non-codec baselines are passed through a full-band stereo-capable 48 kHz EnCodec during both training and inference. Full hyper-parameters, iteration schedules, batch configurations, and hardware details are deferred to the Appendix E.

## 4.1 Results and Discussions

Tables 1–8 present universal sound separation results, prompt–granularity analyses, architectural ablations, cross-dataset generalization, paraphrase robustness, and full inference complexity under matched training/evaluation protocols. Table 1 reports dnr-v2 test results for speech, music, and SFX: text-guided models use generic prompts for speech/music and ground-truth compositional captions for SFX; we also include a masker ablation to isolate the role of the Transformer masker. Table 5 studies SFX prompt granularity across three regimes—(i) fixed-stem, non-text baselines (TDANet, CodecFormer, SDCodec), (ii) generic 3-stem prompts ({music, speech, sfx}), and (iii) a universal setup with fine-grained, compositional SFX prompts—thereby aligning input conditions for fair comparison with fixed-head systems. Table 6 isolates architecture: decoder-style generation (CodecFormer) vs. an unguided 3-stem masker variant and its text-guided counterpart, all operating in the codec latent space. To assess out-of-domain generalization, Table 2 benchmarks on five additional open-domain corpora (ESC-50, Clotho-v2, AudioSet, VGGSound, AudioCaps) with mixtures of three randomly sampled sources and prompts drawn from captions (not tied to fixed labels).. Finally, Table 9 compares end-to-end and architecture-only GMACs for spectrogram-domain separation versus codec-latent masking, including the practical code-stream case. All models are evaluated against the original (uncompressed) ground truth; our methods are highlighted in **bold**; and we report mean and standard deviation ($1\sigma$).

### 4.1.1 Source Separation Performance: In-Domain, Cross-Domain, and Subjective Evaluation

**Does codec-latent masking outperform spectrogram-domain text-guided separation under matched training on dnr-v2 (cf. Table 1)?** The answer is yes. CodecSep+dnr-v2 outperforms both pretrained AudioSep (zero-shot) and retrained AudioSep+dnr-v2 across all three source categories, with sizable SI-SDR gains in speech (10.0 vs. 4.9/7.7 dB), music (1.2 vs. $-2.5/-5.6$ dB), and SFX (0.9 vs. $-0.3/-4.7$ dB). In ViSQOL, CodecSep matches or exceeds AudioSep in speech and music while slightly trailing on SFX, likely reflecting differences in SFX prompt distributions (AudioSep's diverse training versus CodecSep's compositional SFX prompts derived from dnr-v2). These results indicate that masking in the structured codec latent space is more effective than spectrogram-domain text-guided separation under matched training conditions.

---

[1] https://github.com/tyleryep/torchinfo

Table 1: Results: Separation performance on universal sound separation (**dnr-v2-test**). To reflect the compressed-input deployment scenario studied in this work, zero-shot baselines are evaluated on *codec-processed* audio, while retrained baselines are both trained and inferred on *codec-processed* audio.

| Model | Metric (↑) | Music | Speech | Sfx |
|---|---|---|---|---|
| AudioSep | SI-SDR | $-2.5^{\pm4.06}$ | $4.9^{\pm4.21}$ | $-0.3^{\pm5.39}$ |
| (zero-shot) | ViSQOL | $2.9^{\pm0.63}$ | $3.1^{\pm0.56}$ | $\mathbf{2.6^{\pm0.77}}$ |
| BiModalSS | SI-SDR | $-6.8^{\pm2.73}$ | $1.8^{\pm2.78}$ | $-6.36^{\pm3.57}$ |
| (zero-shot) | ViSQOL | $2.5^{\pm0.57}$ | $2.6^{\pm0.51}$ | $2.3^{\pm0.72}$ |
| AudioSep + dnr-v2 | SI-SDR | $-5.6^{\pm2.89}$ | $7.7^{\pm3.0}$ | $-4.7^{\pm3.68}$ |
| | ViSQOL | $2.6^{\pm0.57}$ | $2.5^{\pm0.37}$ | $2.3^{\pm0.7}$ |
| Sudo rm-rf + FiLM + dnr-v2 | SI-SDR | $-6.7^{\pm2.62}$ | $2.0^{\pm2.76}$ | $-6.6^{\pm3.71}$ |
| | ViSQOL | $2.7^{\pm0.59}$ | $2.9^{\pm0.45}$ | $2.3^{\pm0.72}$ |
| **CodecSep + dnr-v2** | SI-SDR | $\mathbf{1.2^{\pm3.29}}$ | $\mathbf{10.0^{\pm2.92}}$ | $\mathbf{0.9^{\pm4.22}}$ |
| | ViSQOL | $\mathbf{2.9^{\pm0.57}}$ | $\mathbf{3.2^{\pm0.45}}$ | $2.3^{\pm0.73}$ |
| **CodecSep + dnr-v2** | SI-SDR | $-0.2^{\pm3.55}$ | $8.3^{\pm2.60}$ | $-1.0^{\pm4.20}$ |
| (**codes in : codes out, zero-shot**) | ViSQOL | $2.5^{\pm0.52}$ | $3.0^{\pm0.44}$ | $2.3^{\pm0.67}$ |

**Does the code-stream variant remain competitive without fine-tuning (cf. Table 1)?** Again, the answer is largely yes. Our *bitstream-native* variant—CodecSep+dnr-v2 (codes in: codes out, zero-shot)—evaluates the same trained masker directly on reconstructed embeddings $E$ from code streams (§3.4) *without* any fine-tuning. Relative to the continuous-latent path, this incurs only a modest drop (roughly 1–2 dB SI-SDR across sources, with small ViSQOL deltas for music/speech and parity on SFX), yet it still surpasses AudioSep+dnr-v2 on SI-SDR for all three sources (music: $-0.2$ vs. $-5.6$ dB; speech: 8.3 vs. 7.7 dB; SFX: $-1.0$ vs. $-4.7$ dB). Compared to pretrained AudioSep (zero-shot), the codes-in:codes-out variant improves SI-SDR on speech and music, although it lags on SFX SI-SDR and ViSQOL. This shows that a deployment-friendly, *no-finetuning* bitstream path is already competitive; as discussed in §3.4, the remaining gap could be reduced with light fine-tuning on $E$ or an embedding-consistency loss.

**How does CodecSep compare with other text-guided baselines on dnr-v2 (cf. Table 1)?** Beyond AudioSep, CodecSep also outperforms the USS-pretrained BiModalSS model and the retrained text-conditioned Sudo rm-rf + FiLM baseline. The heavier attention-based conditioning used in BiModalSS does not generalize well to the open-domain mixtures in dnr-v2, leading to degraded performance. CodecSep also outperforms Sudo rm-rf + FiLM under the same universal setting, likely for two reasons. First, continuous CLAP embeddings provide much richer semantic conditioning than the fixed one-hot or multi-hot vectors that Sudo rm-rf + FiLM was originally designed for, making open-domain prompting more difficult for that architecture. Second, applying FiLM across all U-Conv blocks while relying on only a single Conv1d layer for audio encoding can destabilize internal representations when conditioned with high-dimensional continuous embeddings. Since AudioSep still outperforms both of these baselines by a substantial margin, we use AudioSep as the primary baseline in the remaining experiments.

**Further benchmarking on ESC-50, Clotho-v2, AudioSet, VGGSound, & AudioCaps (cf. Table 2).** Extending beyond dnr-v2, we evaluate both systems on five additional open-domain benchmarks spanning environmental sounds (ESC-50), audio-captioning-style corpora (Clotho-v2, AudioCaps), weakly labeled web-scale audio (AudioSet), and visually grounded audio (VGGSound). Under matched training data and prompting protocols, CodecSep+dnr-v2 yields consistently stronger separation performance than AudioSep+dnr-v2 across all five datasets in SI-SDR, with mean gains of +1.88 dB on ESC-50, +2.4 dB on Clotho-v2, +1.25 dB on AudioSet, +0.92 dB on VGGSound, and +0.30 dB on AudioCaps. The corresponding 95% confidence intervals for SI-SDR exclude zero in every case, indicating that the observed improvements are consistently positive rather than arising from a small number of favorable examples.

Table 2: Results: Benchmarking on **ESC-50**, **Clotho-v2**, **AudioSet**, **VGGSound**, **AudioCaps**. To reflect the compressed-input deployment scenario studied in this work, the retrained baseline is both trained and inferred on *codec-processed* audio. Reported significance statistics compare **CodecSep+dnr-v2** against **AudioSep+dnr-v2** using paired tests over evaluation examples.

| Model | Metric (↑) | ESC-50 | Clotho-v2 | AudioSet | VGGSound | AudioCaps |
|---|---|---|---|---|---|---|
| AudioSep | SI-SDR | $-7.8^{\pm14.46}$ | $-8.6^{\pm17.0}$ | $-7.6^{\pm11.42}$ | $-7.0^{\pm12.65}$ | $-6.4^{\pm11.48}$ |
| + dnr-v2 | ViSQOL | $2.3^{\pm1.12}$ | $2.1^{\pm1.08}$ | $2.1^{\pm1.00}$ | $2.2^{\pm1.10}$ | $\mathbf{2.3^{\pm1.08}}$ |
| **CodecSep** | SI-SDR | $\mathbf{-5.9^{\pm11.55}}$ | $\mathbf{-6.0^{\pm11.10}}$ | $\mathbf{-6.4^{\pm10.53}}$ | $\mathbf{-6.1^{\pm12.12}}$ | $\mathbf{-6.1^{\pm11.62}}$ |
| **+ dnr-v2** | ViSQOL | $\mathbf{2.3^{\pm1.13}}$ | $\mathbf{2.3^{\pm1.09}}$ | $\mathbf{2.2^{\pm1.0}}$ | $\mathbf{2.3^{\pm1.11}}$ | $2.2^{\pm1.16}$ |
| **Statistical Significance** | | | | | | |
| *Mean gain* | SI-SDR | +1.88 | +2.4 | +1.25 | +0.92 | +0.30 |
| | ViSQOL | +0.03 | +0.22 | +0.04 | +0.02 | −0.02 |
| *95% CI of gain* | SI-SDR | [1.61, 2.14] | [2.08, 2.72] | [1.11, 1.39] | [0.78, 1.06] | [0.23, 0.46] |
| | ViSQOL | [0.02, 0.04] | [0.18, 0.28] | [0.03, 0.05] | [0.01, 0.03] | [−0.03, −0.01] |
| *Paired t-test p-value* | SI-SDR | $2.77\times10^{-43}$ | $9.41\times10^{-48}$ | $8.04\times10^{-66}$ | $7.5\times10^{-39}$ | $9.72\times10^{-13}$ |
| | ViSQOL | $8.26\times10^{-7}$ | $4.36\times10^{-5}$ | $1.08\times10^{-11}$ | $3.23\times10^{-17}$ | $1.21\times10^{-4}$ |
| *Wilcoxon p-value* | SI-SDR | $1.18\times10^{-106}$ | $4.29\times10^{-52}$ | $5.35\times10^{-67}$ | $2.1\times10^{-88}$ | $3.23\times10^{-17}$ |
| | ViSQOL | $5.19\times10^{-21}$ | $1.44\times10^{-8}$ | $4.46\times10^{-12}$ | $1.61\times10^{-71}$ | $3.44\times10^{-6}$ |

A similar trend is observed in ViSQOL on four of the five benchmarks, where CodecSep attains positive mean gains of +0.03, +0.22, +0.04, and +0.02 on ESC-50, Clotho-v2, AudioSet, and VGGSound, respectively. These gains are again supported by confidence intervals that remain strictly above zero. On AudioCaps, by contrast, the ViSQOL difference is marginal and slightly favors AudioSep (−0.02), indicating that the two systems are broadly comparable on this perceptual metric for that benchmark even though CodecSep remains competitive in SI-SDR.

The paired statistical analysis indicates that the observed improvements are consistent across evaluation examples. For SI-SDR, the paired mean gains are positive on all five benchmarks, and the corresponding 95% confidence intervals exclude zero throughout. Both the paired *t*-test and the Wilcoxon signed-rank test likewise indicate statistically significant differences in favor of CodecSep across all datasets. A similar pattern is observed for ViSQOL on ESC-50, Clotho-v2, AudioSet, and VGGSound, where positive paired gains are again supported by confidence intervals above zero and significant paired tests. On AudioCaps, the ViSQOL difference is small and slightly favors AudioSep, indicating broadly comparable perceptual quality on that benchmark despite CodecSep's competitive separation fidelity. Overall, these results suggest that the gains of CodecSep are not confined to the in-domain dnr-v2 setting, but generalize consistently across a diverse set of open-domain benchmarks.

**Why evaluate non-codec baselines on codec-processed audio?** We route spectrogram- and waveform-domain baselines through codec processing to compare methods in the deployment regime that motivates CodecSep. In realistic edge–server workflows, audio is typically available as codec-compressed bitstreams rather than as ideal raw waveforms. Under such conditions, non-codec separators must first decode the signal, perform separation in the audio or spectrogram domain, and then re-encode if codec-compatible output is required. CodecSep is explicitly designed to avoid this decode–separate–re-encode overhead by operating directly on codec representations. Accordingly, training and evaluating non-codec baselines on codec-processed audio reflects the compressed-input conditions under which these systems would actually be deployed. We stress, however, that this setup is intended to assess *deployment realism* and interface compatibility, rather than to represent the native best-case raw-audio performance of the non-codec baselines.

Table 3: Results on *unprocessed audio*, provided for completeness. Table Table 3a presents separation performance on universal sound separation (**dnr-v2-test**). The zero-shot baselines are evaluated on raw audio in their native setting, whereas the retrained baselines are both trained and evaluated on raw audio. Table 3b presents further raw-audio benchmarking results on **ESC-50**, **Clotho-v2**, **AudioSet**, **VGGSound**, and **AudioCaps** for AudioSep retrained on **dnr-v2**.

(a) Separation performance on universal sound separation (**dnr-v2-test**) with unprocessed audio. Zero-shot baselines are evaluated in their native raw-audio setting, whereas retrained baselines are both trained and evaluated on raw audio.

| Model | Metric ($\uparrow$) | Music | Speech | Sfx |
|---|---|---|---|---|
| AudioSep | SI-SDR | $-1.1^{\pm 3.72}$ | $5.4^{\pm 3.46}$ | $0.5^{\pm 4.41}$ |
| (zero-shot) | ViSQOL | $3.1^{\pm 0.52}$ | $3.2^{\pm 0.46}$ | $2.7^{\pm 0.63}$ |
| BiModalSS | SI-SDR | $-6.0^{\pm 2.48}$ | $2.2^{\pm 2.31}$ | $-5.45^{\pm 3.21}$ |
| (zero-shot) | ViSQOL | $2.5^{\pm 0.49}$ | $2.7^{\pm 0.43}$ | $2.3^{\pm 0.66}$ |
| AudioSep + dnr-v2 | SI-SDR | $-4.9^{\pm 2.61}$ | $7.9^{\pm 2.52}$ | $-3.9^{\pm 3.08}$ |
| | ViSQOL | $2.7^{\pm 0.47}$ | $2.6^{\pm 0.35}$ | $2.3^{\pm 0.61}$ |
| Sudo rm-rf + FiLM + dnr-v2 | SI-SDR | $-5.9^{\pm 2.36}$ | $3.1^{\pm 2.18}$ | $-5.7^{\pm 3.27}$ |
| | ViSQOL | $2.8^{\pm 0.50}$ | $3.0^{\pm 0.39}$ | $2.3^{\pm 0.65}$ |

(b) Further raw-audio benchmarking of AudioSep retrained on **dnr-v2** across **ESC-50**, **Clotho-v2**, **AudioSet**, **VGGSound**, and **AudioCaps**.

| Model | Metric ($\uparrow$) | ESC-50 | Clotho-v2 | AudioSet | VGGSound | AudioCaps |
|---|---|---|---|---|---|---|
| AudioSep | SI-SDR | $-7.0^{\pm 14.31}$ | $-8.1^{\pm 16.72}$ | $-7.2^{\pm 11.09}$ | $-6.4^{\pm 12.48}$ | $-5.9^{\pm 11.21}$ |
| + dnr-v2 | ViSQOL | $2.3^{\pm 1.09}$ | $2.2^{\pm 1.05}$ | $2.1^{\pm 0.98}$ | $2.3^{\pm 1.07}$ | $2.3^{\pm 1.03}$ |

**Native raw-audio performance of non-codec baselines (cf. Table 3).**  For completeness, we also report the performance of the non-codec baselines in their native *raw-audio* setting, without codec processing (cf. Table 3). These results help separate two effects: the intrinsic separation capability of the baseline architectures in their preferred operating regime, and their performance under the codec-mediated deployment setting that motivates CodecSep. On **dnr-v2-test** (cf. Table 3a), evaluating AudioSep on raw audio improves its absolute scores relative to the codec-processed setting, as expected, since the model is no longer exposed to codec-induced mismatch. A similar trend is observed for the retrained AudioSep+dnr-v2 and the other non-codec baselines. Even so, CodecSep+dnr-v2 evaluated in the codec-processed regime (Table 1) still exceeds the raw-audio AudioSep baselines in SI-SDR on all three dnr-v2 stems, while remaining broadly competitive in perceptual quality, though the raw-audio AudioSep zero-shot model retains an advantage in ViSQOL on music and SFX. Likewise, on the additional open-domain benchmarks (cf. Table 3b), AudioSep+dnr-v2 evaluated on unprocessed audio yields modestly stronger absolute scores than its codec-processed counterpart. However, CodecSep+dnr-v2 in the codec-processed setting (Table 2) still maintains consistently stronger SI-SDR across all five datasets, with broadly comparable ViSQOL and only a small deficit on AudioCaps. These raw-audio results clarify that codec processing does affect non-codec separators in absolute terms. At the same time, they do not alter the main claim of this work: CodecSep is designed specifically for the *codec-mediated deployment* regime, where audio is exchanged as compressed bitstreams. Our primary comparisons therefore remain those in Tables 1 and 2, which evaluate all methods under the compressed-input conditions relevant to that deployment setting.

**Subjective evaluation (MOS–LQS) (cf. Table 4).**  We ran a human evaluation test with $n{=}20$ participants on 20 dnr-v2 3-stem test mixtures, comparing paired outputs from CodecSep+dnr-v2 and the official AudioSep model using fixed *speech*/*music* prompts and per-clip *sfx* prompts. We used the official pretrained *AudioSep* model, rather than our retrained variant, because the pretrained checkpoint performed

Table 4: Subjective evaluation (MOS–LQS) on **dnr-v2** 3-stem mixtures. Mean $\pm$ standard deviation across $n$=20 raters and 20 test clips. Each stem was rated independently (1=bad, 5=excellent). Statistical significance is computed using paired tests on *per-clip* mean ratings ($n$=20 clips), comparing CodecSep against AudioSep.

| Model | Overall ($\uparrow$) | Music ($\uparrow$) | Speech ($\uparrow$) | Sfx ($\uparrow$) |
|---|---|---|---|---|
| AudioSep | $2.61 \pm 1.04$ | $2.49 \pm 0.95$ | $2.50 \pm 1.02$ | $2.84 \pm 1.16$ |
| **CodecSep + dnr-v2** | $\mathbf{3.34 \pm 1.00}$ | $\mathbf{3.17 \pm 1.01}$ | $\mathbf{3.49 \pm 1.00}$ | $\mathbf{3.37 \pm 0.97}$ |
| **Statistical Significance** | | | | |
| *Mean gain* | $+0.74$ | $+0.68$ | $+0.99$ | $+0.53$ |
| *95% CI of gain* | $[0.55, 0.92]$ | $[0.47, 0.91]$ | $[0.78, 1.20]$ | $[0.20, 0.84]$ |
| *Paired t-test p-value* | $9.78 \times 10^{-8}$ | $2.74 \times 10^{-6}$ | $6.55 \times 10^{-9}$ | $2.81 \times 10^{-3}$ |
| *Wilcoxon p-value* | $1.03 \times 10^{-4}$ | $2.29 \times 10^{-4}$ | $8.82 \times 10^{-5}$ | $8.48 \times 10^{-3}$ |

substantially better on *dnr-v2*; notably, this pretrained model was trained on approximately 14,100 hours of audio from multiple datasets with diverse natural-language prompts. Raters scored each stem independently in randomized order on the MOS–LQS scale (1=bad, 5=excellent); we report mean $\pm 1\sigma$. As shown in Table 4, CodecSep scored $3.34^{\pm 1.00}$ vs. AudioSep $2.61^{\pm 1.04}$ overall. By source, CodecSep achieved $3.17^{\pm 1.01}$ (music), $3.37^{\pm 0.97}$ (sfx), and $3.49^{\pm 1.00}$ (speech), while AudioSep obtained $2.49^{\pm 0.95}$, $2.84^{\pm 1.16}$, and $2.50^{\pm 1.02}$, respectively. These outcomes align with objective trends (SI-SDR/ViSQOL) and indicate consistent perceptual gains for CodecSep.

We further assessed statistical significance using paired tests on the *per-clip* MOS–LQS means over 20 test clips. CodecSep significantly outperformed AudioSep on all stems as well as on the overall score. For the overall MOS–LQS, CodecSep improved the clip-level mean from 2.61 to 3.34, yielding a mean paired gain of 0.74 points (95% CI: $[0.55, 0.92]$, paired $t$-test: $p = 9.78 \times 10^{-8}$; Wilcoxon signed-rank: $p = 1.03 \times 10^{-4}$). The improvements were also significant for Music (mean gain: 0.68, 95% CI: $[0.47, 0.91]$, $p = 2.74 \times 10^{-6}$), Speech (mean gain: 0.99, 95% CI: $[0.78, 1.20]$, $p = 6.55 \times 10^{-9}$), and SFX (mean gain: 0.53, 95% CI: $[0.20, 0.84]$, $p = 2.81 \times 10^{-3}$). These findings indicate that the subjective preference for CodecSep is consistent across clips and not driven by a small subset of examples. Paired model outputs and reference stems are included in the supplementary materials for side-by-side listening.

### 4.1.2 Effect of Prompt Granularity on Source Separation Performance

**How should the prompt granularity analysis be interpreted (cf. Tables 1,5)?** We view this experiment primarily as an analysis of *how semantic specificity in the prompt shapes universal source separation*, rather than as a pure leaderboard comparison. To make this concrete, we consider three regimes: (i) *fixed-stem* systems without text guidance (TDANet, Sudo rm-rf, CodecFormer, SDCodec), which serve only as closed-set reference points; (ii) *generic 3-stem prompting* using {"music", "speech", "sfx"}; and (iii) *universal prompting* that keeps generic prompts for speech and music but replaces the coarse "sfx" label with fine-grained, compositional SFX descriptions. For the text-guided models, separate versions of CodecSep and AudioSep are trained and evaluated under each prompt regime, so the comparison is matched within each setting.

**What changes when detailed SFX prompts are used during training (cf. Tables 1,5)?** For CodecSep, we observe that training with finer-grained SFX descriptions improves not only the queried SFX stem, but also the overall separation behavior of the system. In particular, moving from generic prompts to detailed SFX supervision improves SFX extraction while also yielding better speech and music results, including perceptual quality and SI-SDR. We interpret this as evidence that richer semantic supervision helps the model partition the mixture more cleanly at the scene level, rather than merely refining the target SFX estimate in isolation. In other words, more informative prompts appear to make the separation problem better specified for the model as a whole.

Table 5: Results: Impact of SFX Prompt Granularity on Universal Sound Separation (**dnr-v2-test**)

| Model | Metric (↑) | Music | Speech | Sfx |
|---|---|---|---|---|
| **3-Stem: Fixed stem baselines (no text-guidance)** | | | | |
| TDANet | SI-SDR | $1.8^{\pm 3.55}$ | $10.2^{\pm 2.91}$ | $1.4^{\pm 4.90}$ |
| | ViSQOL | $2.9^{\pm 0.58}$ | $3.1^{\pm 0.43}$ | $2.4^{\pm 0.72}$ |
| Sudo rm-rf | SI-SDR | $-0.9^{\pm 4.01}$ | $9.0^{\pm 2.60}$ | $0.6^{\pm 4.74}$ |
| | ViSQOL | $2.7^{\pm 0.59}$ | $2.9^{\pm 0.45}$ | $2.3^{\pm 0.72}$ |
| CodecFormer | SI-SDR | $-5.7^{\pm 3.44}$ | $2.3^{\pm 2.32}$ | $-6.5^{\pm 4.36}$ |
| | ViSQOL | $2.2^{\pm 0.47}$ | $2.5^{\pm 0.49}$ | $2.1^{\pm 0.67}$ |
| SDCodec | SI-SDR | $\mathbf{1.9^{\pm 3.68}}$ | $\mathbf{11.3^{\pm 2.98}}$ | $\mathbf{1.8^{\pm 4.08}}$ |
| | ViSQOL | $\mathbf{3.0^{\pm 0.56}}$ | $\mathbf{3.5^{\pm 0.40}}$ | $\mathbf{2.6^{\pm 0.73}}$ |
| **3-Stem: {"music","speech","sfx"} as generic prompt** | | | | |
| AudioSep | SI-SDR | $\mathbf{-2.5^{\pm 4.06}}$ | $4.9^{\pm 4.21}$ | $-6.7^{\pm 4.73}$ |
| (zero-shot) | ViSQOL | $\mathbf{2.9^{\pm 0.63}}$ | $\mathbf{3.1^{\pm 0.56}}$ | $2.1^{\pm 0.68}$ |
| AudioSep + dnr-v2 | SI-SDR | $-6.2^{\pm 2.77}$ | $\mathbf{7.7^{\pm 3.11}}$ | $-2.1^{\pm 3.90}$ |
| | ViSQOL | $2.6^{\pm 0.57}$ | $2.5^{\pm 0.37}$ | $2.4^{\pm 0.74}$ |
| **CodecSep + dnr-v2** | SI-SDR | $-7.7^{\pm 2.84}$ | $4.6^{\pm 2.48}$ | $\mathbf{0.6^{\pm 4.15}}$ |
| | ViSQOL | $2.5^{\pm 0.55}$ | $2.7^{\pm 0.49}$ | $\mathbf{2.4^{\pm 0.70}}$ |

**Does CodecSep still remain effective under coarse generic prompts (cf. Table 5)?** Yes. Under the matched generic-prompt setting, CodecSep still remains competitive with spectrogram-domain AudioSep, even though the gains are not uniform across all stems and metrics. In this regime, CodecSep achieves stronger speech perceptual quality, better SFX SI-SDR, and broadly comparable SFX perceptual quality, while music remains a weaker case and does not surpass AudioSep. We therefore do not interpret the generic-prompt results as showing a uniform advantage for CodecSep across the board. Rather, they show that CodecSep retains strong performance in a coarser semantic setting and continues to compare favorably on key aspects of speech and SFX separation, indicating that its effectiveness does not depend entirely on unusually detailed prompt engineering. Relative to the fixed-stem baselines, the comparison is naturally mixed, since those systems solve a more restrictive closed-set problem and are included here mainly as reference points rather than as the primary target of this analysis.

**Overall interpretation.** Taken together, these results suggest that prompt granularity is an important design variable for universal sound separation. For CodecSep, replacing a generic "sfx" label with finer-grained SFX descriptions improves not only the target SFX stem, but also speech and music quality, including perceptual measures and SI-SDR. At the same time, the generic-prompt setting shows that CodecSep remains effective even under coarser supervision, although the benefits are more selective and do not extend uniformly to every stem, particularly music. We therefore view the fixed-stem baselines as useful closed-set reference points, while the main conclusion of this analysis is that finer-grained semantic supervision strengthens universal prompt-conditioned separation and makes the gains from text guidance more consistent across the mixture.

These controlled studies cover multiple prompt granularities, but they also suggest a broader direction: training on larger and more diverse corpora with a wider spectrum of prompt specificities may yield further gains, which we leave for future work.

Table 6: Results: Architectural advantages in using CodecFormer decoder as masker (**dnr-v2-test**)

| Model | Metric ($\uparrow$) | Music | Speech | Sfx |
|---|---|---|---|---|
| CodecFormer | SI-SDR | $-5.8^{\pm 3.44}$ | $2.3^{\pm 2.32}$ | $-6.5^{\pm 4.36}$ |
| | ViSQOL | $2.2^{\pm 0.47}$ | $2.5^{\pm 0.49}$ | $2.1^{\pm 0.67}$ |
| **CodecSep + dnr-v2** | SI-SDR | $1.2^{\pm 3.35}$ | $10.0^{\pm 2.91}$ | $0.9^{\pm 4.18}$ |
| **(unguided, 3-stem)** | ViSQOL | $2.8^{\pm 0.55}$ | $3.1^{\pm 0.45}$ | $\mathbf{2.5^{\pm 0.72}}$ |
| **CodecSep + dnr-v2** | SI-SDR | $\mathbf{1.2^{\pm 3.29}}$ | $\mathbf{10.0^{\pm 2.92}}$ | $\mathbf{0.9^{\pm 4.22}}$ |
| **(text-guided)** | ViSQOL | $\mathbf{2.9^{\pm 0.57}}$ | $\mathbf{3.2^{\pm 0.45}}$ | $2.3^{\pm 0.73}$ |
| **CodecSep + dnr-v2** | SI-SDR | $-6.8^{\pm 2.77}$ | $2.0^{\pm 2.84}$ | $-6.8^{\pm 3.83}$ |
| (ablate **Masker**) | ViSQOL | $2.5^{\pm 0.58}$ | $2.6^{\pm 0.50}$ | $2.1^{\pm 0.74}$ |

### 4.1.3 Architectural Choice in Codec Latent Space: Masking vs. Generation

**Is the Transformer masker itself necessary (cf. Table 6)?** The ablation suggests that it is. The lightweight CodecSep+dnr-v2 (ablate Masker) removes the transformer masker and applies FiLM directly to the encoder. While this variant attains SI-SDR comparable to AudioSep+dnr-v2 (cf. Table 1) and yields better perceptual speech quality, its overall separation quality drops relative to full CodecSep. This supports our architectural choice: FiLM modulation is more effective when applied in a dedicated masker than when injected directly into the encoder, where it perturbs the mixture latents themselves.

**Is masking better than generation in codec latent space (cf. Table 6)?** To isolate this architectural question, we compare (i) CodecFormer, which performs decoder-style source generation, (ii) CodecSep (unguided, 3-stem), which repurposes the CodecFormer Transformer as a masker over codec latents, and (iii) CodecSep (text-guided), which adds prompt conditioning on top of the same masking formulation.

**What changes when decoder-style generation is replaced by masking?** The results on dnr-v2-test show a clear and consistent pattern: replacing decoder-style generation with masking strengthens separation across music, speech, and SFX. This supports our design rationale that, in the DAC latent domain, it is more effective to modulate existing, semantically structured content than to synthesize new source latents from scratch. In particular, masking reduces artifacts and cross-talk leakage, preserves long-range periodicity, timbre, and transient organization already encoded by the codec, and yields a more stable optimization than end-to-end generation.

**What is gained by adding text guidance on top of masking?** Text conditioning provides a further uniform improvement over the unguided masker. This suggests that once separation is formulated as latent selection, semantic prompting can steer that selection more precisely. Put differently, the masker formulation concentrates Transformer capacity on deciding *where* and *how much* information to pass, rather than *what* new content to generate. This is exactly the operating regime we want in structured codec latent space.

### 4.1.4 Qualitative Evidence for Structured Source Organization in Codec Latent Space

**Does the codec latent space exhibit source-dependent structure?** Figure. 3 provides direct qualitative evidence that the frozen codec encoder organizes mixtures into a latent representation that already preserves source-discriminative structure. The reference source latents for speech (Figure. 3c), music (Figure. 3e), and SFX (Figure. 3g) display visibly different activation patterns across channels and time, rather than appearing as weak perturbations of a shared, unstructured representation. This observation supports the core hypothesis underlying CodecSep: modern neural audio codec latents are sufficiently structured for source extraction to be performed through masking, without requiring source generation or explicit re-encoding.

**What do the estimated masks reveal about the separation mechanism?** The most striking feature of the learned masks is that they are explicitly *channel-wise*. As seen in Figures. 3d, 3f, and 3h, each estimated mask forms nearly horizontal bands with essentially constant values over time. In other words, the masks

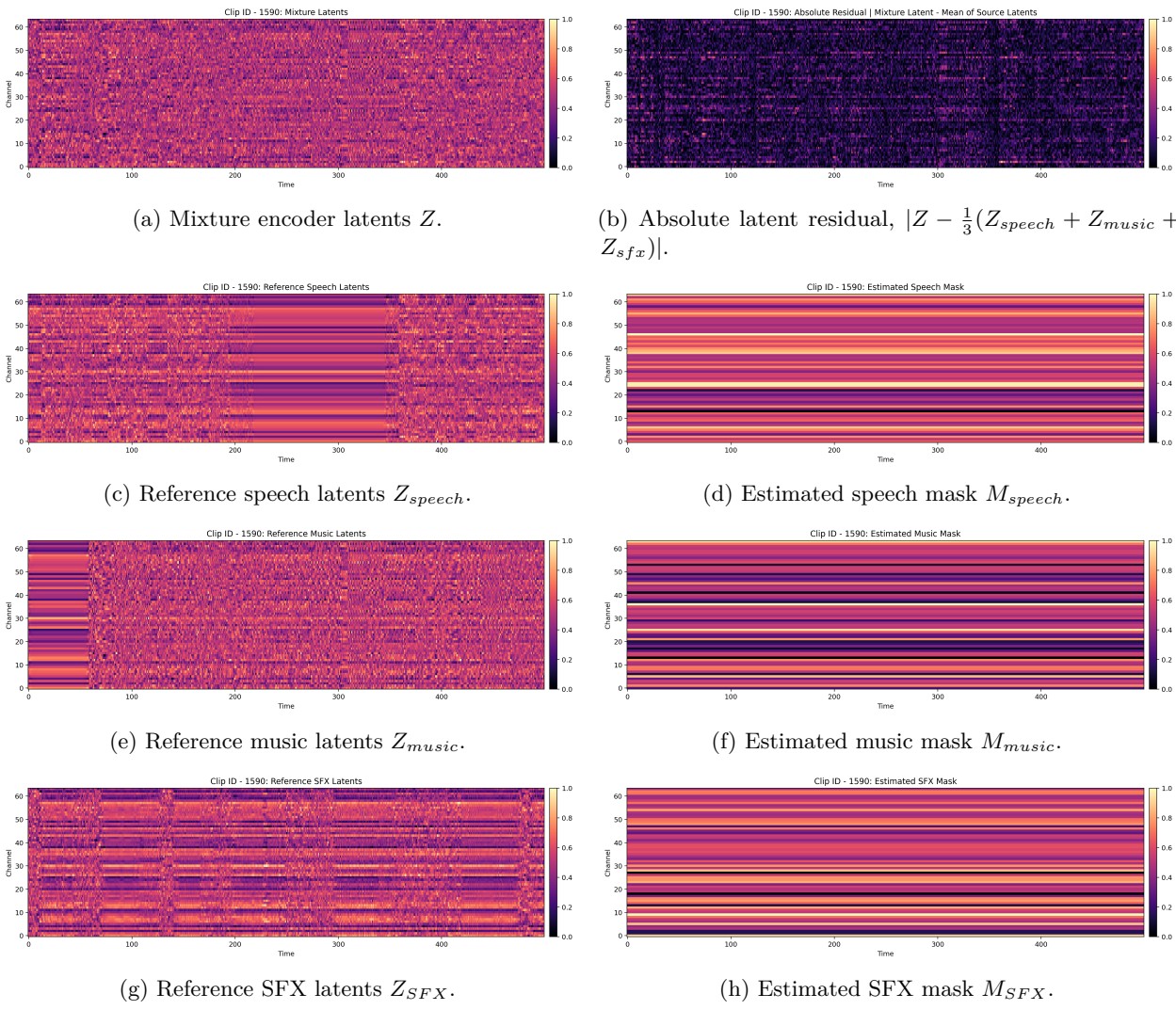

(a) Mixture encoder latents $Z$.

(b) Absolute latent residual, $|Z - \frac{1}{3}(Z_{speech} + Z_{music} + Z_{sfx})|$.

(c) Reference speech latents $Z_{speech}$.

(d) Estimated speech mask $M_{speech}$.

(e) Reference music latents $Z_{music}$.

(f) Estimated music mask $M_{music}$.

(g) Reference SFX latents $Z_{SFX}$.

(h) Estimated SFX mask $M_{SFX}$.

Figure 3: Qualitative latent-space analysis for a representative dnr-v2 mixture (clip 1590). We visualize the mixture encoder latent $Z$ in Fig. 3a, the corresponding reference source latents for speech $Z_{\text{speech}}$, music $Z_{\text{music}}$, and SFX $Z_{\text{SFX}}$ in Figs. 3c, 3e, and 3g, the estimated source-conditioned masks $M_{\text{speech}}$, $M_{\text{music}}$, and $M_{\text{SFX}}$ in Figs. 3d, 3f, and 3h, and the absolute residual $\left| Z - \frac{1}{3}(Z_{\text{speech}} + Z_{\text{music}} + Z_{\text{SFX}}) \right|$ in Fig. 3b. The reference source latents exhibit visibly different patterns across sources. By contrast, the estimated masks are predominantly channel-wise: each mask remains nearly constant along the temporal axis and varies mainly across latent channels. This suggests that CodecSep separates sources primarily through source-conditioned reweighting and attenuation of latent channels rather than strongly time-localized masking. The residual remains sparse over broad regions of the latent plane, providing qualitative support for the view that the mixture latent retains a meaningful relationship to the latent organization of its constituent sources, without implying exact disentanglement or linear compositionality. The corresponding audio clip and separated outputs are included in the supplementary material for side-by-side listening.

do not exhibit meaningful temporal variation; instead, they act by selecting and attenuating specific latent channels according to the text query. This qualitative behavior is important, because it shows that CodecSep does not separate sources by applying sharply localized temporal gating. Rather, the model operates by modulating a structured latent basis in a source-dependent manner. This also helps explain the slight residual leakage of non-target sources that can still be heard in some separated outputs in the supplementary audio examples: because the masks re-weight shared latent channels rather than enforcing perfectly source-exclusive,

Table 7: Oracle codec reconstruction versus separated outputs on **dnr-v2-test**. The DAC (Oracle) rows report codec-only reconstruction of the corresponding reference signals with *no separation model*: the mixture waveform and each ground-truth stem are passed independently through the frozen DAC encoder–decoder and then evaluated against their original references. This provides a codec-limited reference that isolates distortion introduced by the codec itself. The CodecSep + dnr-v2 rows report the actual separated outputs obtained by prompt-guided masking in codec latent space. Comparing the two clarifies how much degradation is attributable to the codec bottleneck versus the separation stage.

| Model | Metric ($\uparrow$) | Mixture | Music | Speech | Sfx |
|---|---|---|---|---|---|
| **DAC** | SI-SDR | $6.3^{\pm 2.35}$ | $6.8^{\pm 4.3}$ | $7.4^{\pm 3.1}$ | $1.8^{\pm 5.4}$ |
| **(Oracle)** | ViSQOL | $4.1^{\pm 0.15}$ | $3.8^{\pm 0.27}$ | $4.2^{\pm 0.15}$ | $3.8^{\pm 0.33}$ |
| **CodecSep + dnr-v2** | SI-SDR | $4.1^{\pm 2.06}$ | $1.2^{\pm 3.29}$ | $\mathbf{10.0^{\pm 2.92}}$ | $0.9^{\pm 4.22}$ |
| **(text-guided)** | ViSQOL | $3.7^{\pm 0.22}$ | $2.9^{\pm 0.57}$ | $3.2^{\pm 0.45}$ | $2.3^{\pm 0.73}$ |

time-localized suppression, channels that carry partially overlapping information may remain weakly active in the reconstructed signal.

**What does the residual analysis show?** To further examine the structured-latent hypothesis, we compare the mixture latent with the mean of the corresponding reference source latents through the absolute residual

$$\left| Z - \frac{1}{3}\left( Z_{\text{speech}} + Z_{\text{music}} + Z_{\text{SFX}} \right) \right|. \tag{13}$$

The residual remains sparse over broad regions, suggesting that the mixture latent retains a visible relationship to the latent organization of its constituent sources, rather than appearing completely unstructured. This provides *qualitative support* for the view that the codec latent space preserves some source-related organization. At the same time, we do not interpret this as evidence of exact source disentanglement or linear compositionality, especially because DAC was not trained with an explicit source-disentanglement objective. In dnr-v2, mixtures are generated through a non-trivial mixing and normalization pipeline rather than by simple uniform averaging of the constituent source waveforms (cf. Appendix C). Consequently, the resulting mixture latent need not coincide exactly with an equal-weight combination of the source latents, and some residual activity is therefore expected.

**What is the main takeaway from this analysis?** Taken together, these heatmaps provide direct qualitative support for the design premise of CodecSep. The codec latent space is not merely compact; it also appears to preserve meaningful source-dependent organization, and the learned masks exploit this structure primarily through channel-wise modulation. This qualitative evidence complements the architectural results in Table 6, where masking-based separation outperforms decoder-style latent generation, and supports our interpretation that source extraction in CodecSep is enabled by structured source organization already present in the codec representation. For transparency and independent inspection, the corresponding clip and separated outputs are provided in the supplementary material for side-by-side listening.

### 4.1.5 Oracle Codec Reconstruction vs. Separated Output

**How much of the observed distortion comes from the codec bottleneck itself (cf. Table 7)?** To address this directly, we compare CodecSep's separated outputs against a codec-only oracle reference. In the oracle setting, no separator is used: each clean reference signal—the mixture waveform and each ground-truth source stem—is independently passed through the frozen DAC encoder–decoder, and the reconstructed signal is then evaluated against its original reference. This isolates the distortion introduced purely by the codec backbone and therefore provides a useful reference point for interpreting the artifacts seen in separated outputs.

**What does the oracle comparison show?** The DAC oracle achieves strong reconstruction quality on the mixture and reference stems, with SI-SDR / ViSQOL of 6.3/4.1 for the mixture, 6.8/3.8 for music, 7.4/4.2

Table 8: Results: Using ambiguous prompts for Speech and Music (**dnr-v2-test**)

| Model | Metric (↑) | Music | Speech |
|---|---|---|---|
| AudioSep + dnr-v2 | SI-SDR | $-6.4^{\pm 3.29}$ | $4.1^{\pm 3.77}$ |
| | ViSQOL | $2.5^{\pm 0.57}$ | $2.6^{\pm 0.47}$ |
| **CodecSep + dnr-v2** | SI-SDR | $\mathbf{-5.6^{\pm 3.61}}$ | $\mathbf{4.2^{\pm 4.18}}$ |
| | ViSQOL | $\mathbf{2.6^{\pm 0.58}}$ | $\mathbf{2.7^{\pm 0.51}}$ |

for speech, and 1.8/3.8 for SFX. These values indicate that the codec itself is not lossless, and that part of the degradation observed in CodecSep outputs is inherited from the frozen DAC reconstruction bottleneck. This is especially clear in ViSQOL, where CodecSep remains below the oracle across all categories.

**What artifacts are due to separation rather than the codec alone?** Comparing CodecSep against the oracle shows that the remaining gap is source-dependent. For mixture reconstruction, CodecSep trails the oracle (4.1 vs. 6.3 SI-SDR; 3.7 vs. 4.1 ViSQOL), indicating that the separation process introduces additional distortion beyond codec reconstruction alone. A similar pattern holds for music and SFX, where CodecSep remains below the oracle in both SI-SDR and ViSQOL. This suggests that these stems are still limited primarily by separation error—e.g., incomplete source selection, residual interference, or masking imprecision—rather than by the codec alone.

At the same time, this gap should not be interpreted as arising solely from separator imperfections. In dnr-v2, the separated outputs are generated from *mixtures*, whereas the oracle source rows are obtained by reconstructing the corresponding *clean source stems* directly. Because the dnr-v2 mixtures are created through a non-trivial mixing and normalization procedure (cf. Appendix C), recovering a target stem from the mixture is inherently more challenging than direct codec reconstruction of an isolated clean source. The oracle comparison therefore isolates codec-induced distortion, but the remaining gap also reflects the intrinsic difficulty of mixture-conditioned source recovery on dnr-v2.

**Why can speech SI-SDR exceed the oracle?** Speech shows a different pattern: CodecSep attains higher SI-SDR than the DAC oracle (10.0 vs. 7.4), while still remaining clearly below it in ViSQOL (3.2 vs. 4.2). This is best understood as a difference between *signal-level* and *perceptual* evaluation. The DAC oracle reflects codec reconstruction fidelity of the clean speech stem and therefore captures waveform distortion introduced by the codec. By contrast, SI-SDR mainly measures target-aligned signal energy after optimal scaling and is not designed to reflect perceptual naturalness. If the separator preserves the dominant speech structure while effectively suppressing interfering mixture components, it can achieve a higher SI-SDR even though the resulting waveform still contains masking artifacts or reduced perceptual fidelity. The lower ViSQOL for CodecSep confirms exactly this point: speech separation is strong in a signal-recovery sense, but perceptual quality remains constrained by codec and masking distortions.

**What is the main takeaway from this analysis?** This experiment clarifies that the observed artifacts in CodecSep arise from multiple factors: the reconstruction bottleneck imposed by the frozen DAC backbone, the difficulty of recovering sources from dnr-v2's loudness-controlled mixtures, and an additional separation gap introduced by imperfect masking. The oracle comparison shows that the codec already preserves strong perceptual quality, while the remaining gap for mixture, music, and SFX indicates that a substantial portion of the error is due to the separation stage itself. For speech, the higher SI-SDR but lower ViSQOL of CodecSep relative to the oracle highlights an important metric distinction: signal-level recovery can improve even when perceptual fidelity remains bounded by codec and masking artifacts. Overall, the results show that CodecSep is not limited by codec reconstruction alone; the residual errors reflect codec-induced distortion, mixture-formation difficulty, and separation-specific imperfections.

### 4.1.6 Source Separation Robustness Under Prompt Paraphrasing

**Does CodecSep remain effective under prompt paraphrasing (cf. Table 8)?** To probe lexical sensitivity, we re-evaluate both CodecSep + dnr-v2 and AudioSep + dnr-v2 on the dnr-v2 test split by replacing

Table 9: Full Inference Compute Benchmarking Across Six Settings on 24GB NVIDIA A-30 GPU

(a) Inference GMACs (↓)

| Model | Audio Stream I/O | Code Stream I/O | Architecture-only |
|---|---|---|---|
| AudioSep | 33.5 | 73.6 | 33.5 |
| Sudo rm-rf | **16.44** | 56.54 | 16.44 |
| **CodecSep** | 41.45 | **1.35** | **1.35** |

*Codec GMACs:* Enc=12.28, Dec=27.82

(b) Inference Time (s) (↓)

| Model | Audio Stream I/O | Code Stream I/O | Architecture-only |
|---|---|---|---|
| AudioSep | **0.33** | 1.63 | 0.33 |
| Sudo rm-rf | 0.51 | 1.81 | 0.51 |
| **CodecSep** | 1.49 | **0.19** | **0.19** |

*Codec Inference Times (s):* Enc=1.16, Dec=0.14

(c) Parameter count (in million) (↓)

| Model | Audio Stream I/O | Code Stream I/O | Architecture-only |
|---|---|---|---|
| AudioSep | 39 | 112.8 | 39 |
| Sudo rm-rf | **5** | 78.8 | **5** |
| **CodecSep** | 90.1 | **16.3** | 16.3 |

*Codec Parameter Count (in million):* Enc=21.5, Dec=52.3

(d) Parameter-only Memory Footprint (MB) (↓)

| Model | Audio Stream I/O | Code Stream I/O | Architecture-only |
|---|---|---|---|
| AudioSep | 156.13 | 447.62 | 156.13 |
| Sudo rm-rf | **20.07** | 311.25 | **20.07** |
| **CodecSep** | 339.89 | **48.4** | 48.4 |

*Codec Parameter-only Memory Footprint (MB):* Enc=85.1, Dec=206.39

(e) Forward/Backward Pass Memory Footprint (MB) (↓)

| Model | Audio Stream I/O | Code Stream I/O | Architecture-only |
|---|---|---|---|
| AudioSep | **804.3** | 1580.6 | 804.3 |
| Sudo rm-rf | 2032.13 | 2836.43 | 2032.13 |
| **CodecSep** | 804.4 | **28.06** | **28.06** |

*Codec Forward/Backward Pass Memory Footprint (MB):* Enc=310.48, Dec=465.82

(f) Full Memory Footprint (MB) (↓)

| Model | Audio Stream I/O | Code Stream I/O | Architecture-only |
|---|---|---|---|
| AudioSep | **960.43** | 2028.22 | 960.43 |
| Sudo rm-rf | 2052.2 | 3147.68 | 2052.2 |
| **CodecSep** | 1144.25 | **76.46** | **76.46** |

*Codec Full Memory Footprint (MB):* Enc=395.58., Dec=672.21

the generic training-time prompts for *speech* and *music* with three unseen paraphrases per class—*speech*: {"spoken voice", "human conversation", "people talking"}; *music*: {"instrumental music", "band playing", "melody with instruments"}. This constitutes a zero-shot paraphrase generalization test: the models are trained with generic category cues but must respond to synonymic, potentially broader descriptors at inference.

**How do CodecSep and AudioSep behave under lexical variation?** Both exhibit the expected degradation when moving from generic to paraphrased prompts, confirming that lexical ambiguity weakens query–audio alignment. However, CodecSep degrades more gracefully overall, maintaining stronger separation and perceptual quality for *speech*, while also retaining a small but persistent advantage for *music*. Although the gap between the two systems narrows under paraphrasing, the relative ranking is preserved. This suggests that FiLM-conditioned masking over structured codec latents confers a degree of robustness to synonym-level prompt shifts.

**What does this experiment establish, and what does it not test?** This experiment isolates lexical paraphrases only. We do not consider prompts with explicit temporal or relational structure (e.g., "applause follows a song"), which would require the model to respond not just to synonymy but also to event ordering or compositional temporal cues. We leave such prompt variations for future work.

### 4.1.7 Efficiency in Code-Stream Deployment

**What is gained in code-stream deployment (cf. Table 9)?** The principal efficiency advantage of CodecSep appears in the *code-stream* setting, where separation is performed directly on codec representations rather than on decoded waveforms. This is the practically relevant regime for edge–server systems, where audio is typically transmitted, stored, and exchanged as codec bitstreams. In this setting, CodecSep is markedly more efficient than spectrogram-domain and waveform-domain separators across all measured axes. We therefore interpret the headline efficiency gains of CodecSep primarily as a *deployment-level* advantage in codec-mediated pipelines, rather than as a universal claim about raw-audio separation cost.

**How large are the compute and latency gains in code-stream mode?** In terms of hardware-agnostic compute (cf. Table 9a), CodecSep requires only 1.35 GMACs when operating on codec bitstreams, compared to 73.6 GMACs for AudioSep and 56.5 GMACs for Sudo rm-rf, yielding roughly 54× and 40× reductions, respectively (and 25×/12× under the architecture-only comparison). These savings translate directly to latency (cf. Table 9b): CodecSep achieves 0.19 s inference in code-stream mode—about 8× faster than AudioSep and approximately 10× faster than Sudo rm-rf.

Table 10: Results: Extending CodecSep to 48 kHz full-band (**dnr-v2-test**)

| Model | Sampling Rate | Metric (↑) | Music | Speech | Sfx |
|---|---|---|---|---|---|
| AudioSep (zero-shot) | 32 kHz | SI-SDR | $-2.5^{\pm 4.06}$ | $4.9^{\pm 4.21}$ | $-0.3^{\pm 5.39}$ |
| | | ViSQOL | $2.9^{\pm 0.63}$ | $3.1^{\pm 0.56}$ | $\mathbf{2.6^{\pm 0.77}}$ |
| AudioSep + dnr-v2 | 32 kHz | SI-SDR | $-5.6^{\pm 2.89}$ | $7.7^{\pm 3.0}$ | $-4.5^{\pm 3.68}$ |
| | | ViSQOL | $2.6^{\pm 0.57}$ | $2.5^{\pm 0.37}$ | $2.3^{\pm 0.7}$ |
| **CodecSep + dnr-v2** **(DAC Backbone)** | 16 kHz | SI-SDR | $\mathbf{1.2^{\pm 3.29}}$ | $\mathbf{10.0^{\pm 2.92}}$ | $\mathbf{0.9^{\pm 4.22}}$ |
| | | ViSQOL | $\mathbf{2.9^{\pm 0.57}}$ | $\mathbf{3.1^{\pm 0.45}}$ | $2.3^{\pm 0.73}$ |
| **CodecSep + dnr-v2** **(DAC Backbone)** | 24 kHz | SI-SDR | $0.2^{\pm 3.3}$ | $8.8^{\pm 2.9}$ | $0.6^{\pm 4.2}$ |
| | | ViSQOL | $2.7^{\pm 0.56}$ | $3.0^{\pm 0.44}$ | $2.3^{\pm 0.72}$ |
| **CodecSep + dnr-v2** **(DAC Backbone)** | 44.1 kHz | SI-SDR | $-2.3^{\pm 3.27}$ | $5.9^{\pm 2.48}$ | $-0.3^{\pm 3.79}$ |
| | | ViSQOL | $2.5^{\pm 0.46}$ | $2.7^{\pm 0.42}$ | $2.4^{\pm 0.68}$ |
| **CodecSep + dnr-v2** **(EnCodec Backbone)** | 48 kHz | SI-SDR | $-2.8^{\pm 3.5}$ | $5.4^{\pm 2.36}$ | $-0.5^{\pm 3.83}$ |
| | | ViSQOL | $2.4^{\pm 0.5}$ | $2.6^{\pm 0.41}$ | $2.4^{\pm 0.65}$ |

**What is gained in parameter and memory footprint?** Parameter efficiency follows the same trend (cf. Table 9d): when fed code streams, CodecSep uses only 16.3M parameters, substantially smaller than AudioSep (112.8M) and Sudo rm-rf (78.8M), reflecting its lightweight masker-only design. The memory savings are even more pronounced. For forward/backward activations (cf. Table 9e), CodecSep requires only 28 MB in code-stream mode, compared to 1.58 GB for AudioSep and 2.84 GB for Sudo rm-rf—over 50× to 100× reductions. The same pattern holds for full memory footprint (cf. Table 9f): CodecSep uses only 76.5 MB, versus 2.03 GB for AudioSep and 3.15 GB for Sudo rm-rf, corresponding to roughly 27× and 41× reductions, respectively.

**How should the audio-stream results be interpreted?** In the *audio-stream* setting, CodecSep must additionally run the codec encoder and decoder, so its end-to-end compute is *not* lower than AudioSep. Accordingly, the audio-stream comparison should be interpreted as showing that CodecSep remains broadly comparable in this regime, rather than uniformly more efficient. This distinction is important: the main efficiency claim of CodecSep is not that it is cheaper for generic waveform-side separation, but that it becomes substantially more efficient when separation is carried out directly on code streams. Even in the audio-stream setting, however, the results remain informative: CodecSep still uses less memory than Sudo rm-rf (1.14 GB vs. 2.05 GB), indicating that codec overhead, while non-trivial, is not itself the dominant bottleneck.

Taken together, these results clarify the intended efficiency claim of CodecSep. Its main advantage is not a universal reduction in raw-audio separation cost, but a substantial *systems-level* gain in codec-mediated deployment, where operating directly on code streams avoids the decode–separate–re-encode pathway required by conventional baselines. That is precisely the regime in which compute, memory, and latency all improve dramatically, making CodecSep especially well-suited to scalable edge–server pipelines.

### 4.1.8 Bandwidth Scaling: Extending CodecSep to Full-Band Audio

**How should the sampling-rate mismatch with AudioSep be interpreted, and does the masking formulation remain effective as bandwidth increases (cf. Table 10)?** The main CodecSep system in this paper uses a **16 kHz DAC backbone**. This choice is deliberate: our primary objective is to study *low-MAC, codec-mediated separation*, and the 16 kHz DAC variant provides the most favorable tradeoff between separation quality and deployment efficiency. By contrast, the official AudioSep checkpoint is available as a **32 kHz** model trained on a much larger and more diverse corpus (roughly 15K hours), and for fairness we also retrain the same 32 kHz AudioSep variant under our matched training protocol. We do not alter AudioSep to another sampling rate, since doing so would require nontrivial redesign and careful

hyperparameter re-optimization, introducing a new confound by evaluating a materially altered version of the baseline rather than the standard model itself.

At the same time, lower bandwidth can simplify separation, so this mismatch should not simply be ignored. To contextualize it directly, we perform a dedicated *bandwidth-scaling study* for CodecSep by swapping the frozen codec backbone from 16 kHz DAC to higher-bandwidth codecs while keeping the FiLM-conditioned masker and training objective unchanged. Specifically, we evaluate **24 kHz** and **44.1 kHz** DAC variants as well as a **48 kHz EnCodec** backbone. Although our paper targets *mono* separation, the 48 kHz EnCodec experiment is still carried out in the same mono setting so that the comparison remains architectural rather than spatial.

**Why does separation become harder at higher bandwidth?** As the sampling rate $F_s$ increases, the representation must account for progressively richer high-frequency structure, finer temporal transients, and more densely packed spectral detail. In codec latent space, these additional details are typically less cleanly isolated than the lower-frequency coarse structure and tend to become more tightly entangled across sources. As a result, a masker-based separator must make more delicate source-selection decisions over a denser and less separable latent organization. At the same time, the latent sequence length typically increases ($T \uparrow$), which raises both modeling difficulty and compute. Together, these effects make full-band separation intrinsically harder than narrow-band or mid-band separation, even though the underlying masking formulation remains unchanged.

**What pattern is revealed by Table 10?** The results show a clear trend: for a fixed masker capacity, *absolute* SI-SDR and, to a lesser extent, ViSQOL generally decrease as sampling rate increases. Relative to CodecSep at **16 kHz** (1.2/10.0/0.9 SI-SDR and 2.9/3.1/2.3 ViSQOL for music/speech/SFX), the **24 kHz** DAC variant remains strong but drops modestly (0.2/8.8/0.6 SI-SDR; 2.7/3.0/2.3 ViSQOL), while the **44.1 kHz** DAC and **48 kHz** EnCodec variants degrade further ($-2.3/5.9/-0.3$ and $-2.8/5.4/-0.5$ SI-SDR, respectively). This confirms that the 16 kHz operating point is indeed favorable and should not be interpreted as bandwidth-independent evidence. A small exception appears in SFX ViSQOL, which shows a slight improvement at higher bandwidth (e.g., 2.4 at 44.1/48 kHz versus 2.3 at 16/24 kHz). We interpret this cautiously: although overall separation fidelity declines with bandwidth, the additional high-frequency detail available at higher sampling rates may slightly improve the perceptual realism of certain broadband and transient-heavy SFX signals.

**How do the bandwidth-scaled CodecSep variants compare with AudioSep?** The bandwidth study is not only a scaling experiment; it also helps contextualize the sampling-rate mismatch with AudioSep. Against the **retrained 32 kHz AudioSep+dnr-v2** baseline ($-5.6/7.7/-4.5$ SI-SDR; 2.6/2.5/2.3 ViSQOL), **CodecSep at 24 kHz** still performs better on all three stems in SI-SDR and also improves ViSQOL for music and speech while matching SFX ViSQOL. Even the **44.1 kHz** and **48 kHz** CodecSep variants remain substantially stronger than retrained AudioSep on music and SFX SI-SDR, although their speech SI-SDR drops below the 16/24 kHz CodecSep variants.

Relative to the much stronger **pretrained 32 kHz AudioSep** checkpoint ($-2.5/4.9/-0.3$ SI-SDR; 2.9/3.1/2.6 ViSQOL), the comparison is more mixed. **CodecSep at 24 kHz** still exceeds pretrained AudioSep in SI-SDR on all three stems, while the **44.1 kHz** and **48 kHz** CodecSep variants continue to outperform it on speech SI-SDR and remain competitive on music/SFX SI-SDR. In ViSQOL, however, the pretrained AudioSep checkpoint remains especially strong perceptually, particularly for SFX, which is unsurprising given its much larger training corpus. These comparisons therefore suggest two things simultaneously: lower bandwidth does help CodecSep, but the underlying masking formulation remains viable beyond 16 kHz rather than collapsing outside the narrowband case.

**Why do we not report a 32 kHz codec-domain counterpart?** There is currently no publicly available neural audio codec in our setup operating at **32 kHz**, so an exactly matched codec-domain 32 kHz counterpart was not available for a cleaner apples-to-apples comparison. We therefore use the best available higher-bandwidth codec backbones (24, 44.1, and 48 kHz) to test whether the core method still functions beyond the 16 kHz case. Importantly, all of these codec-based variants preserve the main deployment advantage of CodecSep: they can operate directly on code streams, including a codes-in / codes-out path, with minimal loss relative to the continuous-latent path.

**What is the main takeaway?** The main takeaway is twofold. First, the **16 kHz DAC model is used as the primary CodecSep system** because it best matches the practical low-MAC setting that motivates this work. Second, the bandwidth-scaling results show that the *same masking interface remains effective across multiple codec backbones and sampling rates*, even though performance degrades as bandwidth increases. The comparison to AudioSep should therefore be read as follows: AudioSep is benchmarked in its standard and strongest available **32 kHz** form, while CodecSep is benchmarked in the codec-native settings supported by available neural codec backbones, with **16 kHz** used as the primary low-MAC operating point and higher-bandwidth experiments included specifically to contextualize the sampling-rate mismatch. Our claim is therefore not that 16 kHz and 32 kHz are perfectly matched operating points, but that (i) CodecSep is effective in the codec-mediated deployment regime, (ii) its masking interface remains valid as bandwidth increases, and (iii) higher-bandwidth operation becomes progressively harder. We view these results as an initial step toward full-band, and eventually stereo/spatial, operation within the same masking framework.

### 4.1.9   Additional experiments (cf. Appendix F–H).

For readability, several extended studies are deferred to the Appendix, which provides full details on data construction, prompt protocols (including *generic* vs. *universal* prompting and paraphrased variants), training/evaluation splits, and metric definitions. We summarize here the main conclusions of those studies.

**Does CodecSep remain competitive under a harder cross-domain stress test (cf. Appendix F)?** Yes, but the appendix uses **AudioCaps** for a more specific purpose than the other external benchmarks: it serves as a *stress test* for cross-domain transfer rather than as a representative success case. Broader benchmarking in the paper already shows that the **dnr-v2-trained** CodecSep model generalizes favorably across multiple external datasets. By contrast, **AudioCaps** exposes a more difficult transfer setting because it differs substantially from **dnr-v2** in both source composition and prompt distribution, while the official pretrained **AudioSep** also benefits from broader upstream exposure to AudioSet-like data. Under zero-shot transfer from **dnr-v2** to **AudioCaps**, **CodecSep+dnr-v2** remains competitive with **AudioSep+dnr-v2** in SI-SDR, but neither retrained model matches the official pretrained AudioSep. We therefore treat this direction primarily as an informative *failure-case / stress-test regime* that helps reveal the limits of transfer beyond the training distribution. At the same time, the appendix also shows that this is not a fundamental weakness of CodecSep on AudioCaps itself: under *matched AudioCaps training*, **CodecSep+AudioCaps** achieves higher SI-SDR than **AudioSep+AudioCaps**, and in the reverse direction the **AudioCaps-trained** CodecSep model transfers favorably to the denser **dnr-v2** mixtures, especially in SI-SDR. We therefore interpret Appendix F not as the main evidence for cross-benchmark success, but as a targeted analysis showing that CodecSep remains competitive even in a substantially harder transfer regime, while also clarifying an important boundary of its generalization behavior.

**How consistent are the gains over AudioSep across datasets and prompt settings (cf. Appendix G)?** The appendix also reports *relative gain summaries* of CodecSep over AudioSep under matched training data and prompt conditions. These summaries are intended to complement, rather than replace, the absolute tables in the main paper. The results show that CodecSep yields positive SI-SDR gains on **dnr-v2**, under **paraphrased prompts**, and across additional open-domain benchmarks including **ESC-50**, **Clotho-v2**, **AudioSet**, **VGGSound**, and **AudioCaps**. The trend is strongest on dnr-v2, remains visible under cross-benchmark transfer, and becomes smaller under prompt ambiguity. ViSQOL improvements are generally smaller than the SI-SDR gains and are more mixed across datasets, especially on AudioCaps, but the overall pattern remains favorable to CodecSep. Taken together, these summaries reinforce the main conclusion that CodecSep delivers *consistent signal-level separation gains* over AudioSep across a broad range of datasets and prompt settings, with perceptual improvements that are more modest but still competitive overall.

**What do reconstruction diagnostics reveal about leakage, consistency, and the role of masking (cf. Appendix H)?** The appendix includes a *single-source reconstruction* diagnostic on **dnr-v2**, in which each model—although trained for *source separation*—is given an isolated source and asked to reproduce it. For the text-guided models, the matching prompt is provided; for the fixed-stem models, the appropriate output head is used. In addition, we report *mixture reconstruction* by summing the predicted stems for a mixture and comparing the result to the original mixture. This experiment is intended as a diagnostic study of leakage and mixture consistency, rather than as a primary separation benchmark.

Three conclusions are most relevant. First, within codec-latent separation models, **explicit masking** is substantially more effective than **decoder-style latent generation**: **CodecSep** reconstructs much more reliably than **CodecFormer**, indicating that reweighting information already present in the codec latent space is more stable than attempting to regenerate source latents through a decoder. Second, this does not imply that codec-latent masking is the strongest reconstruction strategy overall: **TDANet** and especially **SDCodec** remain strong reconstruction baselines, with SDCodec being architecturally advantaged by its source-specific codebooks. Third, the **masker ablation** clarifies the role of the full CodecSep architecture. In the ablated variant, FiLM is applied directly within the NAC encoder and the decoder reconstructs from the conditioned encoder representation. This yields very strong source-consistent reconstruction when the input already matches the prompt, but it performs poorly on actual source separation, showing that **direct FiLM-based affine conditioning alone is not sufficient for source extraction from mixtures**. A likely reason is that affine modulation at the intermediate NAC encoder layers tends to **collapse the latent space toward a prompt-biased representation**, rather than preserving the separable structure needed for disentanglement and source selection. Instead, an **explicit masker** is needed to perform source selection and isolate the target source from competing mixture content. We therefore interpret these diagnostics as supporting the view that CodecSep's main strength lies in *source-selective separation through explicit masking in codec latent space*, rather than in exact waveform-faithful reconstruction.

### 4.1.10 Extension to multi-modal prompting.

Because conditioning enters only via a fixed-dimensional query embedding $e_\tau$ that drives FiLM in the masker, the architecture is agnostic to the prompt modality. Concretely, one can replace the text encoder with (i) an *audio* encoder to accept audio prompts $\left(e_\tau^{\text{aud}}\right)$, (ii) an *image/vision–language* encoder (e.g., CLIP) to accept image prompts $\left(e_\tau^{\text{vis}}\right)$, or (iii) a lightweight fusion (e.g., gated additive or attention pooling) of $\left(e_\tau^{\text{text}}, e_\tau^{\text{aud}}, e_\tau^{\text{vis}}\right)$ to support mixed prompts—all without modifying the masker or the codec.

## 5 Conclusion

We presented **CodecSep**, a text-guided universal sound separation framework that operates directly in neural audio codec latent space using a FiLM-conditioned Transformer *masker*. Unlike spectrogram-domain text-guided systems such as AudioSep, CodecSep performs *source selection* over compact codec representations rather than separation over waveform or STFT features, enabling a substantially lighter separation pipeline in codec-mediated settings.

Across **dnr-v2** and five additional open-domain benchmarks, CodecSep consistently delivers stronger **SI-SDR** than AudioSep under matched training and prompt protocols, while remaining competitive in **ViSQOL** and achieving clear gains in human **MOS–LQS**. The model also remains effective under prompt paraphrasing, benefits from finer-grained semantic supervision, and extends naturally to a deployment-ready *codes in: codes out* pathway that remains competitive even without additional fine-tuning. Architectural studies further show that, in codec latent space, **explicit masking is more effective than decoder-style generation** for source separation.

The qualitative and diagnostic analyses provide additional support for the central design premise. The codec latent space appears to preserve meaningful source-dependent organization, and the learned masks exploit this structure primarily through channel-wise modulation. The oracle and reconstruction studies further clarify the operating regime of CodecSep. They show that the residual errors in the separated outputs are not explained by the codec bottleneck alone, but also reflect the intrinsic difficulty of mixture-conditioned source recovery and additional separation-specific error. At the same time, the auxiliary reconstruction diagnostics should be interpreted separately from the main separation task: CodecSep is designed for target-source extraction from mixtures, not for maximizing reconstruction scores in the isolated-source diagnostic setting. The masker ablation further shows that direct FiLM-based affine conditioning inside the NAC encoder can support source-consistent reconstruction, but an explicit masker over NAC latents is required to extract target sources from mixtures.

From a systems perspective, the principal advantage of CodecSep appears in the **code-stream deployment regime**, where the edge device already transmits audio to the server as *neural audio codec (NAC) codes* rather than as raw waveform samples. In such a pipeline, conventional separators such as AudioSep cannot operate directly on the transmitted representation: the server must first *decode* the codec stream back to audio, perform separation in the waveform or spectrogram domain, and then *re-encode* the separated outputs if codec-compatible transmission or storage is required.

CodecSep avoids this additional decode–separate–re-encode cycle. Instead, on the server side, the transmitted codec codes are mapped to codec embeddings through *codebook lookup*, after which the separator operates *directly* in codec space to estimate source-specific latent representations. These latent estimates can then either be decoded to audio or re-quantized back into codec codes, yielding a practical *codes in: codes out* pathway for codec-mediated edge–server deployment.

In this setting, CodecSep requires only **1.35 GMACs** end-to-end, corresponding to roughly $54\times$ **lower compute** than AudioSep in the same codec-mediated pipeline ($25\times$ under architecture-only comparison), while also reducing latency and memory footprint substantially. More broadly, this design provides a concrete **deployment blueprint** for codec-mediated separation and related downstream audio processing: whenever a pipeline already contains codec-domain representations—whether transmitted codes or embeddings reconstructed from them—downstream modules can operate directly on those representations to estimate source-specific latent structure, rather than repeatedly decoding to waveform and re-encoding after each stage. Taken together, these results position CodecSep as a practically attractive framework for **low-latency, codec-native source separation**.

Overall, the results show that modern neural audio codec latents are sufficiently structured to support effective prompt-guided source extraction through masking alone. This suggests that codec-native separation is not only computationally attractive, but also a viable modeling direction for universal sound separation. Code is publicly available at: `https://github.com/adhiraj69/CodecSep`.

## 6 Limitations and Clarifications.

We discuss the limitations of our work as follows–

(1) *Data and prompts.* Training data scale and prompt diversity are modest relative to open-domain audio. As shown in Table 5, finer SFX supervision sharpens SFX extraction *and* improves speech/music stems; larger, more heterogeneous corpora spanning multiple prompt granularities—including temporal/relational cues—should yield further gains.

(2) *Temporal prompting.* While CodecSep is robust to synonymic paraphrases, we did not evaluate prompts with explicit temporal structure (e.g., causal ordering), which remains an open direction.

(3) *Perceptual SFX quality.* In some settings, SFX perceptual quality trails the best competing scores despite superior SI-SDR; improving SFX naturalness without sacrificing separation is future work.

(4) *Channel-wise masking and residual overlap.* Our qualitative analysis suggests that the learned masks are predominantly channel-wise rather than strongly time-localized. While this is sufficient for effective separation in many cases, it also means that sources sharing partially overlapping latent channels may not be perfectly disentangled. As a result, some non-target sources can remain weakly active in the separated outputs, especially in dense mixtures with overlapping events. Addressing such residual overlap may require richer masking mechanisms that incorporate stronger temporal selectivity or more explicit source disentanglement objectives.

## 7 Future Work

Several directions could further strengthen codec-native universal sound separation.

**Richer data and prompt supervision.** A natural next step is to scale training to larger and more heterogeneous audio corpora with broader prompt diversity. In particular, training with mixtures of prompt granularities—from coarse category prompts to fine-grained compositional descriptions—may improve both generalization and controllability. Extending the prompt space to include temporal, relational, and referring-expression cues is also an important direction.

**Beyond channel-wise masking.** Our qualitative analysis suggests that CodecSep primarily relies on channel-wise latent reweighting. While effective, this also leaves residual overlap in dense mixtures where multiple sources share partially overlapping latent channels. Future work could therefore explore richer masking mechanisms that combine channel-wise modulation with stronger temporal selectivity, explicit cross-source competition, or additional disentanglement-oriented objectives.

**Improved perceptual quality.** Although CodecSep achieves strong SI-SDR and competitive ViSQOL, perceptual quality for some SFX conditions still trails the best competing systems. A promising direction is to incorporate auxiliary perceptual objectives, embedding-consistency losses, or lightweight refinement stages that improve naturalness without sacrificing the efficiency advantages of codec-domain separation.

**Higher-bandwidth and spatial audio.** Our bandwidth-scaling study suggests that the masking interface remains valid at higher sampling rates, but separation becomes more difficult as high-frequency and transient detail become more entangled. Future work should therefore investigate larger-capacity maskers, improved codec backbones, and training at 24 kHz, 44.1 kHz, and 48 kHz more systematically. Extending the framework to higher-bandwidth as well as stereo and spatial audio is another important direction, for example with 48 kHz stereo EnCodec, HO–DirAC Hold et al. (2024), or SpatialCodec Xu et al. (2024).

**Multimodal prompting.** Because conditioning enters only through a fixed-dimensional query embedding, CodecSep naturally admits extensions beyond text. Future work could study audio-guided, image-guided, or mixed multimodal prompting, allowing users to specify targets through reference sounds, images, or combined cues without changing the core masking architecture.

**Deployment and on-device validation.** While our current evidence supports the usefulness of CodecSep for codec-mediated edge/server pipelines through compute, memory, and latency analysis, direct deployment on mobile or embedded hardware remains to be demonstrated. An important future direction is therefore end-to-end validation on real on-device platforms, including power, latency, and memory measurements under realistic streaming workloads.

## 8 Broader Impact

CodecSep is motivated by a practical systems question: can universal, prompt-guided sound separation be made lightweight enough to operate in codec-mediated edge–server pipelines, rather than only in compute-heavy waveform or spectrogram domains? In that sense, the main positive impact of this work is not only improved separation quality under matched evaluation, but also a shift toward more deployment-realistic audio models. By operating directly on neural audio codec latents, CodecSep reduces compute, memory, and latency substantially in code-stream settings, which may broaden access to source-separation technology in resource-constrained environments. This could benefit assistive listening, hearing augmentation, speech enhancement in communication systems, low-bandwidth audio editing, interactive media production, and on-device or edge-based content manipulation where repeated decode–separate–re-encode cycles are undesirable.

A second potential positive impact is flexibility. Unlike fixed-stem separation systems that are restricted to a predefined output taxonomy, CodecSep supports prompt-guided extraction and therefore moves toward more open-vocabulary interaction with audio mixtures. In principle, this may make audio tools easier to control for non-expert users, since extraction can be specified semantically rather than through fixed source heads. The architectural result may also be of broader research interest: the paper suggests that modern neural audio codec latents already contain enough source-structured information for masking-based extraction, which may encourage more efficient codec-native approaches for related audio understanding and generation tasks.

More broadly, the deployment path explored here may be useful beyond source separation itself. By showing that downstream processing can be performed directly on codec representations, rather than repeatedly decoding to waveform, processing, and re-encoding, CodecSep also illustrates a more general *codec-native* systems pattern for audio inference. This perspective may be relevant to other deployment-oriented tasks such as target speaker extraction, speech enhancement, denoising, dereverberation, or prompt-guided audio editing, where one may wish to conditionally refine or modulate an already compressed representation under tight latency, memory, or bandwidth constraints. In that sense, the positive systems impact of this work is not only the efficiency of one separator, but also the suggestion that neural audio codecs can serve as a shared representational backbone for a broader family of practical audio processing pipelines.

At the same time, such flexibility introduces risks. Prompt-guided extraction can be misused for privacy-invasive or surveillance-oriented applications, such as isolating speech, speakers, or background events from recorded mixtures without the knowledge or consent of those being recorded. More generally, any method that lowers the computational barrier for source separation can also lower the barrier for extracting sensitive content from audio. We do not claim that CodecSep solves speaker identification, diarization, or forensic enhancement, but the ability to isolate semantically specified content could still be misapplied in ways that raise privacy concerns.

There are also risks of misuse in media manipulation. Improved sound separation can facilitate unauthorized remixing, decontextualization, or repurposing of copyrighted or personal audio content. In creative settings this may be beneficial for editing and accessibility, but in adversarial settings it could enable misleading edits, selective removal of contextual sounds, or extraction of material that creators did not intend to be isolated. Because CodecSep is text-conditioned, an additional concern is that users may over-trust the semantic controllability of the system and assume that a prompt uniquely identifies a source, even when prompts are ambiguous or multiple events overlap.

Several properties of the present work limit these risks but do not eliminate them. First, the model is evaluated primarily in mono, prompt-guided source extraction rather than speaker-level forensic recovery. Second, the paper explicitly shows that performance degrades under paraphrased or ambiguous prompts and at higher bandwidths, and that perceptual quality for some SFX settings remains imperfect. Third, the strongest efficiency gains arise in codec-mediated deployments, so the system is most naturally suited to applications where codec infrastructure already exists. These limitations mean that CodecSep should not be interpreted as a turnkey solution for arbitrary audio surveillance or perfect open-world extraction. Nevertheless, even imperfect systems can be misused, especially if integrated into larger pipelines.

We therefore view responsible deployment as important. In practical applications, safeguards could include clear user-facing disclosure that separated outputs are model-generated estimates rather than ground-truth stems, access controls for sensitive domains, provenance logging for edited audio, and restrictions on use in privacy-sensitive or consent-critical settings. Benchmarking should also expand beyond fidelity to include misuse-relevant evaluations such as prompt ambiguity, failure under overlapping sources, and robustness against extracting unintended content.

Finally, the environmental impact of this work is mixed. Training modern audio models still consumes non-trivial compute, and our experiments use GPU resources. However, one motivation of CodecSep is to reduce inference-time cost in realistic deployment settings. If such codec-native systems replace heavier decode–separate–re-encode pipelines at scale, they may reduce operational energy usage during inference, especially in repeated or latency-sensitive edge–server workflows. We therefore believe the broader impact of this work is potentially positive, provided that efficiency gains are paired with clear communication of limitations and with responsible use in privacy- and consent-sensitive contexts.

## 9 Declaration of LLM Usage.

LLM is used only to aid or polish writing and does not impact the core methodology, scientific rigorousness, or originality of the research.

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

# A  Failure Modes of PIT and MixIT for Universal Sound Separation

Permutation-Invariant Training (PIT) and Mixture-Invariant Training (MixIT) have historically been effective for closed-domain separation tasks where the number of underlying sources is known, fixed, or varies within a narrow, well-defined range. However, their underlying assumptions lead to structural limitations when applied to open-domain universal source separation (USS), where mixtures may contain an arbitrary and potentially large number of heterogeneous sound events. In this section, we summarize the key failure modes observed when training with PIT/MixIT to extend fixed-stem models to open-domain mixtures.

A core limitation of PIT/MixIT is the requirement to specify a maximum number of output sources, denoted by $N$. During training, the model produces exactly $N$ outputs for every mixture, and the PIT or MixIT objective establishes a correspondence between these outputs and the underlying reference sources (or intermediate MixIT partitions). This design is brittle in scenarios where the true number of sources varies widely. When the mixture contains more than $N$ sources—a frequent occurrence in open-domain audio—the model has no mechanism to create additional outputs. Instead, it suffers source collision and collapses multiple sources into a single output stem, resulting in unavoidable leakage, loss of fine structure, and a sharp degradation in separation quality. The post-hoc identification step cannot recover the missing sources, because the model never produced separate representations for them in the first place; those sources simply do not exist within the model's output space.

Conversely, when the mixture contains fewer than $N$ sources, the model is still obligated to return $N$ outputs. This mismatch introduces new problems: several outputs correspond to no actual source and become "inactive" stems, while others may capture residual background energy or hallucinated content. These false positives degrade metrics such as SI-SDR and create ambiguity during evaluation because the model does not encode which stems are meant to be meaningful. Such outputs also make deployment difficult, as downstream systems must decide which stems to trust and which to ignore.

The reliance on a fixed maximum number of sources $N$ also places a heavy burden on both training stability and computational cost. As $N$ increases, the permutation space in PIT expands combinatorially, and MixIT assignments become increasingly complex, making training slow, unstable, and in many cases prone to divergence. In open-domain datasets such as dnr-v2, mixtures may contain eight or more concurrent sound events, forcing PIT/MixIT baselines to adopt impractically large values of $N$ to avoid source collisions. In practice, such configurations are computationally prohibitive and empirically unreliable.

These limitations collectively illustrate why PIT and MixIT, despite their historical success in speech separation and other closed-set tasks, are poorly suited for open-domain universal separation. Their fixed-output architecture is fundamentally mismatched to real-world mixtures that contain highly variable and unpredictable numbers of sources. In contrast, CodecSep bypasses this bottleneck entirely through free-form text-guided inference: the model extracts only the requested source category, emits no unused stems, and scales naturally to mixtures with arbitrary levels of overlap. This flexibility enables CodecSep to support both closed-set and open-domain use cases, while also providing a foundation for future extensions to fine-grained extraction of individual speaker stems, instrument stems, or sound-effect stems.

# B  Extended Design Rationale: FiLM-Conditioned Masking in NAC Latent Space

This appendix expands the rationale behind the design choices summarized in Section 3. We follow the same presentation order requested by the reviewer and adopted in the main paper: (i) task formulation and pipeline contrast, (ii) model architecture and the rationale for masking in codec latent space, (iii) training and representation considerations, and (iv) deployment discussion. The goal of this appendix is not to redefine the method, but to make explicit the technical motivations underlying the main architectural choices: why we operate on neural audio codec (NAC) latents rather than spectrograms, why we use a FiLM-conditioned masker rather than a decoder-style generator, why conditioning is injected inside the masker, why the main experiments are reported on continuous latents $Z$, and why the same formulation extends naturally to code-stream deployment. We also connect these design choices to the qualitative analysis in Section 4.1.4 and the oracle codec analysis in Section 4.1.5.

## B.1  Task formulation and pipeline contrast

Let $x(t) \in \mathbb{R}$ denote a mono mixture waveform composed of sources $\{y_s(t) \mid s \in \mathcal{S}\}$:

$$x(t) = \sum_{s \in \mathcal{S}} y_s(t). \tag{14}$$

Given a natural-language query $\tau$ and its text embedding $e_\tau$, the goal is to recover the waveform of the source consistent with the query.

In spectrogram-domain text-guided separation systems such as AudioSep, separation is performed after transforming the waveform to a complex time–frequency representation:

$$x(t) \xrightarrow{\text{STFT}} X \in \mathbb{C}^{F \times T_{\text{spec}}} \xrightarrow{Spec(X, e_\tau)} \tilde{Y}_s = |\hat{M}_s| \odot |X| \exp\big(\angle X + \angle \hat{M}_s\big) \xrightarrow{\text{ISTFT}} \tilde{y}_s(t), \tag{15}$$

where $Spec(\cdot, e_\tau)$ denotes a FiLM-conditioned spectrogram separator that predicts a magnitude mask $|\hat{M}_s| \in [0,1]^{F \times T_{\text{spec}}}$ and a phase residual $\angle \hat{M}_s$, conditioned jointly on the mixture spectrogram $X$ and the text embedding $e_\tau$.

In CodecSep, the task is instead posed directly in the codec latent domain:

$$x(t) \xrightarrow[\text{DAC}]{Enc(\cdot)} Z \in \mathbb{R}^{d \times T} \xrightarrow{Mask(Z, e_\tau)} \tilde{Z}_s = M_s \odot Z \xrightarrow[\text{DAC}]{Dec(\cdot)} \tilde{y}_s(t), \tag{16}$$

where $Enc(\cdot)$ and $Dec(\cdot)$ are the frozen DAC encoder and decoder, and $Mask(\cdot, e_\tau)$ is a FiLM-conditioned transformer masker that predicts an element-wise soft mask $M_s \in [0,1]^{d \times T}$ over codec latents $Z = Enc(x)$.

Thus, for an input clip producing $T$ latent frames, the separator receives $Z \in \mathbb{R}^{d \times T}$, predicts a same-shape mask $M_s \in [0,1]^{d \times T}$, forms the masked latent $\tilde{Z}_s = M_s \odot Z$, and decodes it to the waveform estimate $\tilde{y}_s(t)$. In the main system, the separator therefore acts as a *selection* mechanism over codec latents rather than a source generator. This distinction is central to the rest of the design rationale.

**Dimensionality and compression advantage.**  A first advantage of the codec-domain formulation is representational compactness. For 1 s audio at 32 kHz, a complex STFT with window size $N{=}1024$ and hop size $M{=}320$ produces approximately $T_{\text{spec}} \approx 100$ frames and $F = 2 \times 1024$ real-valued scalars per frame (real and imaginary parts), yielding

$$F \cdot T_{\text{spec}} \approx 204{,}800 \tag{17}$$

scalars per second. By contrast, a 16 kHz DAC with latent width $d{=}64$ and the same hop size yields approximately $T \approx 50$ latent frames and therefore

$$d \cdot T = 64 \times 50 = 3{,}200 \tag{18}$$

scalars per second, i.e., roughly $64\times$ fewer than the complex STFT representation. Even for higher-rate codecs such as 32 kHz EnCodec-like settings with $d{=}128$, the separator still operates on a representation roughly

$32\times$ smaller than the complex STFT. This reduction directly lowers attention and MLP costs, activation memory, and memory bandwidth inside the separator.

The dimensionality reduction, however, is only part of the motivation. The stronger architectural claim in this paper is that NAC latents are not only smaller, but also more structured for masking-based source extraction than spectrograms. The remainder of this appendix makes that claim precise.

### B.2 Model architecture and design rationale

### B.2.1 Why NAC latents rather than spectrograms?

The spectrogram and codec-latent formulations differ not only in size but also in what structure is already present before separation begins.

**STFT-domain separation must learn both abstraction and separation.** The STFT is a fixed linear transform from waveform space to a complex time–frequency representation. While highly effective as a signal representation, it does not itself impose a task-specific organization aligned to text-guided source extraction. As a result, spectrogram-based systems such as AudioSep typically require a substantial learned encoder–decoder stack (e.g., CNN/ResUNet-style modules) to first compress and reorganize the spectrogram into internal features and then predict source-selective masks. In that regime, the separator must simultaneously learn:

1. how to build higher-level latent features from a high-dimensional input,

2. how to align those features with text conditioning, and

3. how to separate the target source.

This couples representation learning and source selection inside the separation network itself, increasing parameter count, MACs, and optimization burden.

**NAC latents provide a compact codec-induced prior.** By contrast, DAC already maps the waveform to a compact latent representation

$$Enc(\cdot) : x \mapsto Z \in \mathbb{R}^{d \times T}, \tag{19}$$

where the latent space has been shaped by codec pretraining to preserve the factors most relevant to perceptually faithful audio reconstruction under compression constraints. In our setting, the separator is therefore not asked to discover an entirely new internal representation from raw waveform or spectrogram input. Instead, it operates on a representation that has already been compressed, denoised, and organized by the codec.

This is the sense in which the codec acts as a prior for separation. The claim is not that codec pretraining was explicitly designed for source disentanglement. Rather, the claim is that the codec training objectives induce a latent organization in which source-relevant acoustic attributes are encoded compactly enough that source extraction can be formulated as masked selection on $Z$, rather than requiring full source generation.

**What the codec pretraining contributes.** The structure of $Z$ arises from several components of codec training working together:

- **Reconstruction pressure:** the encoder must preserve the information required for accurate waveform recovery after compression.

- **Multi-scale spectral criteria:** these encourage preservation of perceptually important spectral content across resolutions.

- **Time-domain fidelity terms:** these stabilize reconstruction and preserve waveform detail.

- **Adversarial and feature-matching losses:** these push the representation to support realistic reconstruction of periodicity, timbre, fine spectral detail, and transient structure.

- **Residual vector quantization (RVQ):** this forces the representation into a bitrate-constrained, hierarchical encoding.

- **Quantizer dropout / bitrate robustness mechanisms:** these discourage pathological dependence on only late RVQ stages and promote smoother residual allocation across quantizers.

Taken together, these mechanisms do not prove formal source disentanglement, but they do encourage a representation in which salient acoustic factors are compactly organized and recoverable under strong compression. For a masking-based separator, this is exactly the kind of structure that makes latent selection more plausible.

**Why this matters for separation.** Under the codec-domain formulation, the separator learns

$$Mask(\cdot, \cdot) : (Z, e_\tau) \mapsto M_s, \qquad \tilde{Z}_s = M_s \odot Z. \tag{20}$$

The learning problem is therefore to infer *where* in an already-formed latent representation the query-consistent information resides, and *how much* of each latent component should be preserved or suppressed. In spectrogram-domain separation, by contrast, the model must first build its own discriminative internal representation from a much larger and noisier input and then perform selection in that learned space. The codec prior therefore changes the optimization problem from joint abstraction-plus-separation to predominantly query-conditioned latent selection.

### B.2.2 RVQ structure and why it is relevant to masking

A central part of this prior comes from residual vector quantization. Given encoder latents $Z$, the codec produces discrete codes

$$A = [a_t \in [K]^{N_q}]_{t=1}^T, \tag{21}$$

where $K$ is the codebook size and $N_q$ is the number of quantizers. Codebook lookup then yields reconstructed embeddings

$$e_t = \sum_{i=1}^{N_q} \text{lookup}(a_t^{(i)}), \qquad E = [e_t]_{t=1}^T \approx Z. \tag{22}$$

RVQ imposes a coarse-to-fine decomposition of representation capacity. Earlier quantizers must explain the dominant structure that can be captured at low bitrate, while later quantizers refine the residual error left by earlier stages. In audio codecs, this often aligns naturally with a progression from coarse acoustic organization toward finer details: broad spectral envelope, speaker or instrument identity cues, and sustained structure tend to be represented earlier, whereas fine transients, residual texture, and high-frequency detail are refined later. We do not claim that these boundaries are perfectly clean or universally interpretable per quantizer. The more relevant point for separation is that RVQ discourages a flat, unstructured allocation of information and instead induces a layered representation with progressively refined residual content.

For a mask-based model, such hierarchy is useful because the separator does not need to invent a target representation from scratch. It can instead exploit the codec's latent organization by selectively preserving latent components that already encode query-relevant structure and attenuating others. This is precisely the intuition behind choosing masking over generation in codec space.

### B.2.3 Connection to the qualitative latent analysis

The qualitative analysis in Section 4.1.4 provides direct support for this architectural interpretation.

First, the reference source latents in Fig. 3 exhibit visibly distinct activation patterns across speech, music, and SFX. This matters because it suggests that the frozen codec encoder does not collapse different sources into an undifferentiated latent basis. Instead, at least for the representative examples shown, different source categories already induce different channel–time activation structure in the codec latent plane.

Second, the estimated masks in Fig. 3 are predominantly channel-wise: they form largely horizontal bands with relatively modest temporal variation. This is consistent with the interpretation that CodecSep is not performing sharply localized, frame-by-frame temporal gating in the manner of a spectrogram mask, but rather source-conditioned reweighting of a structured latent basis. Put differently, the model appears to separate primarily by selecting and attenuating latent channels that already encode different source-relevant factors.

Third, the residual visualization

$$\left| Z - \frac{1}{3} \left( Z_{\text{speech}} + Z_{\text{music}} + Z_{\text{SFX}} \right) \right| \tag{23}$$

remains sparse over broad latent regions. We do not interpret this as evidence that mixture latents are literal linear superpositions of source latents, nor as evidence of exact disentanglement. The mixture construction process in dnr-v2 involves nontrivial mixing and normalization, and DAC itself is not trained with an explicit source-disentanglement objective. The point of the residual analysis is more specific: it shows that the mixture latent retains a visible relationship to the latent organization of its constituent sources rather than appearing completely unstructured. That observation is compatible with the design premise that source extraction can be framed as modulation and selection within codec latent space.

These qualitative observations therefore do not stand alone as a general proof, but they do provide mechanistic evidence that complements the empirical results: the codec latent space appears to preserve source-dependent organization, and the learned masks exploit that organization primarily through channel-wise modulation.

### B.2.4 Why masking rather than decoder-style generation?

A second major design choice is to predict a mask over codec latents rather than directly generate target latents or waveforms. Concretely, CodecSep estimates

$$M_s = Mask(Z, e_\tau) \in [0,1]^{d \times T}, \qquad \tilde{Z}_s = M_s \odot Z, \qquad \tilde{y}_s(t) = Dec(\tilde{Z}_s). \tag{24}$$

This differs from decoder-style source generation approaches, such as predicting $\tilde{Z}_s$ directly via a conditional generator.

The preference for masking is motivated by several considerations.

**(1) Simpler optimization target.** Learning a mask is a more constrained problem than learning to synthesize the full target representation. The model need not invent missing structure or generate target waveforms from scratch; it only needs to identify which latent components are relevant to the query and to what extent they should be retained. In a compact codec latent space, this is substantially lighter than full conditional generation.

**(2) Better use of the codec prior.** If the codec has already organized audio into a structured latent representation, then a masking model is well matched to that structure: it treats the latent as the object to be *selected from*. A generator-style model, by contrast, uses the codec latent mainly as an input from which a new target latent must be synthesized. That partially forfeits the advantage of operating on a codec-organized representation in the first place.

**(3) Reduced hallucination and lower distortion pressure.** Because the mask-based model only modulates existing latent content, it is less prone to producing content that was not already supported by the mixture representation. This does not eliminate leakage or distortion, but it does constrain the failure modes. In practice, this is one reason why masking is an attractive choice when the goal is source selection rather than conditional audio synthesis.

**(4) Preservation of long-range structure.** The codec encoder has already organized long-horizon attributes such as periodicity, timbre, and transient patterns in $Z$. Masking preserves that organization and propagates it through the frozen decoder. A generator that reconstructs the target from scratch has more freedom, but also more opportunity to drift from the source-consistent latent structure already present in the mixture.

**(5) Efficient use of transformer capacity.** With masking, the transformer's capacity is spent primarily on inferring *where and how strongly* to gate latent information. It is not responsible for synthesizing full source structure. This focus is particularly appropriate in the low-GMAC regime targeted by CodecSep.

The empirical comparison to generation-style alternatives in the paper is therefore not incidental. It directly tests whether the codec-domain prior is more effectively exploited by latent selection than by latent generation. The qualitative latent analysis discussed above provides a mechanistic explanation for why the masking approach is effective: if source-relevant organization is already present in the latent space, then a source-conditioned mask is a natural and efficient interface to that structure.

### B.2.5 FiLM-conditioned transformer masker: architectural details and rationale

The main paper specifies the architecture; here we make explicit why the conditioning and dimensional interface are designed as they are.

**Latent-to-transformer interface.** The frozen DAC encoder outputs codec latents $Z \in \mathbb{R}^{d \times T}$, where $d$ is the codec channel width and $T$ is the latent sequence length. Since the transformer width $d_t$ differs from the codec width, we first project latents channel-wise into transformer space:

$$Z' = \text{Conv}(Z), \qquad Z' \in \mathbb{R}^{d_t \times T}. \tag{25}$$

All transformer processing then occurs at width $d_t$. After the final transformer layer, a convolutional mask head maps the features back to codec dimensionality and predicts the source-conditioned mask

$$M_s \in [0,1]^{d \times T}. \tag{26}$$

This design keeps temporal resolution fixed and uses only channel-wise projections to bridge codec and transformer spaces. It therefore preserves the one-to-one frame alignment needed for element-wise masking in the original codec latent domain.

**Where FiLM is applied and why.** Given a CLAP text embedding $e_\tau \in \mathbb{R}^{d_t}$, a lightweight query network $query(\cdot)$ produces per-layer FiLM parameters for the intermediate transformer blocks. In the implemented system, the query network is a single linear layer and FiLM is injected into layers $l = 2, \ldots, L-1$, leaving the first and last layers unmodulated. For hidden activations $H^l \in \mathbb{R}^{d_t \times T}$, FiLM takes the form

$$\tilde{H}^l = \text{FiLM}(H^l; \gamma^l, \beta^l) = \gamma^l \odot H^l + \beta^l. \tag{27}$$

Placing FiLM *inside the masker*, rather than in the codec encoder or decoder, confines text conditioning to the selection stage. This is important: the objective is not to change the codec representation itself, but to modulate how the separator reads and gates that representation for the current query.

**Why Post-LN FiLM instead of AdaLN.** We use a post-LN FiLM design, meaning the text-conditioned affine modulation is applied to transformer activations after the sublayer computation rather than by conditioning the normalization transform itself. This choice is tied to the nature of the task. CodecSep performs masking-based source selection rather than generative synthesis. In that setting, the conditioning mechanism should act as directly as possible on the features that determine whether a latent component is passed through or suppressed.

Post-LN FiLM does exactly that: it behaves as an explicit channel-wise feature gate. For a given query, it can increase the salience of channels carrying target-consistent structure and attenuate channels associated with interfering sources, without perturbing the transformer's normalization statistics. AdaLN is highly

effective in generative settings where the goal is to steer broad representation formation. Here, however, the desired behavior is sharper and more selective. The channel-wise mask behavior observed in Section 4.1.4 is consistent with this design choice: separation appears to be achieved primarily through source-conditioned reweighting of latent features, which is precisely the kind of mechanism that post-LN FiLM makes explicit.

**FiLM parameterization and stability.** Following AudioSep, we use a simplified FiLM parameterization in which the scale is fixed and only the bias is learned:

$$\gamma^l = \mathbf{1}, \qquad \tilde{H}^l = H^l + \beta^l. \tag{28}$$

The FiLM projections are initialized with Xavier uniform weights and zero bias so that $\beta^l = \mathbf{0}$ at initialization. Consequently, the transformer initially behaves exactly like an unconditioned masker. This initialization is useful because it avoids abrupt conditioning-induced perturbations at the start of training. More generally, fixing $\gamma^l$ avoids uncontrolled multiplicative scaling of hidden activations, which can be undesirable in post-LN architectures. In the context of masking-based separation, additive modulation is sufficient to bias the representation toward the query without destabilizing the latent geometry on which the mask is learned.

### B.3 Training objective and representation-level considerations

The training objective in the main paper is

$$\mathcal{L} = -\sum_s \text{SI-SDR}(y_s, \tilde{y}_s) - \text{SI-SDR}(x, \tilde{x}), \tag{29}$$

where $\tilde{x}$ is obtained by decoding the summed latent estimates. DAC and the CLAP text encoder are frozen; only the FiLM-conditioned masker and the query network are updated.

Beyond the objective itself, three representation-level choices are important for understanding the training setup.

#### B.3.1 Why the main training and analysis operate on continuous latents $Z$

We report the main results using continuous encoder latents $Z = Enc(x) \in \mathbb{R}^{d \times T}$ rather than discrete code indices. This choice is motivated by both optimization and representation quality.

**(1) Clean gradient flow.** Because the codec is frozen, operating on $Z$ allows gradients to flow directly through the masker and decoder without straight-through estimators or other discrete optimization machinery. This makes training substantially simpler and more stable.

**(2) Richer signal for text-conditioned masking.** The continuous latent $Z$ retains the representational organization induced by codec pretraining without the quantization coarsening introduced by hard code assignments. For a FiLM-conditioned masking model that relies on fine channel-wise modulation, this continuous representation offers a smoother signal on which to learn query-conditioned selection.

**(3) Reduced variance from codebook dynamics.** Training directly on discrete or re-embedded code representations introduces additional variability related to codebook utilization, residual quantization allocation, and bitrate truncation effects. By training on $Z$, we isolate the separator behavior without conflating it with those discrete dynamics.

**(4) Better alignment with the qualitative analysis.** The qualitative figures in Section 4.1.4 are most interpretable in the continuous latent domain, where channel–time structure can be inspected directly. Since one of the paper's central mechanistic claims is that codec latents already exhibit source-relevant organization that can be exploited by masking, it is natural that the main analysis is presented in the same latent space in which that organization is clearest.

### B.3.2   Why the codec prior supports stable masking-based training

The codec-domain formulation is also favorable from an optimization perspective. Since the separator acts on a compact and perceptually organized latent space, the mask prediction problem is lower-dimensional and more constrained than spectrogram-domain masking or source generation. In practice, this typically leads to:

- lower activation and gradient noise due to smaller internal representations,

- faster convergence because the model does not need to learn a new front-end representation from raw spectrogram input,

- greater stability because the target operation is modulation of an existing structured signal rather than synthesis of a new signal.

This interpretation is consistent with the architectural ablation in the paper showing that masking outperforms latent generation in the same codec-domain setting. It is also consistent with the observed qualitative behavior of the masks: the model appears to exploit latent channels that already carry structured source-relevant information, rather than learning strongly time-localized or synthesis-heavy transformations.

### B.3.3   What the oracle codec analysis clarifies

The oracle codec comparison in Section 4.1.5 helps separate two questions that would otherwise be conflated:

1. How much distortion is already imposed by the frozen codec bottleneck?

2. How much additional error is introduced by the separator?

The oracle results show that the codec itself is not lossless: even perfect codec reconstruction of clean sources yields nonzero distortion. This is important for interpreting the design rationale because it clarifies that the separator is operating on a compressed representation with an intrinsic reconstruction ceiling. The masking formulation should therefore not be judged against an idealized lossless latent space, but against a codec-limited representation in which some distortion is already present.

At the same time, the gap between CodecSep and the oracle for the mixture, music, and SFX stems indicates that separation error remains a meaningful contributor beyond codec distortion alone. This is exactly where the masking design matters. The codec prior gives the separator a structured latent substrate, but it does not eliminate the intrinsic challenge of recovering sources from a loudness-controlled mixture. The oracle experiment therefore sharpens the interpretation of the design claim: CodecSep benefits from codec-induced latent structure, but its residual errors still reflect the difficulty of mixture-conditioned source extraction and the limitations of imperfect masking.

The speech case is especially informative. CodecSep exceeds the codec oracle in SI-SDR while remaining below it in ViSQOL. This highlights that effective source selection can improve signal-level separation metrics even when perceptual fidelity remains bounded by codec and masking artifacts. In other words, the codec latent space is sufficiently useful for selective recovery, but the perceptual ceiling remains constrained by both codec reconstruction and source-selection errors.

### B.4   Deployment discussion and extension to code streams

### B.4.1   From continuous latents to codec bitstreams

Although the main experiments are reported on continuous latents $Z$, the same masking formulation extends naturally to codec-stream deployment.

Given quantized codec codes

$$A = [a_t \in [1024]^{N_q} \mid t \in [T]], \tag{30}$$

we reconstruct embeddings via codebook lookup:

$$e_t = \sum_{i=1}^{N_q} \text{lookup}\big(a_t^{(i)}\big), \qquad E = [e_t]_{t=1}^T \approx Z. \tag{31}$$

The same separator can then operate on $E$:

$$\tilde{E}_s = M_s \odot E, \qquad \tilde{y}_s(t) = Dec(\tilde{E}_s). \tag{32}$$

If a codes-out interface is required, the masked embeddings can be re-quantized:

$$\hat{A}_s = Quant(\tilde{E}_s), \qquad \hat{E}_s = \text{lookup}(\hat{A}_s), \qquad \tilde{y}_s(t) = Dec(\hat{E}_s). \tag{33}$$

The key point is that no architectural redesign is required. The separator remains a FiLM-conditioned masker; only its input changes from encoder latents $Z$ to lookup embeddings $E$ reconstructed from the code stream.

### B.4.2  Why the $Z \to E$ substitution is reasonable

The feasibility of this extension follows from how codec reconstruction works. At the operating bitrate, $E$ is the representation from which the decoder already reconstructs the waveform with high fidelity. Since CodecSep is a masker rather than a generator, it relies on preserving and modulating the latent content already present in the codec representation. That logic carries over from $Z$ to $E$: if $E \approx Z$ sufficiently well for codec reconstruction, then $E$ also contains the semantic and structural information needed for mask-based source selection.

This does not mean $Z$ and $E$ are identical. The residual gap between continuous-latent and bitstream-path performance reflects the quantization mismatch. But the design remains well matched to code streams precisely because it requires selection over an existing latent representation rather than conditional synthesis from raw audio. In practice, this is why the same trained masker remains competitive on the $E$ path even without fine-tuning, and why light fine-tuning or an embedding-alignment term such as

$$\mathcal{L}_{\text{emb}} = \sum_s \|\tilde{E}_s - Z_s\|_1 \tag{34}$$

can further narrow the gap.

### B.4.3  Why the deployment advantage is fundamentally systems-level

The deployment argument is strongest in the code-stream setting, where edge devices already encode audio and transmit compressed streams rather than raw waveforms.

For conventional spectrogram-domain or audio-stream separators, server-side separation in such a pipeline requires:

$$\text{decode} \; \to \; \text{separate on waveform/STFT} \; \to \; \text{re-encode}. \tag{35}$$

By contrast, CodecSep can operate directly on codec representations:

$$\text{codes in} \; \to \; \text{lookup / mask in codec domain} \; \to \; \text{codes out}. \tag{36}$$

Let $C_{\text{Enc}}$ and $C_{\text{Dec}}$ denote codec encode/decode costs, $C_{\text{Spec}}$ the cost of a spectrogram-domain separator, and $C_{\text{Mask}}$ the cost of the CodecSep masker. Then for code-stream input,

$$\text{AudioSep-like pipeline} : C_{\text{Dec}} + C_{\text{Spec}} + C_{\text{Enc}}, \tag{37}$$

$$\text{CodecSep pipeline} : C_{\text{Mask}}, \tag{38}$$

up to negligible codebook lookup and quantization overhead, and omitting CLAP text encoding cost since it is shared. This is why the efficiency claim in the paper is specifically framed as a code-stream deployment advantage rather than as a universal claim over all possible input settings.

### B.4.4 Operational implications

This design has several direct consequences in realistic deployment scenarios:

- **No redundant decode–re-encode loop on the server:** when inputs are already codec streams, the server can remain in codec space throughout separation.

- **Lower memory and bandwidth pressure during separation:** the separator operates on the compact $Z/E$ representation rather than on large spectrogram tensors.

- **Single-pass conditioning:** FiLM inside the masker introduces negligible overhead and requires no iterative sampling.

- **Interface compatibility:** the same model supports continuous-latent analysis, codes-in inference, and codes-out deployment with minimal changes.

- **Preservation of codec-organized structure:** separation is performed by modulating the representation already used for reconstruction, rather than reconstructing a new waveform representation and re-encoding it.

### B.4.5 Why the deployment path is important beyond source separation

The deployment path is not merely an implementation detail of CodecSep; it is one of the broader methodological contributions of the work. The main point is that CodecSep demonstrates a concrete pattern for building *codec-native* audio systems: once audio is already represented in a neural codec space, downstream processing need not leave that space unless the final application explicitly requires waveform reconstruction. In that sense, the code-stream pathway described in this paper can be viewed as a general blueprint for deployment-oriented audio processing systems, not only for text-guided source separation.

The traditional design pattern in many audio applications is still

$$\text{compressed audio} \rightarrow \text{decode to waveform} \rightarrow \text{task-specific processing} \rightarrow \text{re-encode if needed.} \tag{39}$$

This pattern is convenient from a modeling standpoint because most legacy methods are defined in waveform or spectrogram space, but it is inefficient from a systems standpoint. Once an edge device or upstream service has already paid the cost of encoding the signal into a neural codec representation, repeatedly returning to waveform space creates avoidable latency, memory traffic, and energy overhead. It also breaks representational continuity across the processing stack.

CodecSep instead follows a different deployment principle:

$$\text{compressed audio} \rightarrow \text{codec-space processing} \rightarrow \text{compressed output or optional decode.} \tag{40}$$

What makes this important is that the principle is not inherently tied to separation. It suggests a broader architecture for future deployed audio systems in which the codec representation functions as a common operating space for multiple downstream tasks.

Concretely, the same code-stream design pattern could be relevant to a wide range of audio processing applications. Examples include:

- **Target speaker extraction or enhancement:** instead of reconstructing waveform audio before enhancement, a model could directly reweight or refine codec latents conditioned on speaker identity or enrollment information.

- **Speech denoising and dereverberation:** masking or residual correction in codec space could remove nuisance components while preserving the compressed representation used by the communication pipeline.

- **Prompt-guided audio editing:** applications such as suppressing background music, attenuating environmental noise, or selectively modifying semantic audio attributes could be expressed as conditional transformations in codec space.

- **Audio event extraction or stream routing:** in streaming systems, codec-space masks or selectors could be used to isolate, prioritize, or route specific audio content without a full decode–process–re-encode loop.

- **Multi-stage edge–server audio processing:** when different modules operate at different locations in the system, a codec-native interface allows those modules to exchange compact representations rather than repeatedly materializing waveform audio.

The broader significance is therefore architectural. A codec-native deployment path provides a reusable interface between representation learning and downstream audio processing. In conventional pipelines, the codec is often treated as a peripheral compression component placed before or after the "real" model. The formulation used in CodecSep suggests a different perspective: the codec can be treated as a *shared representational backbone*, and downstream models can be designed to operate directly on that backbone. This opens the possibility of modular audio systems in which compression, transmission, storage, and task-specific processing are no longer separate stages with incompatible representations, but parts of a unified codec-space pipeline.

This viewpoint is especially relevant in practical settings where compute, memory, bandwidth, or latency matter. If a downstream task can be performed directly on codec latents or codec-derived embeddings, then the system may avoid not only redundant decode/re-encode operations, but also the cost of constructing large spectrogram or waveform-domain intermediate tensors. That advantage compounds when several processing stages are chained together. In such cases, the codec-native path is not simply a local optimization for one separator; it becomes a systems design principle for scalable audio inference.

For this reason, we view the deployment discussion in CodecSep as important beyond the immediate separation results. The paper instantiates one concrete example—text-guided universal sound separation—but the same deployment logic can guide the design of other efficient audio models that must operate under realistic communication and inference constraints. In that sense, the proposed code-stream path is best understood not only as an implementation route for CodecSep, but as a template for future codec-aware audio processing systems.

### B.5 Summary of the design rationale

The design of CodecSep is guided by a single overarching principle: if a neural audio codec already provides a compact and perceptually organized latent representation, then prompt-guided source extraction can be formulated more effectively as *conditional masking in codec latent space* than as spectrogram-domain separation or decoder-style source generation. The preceding discussion motivates this claim from the perspectives of representation, architecture, optimization, and deployment.

**Why NAC latents?** NAC latents are substantially more compact than spectrogram representations, which immediately reduces the computational and memory burden of the separator. More importantly, they are not merely smaller tensors. Through codec pretraining, they are already shaped to preserve perceptually meaningful acoustic structure under compression. This means the separator does not need to learn a task-specific representation from a large and relatively unstructured time–frequency input. Instead, it can operate on a latent space that has already been compressed and organized by the codec, shifting the problem from representation learning toward source-selective modulation.

**Why masking rather than generation?** Once the codec latent space is viewed as a structured representation of the mixture, masking becomes the natural separation interface. A masking model is asked to determine which latent components should be preserved or attenuated for a given query, rather than to synthesize a new target representation from scratch. This better exploits the codec prior, imposes a more constrained and stable learning problem, and limits the model to modulating mixture-supported latent

content. In that sense, the architecture is intentionally selection-centric: the separator is designed to extract from an existing latent organization, not to replace it with a newly generated one.

**Why FiLM inside the masker?**   The text query should influence the *selection mechanism* rather than alter the codec representation itself. Injecting FiLM inside the transformer masker achieves exactly this. It provides a lightweight and explicit conditioning mechanism that biases the hidden features toward query-relevant latent structure while leaving the frozen codec backbone unchanged. This preserves the codec manifold and makes the role of conditioning operationally clear: the text embedding guides which latent channels or components are emphasized or suppressed during masking.

**Why continuous latents in the main experiments?**   Using continuous encoder latents $Z$ isolates the separator behavior in the cleanest setting. It provides straightforward gradient flow through the masker and decoder, avoids complications associated with discrete code optimization, and preserves the richest view of the codec-induced latent structure. It also makes the qualitative analysis most interpretable, since the channel–time organization of the latent space can be inspected directly. For these reasons, the $Z$ domain is the most appropriate setting for establishing the core separation mechanism before turning to the discrete deployment path.

**Why does the same design extend naturally to deployment?**   The extension to code-stream deployment follows directly from the fact that CodecSep is a masker rather than a generator. Replacing encoder latents $Z$ with codec lookup embeddings $E$ preserves the essential operation of the model: query-conditioned modulation of an existing codec representation. Because the separator does not depend on reconstructing a waveform-domain representation before processing, the same architecture can support codes-in / codes-out inference with minimal modification. This is what makes the deployment path more than a narrow efficiency trick; it reflects a codec-native design that is well matched to realistic edge–server audio pipelines and, more broadly, suggests a reusable blueprint for other deployment-oriented audio processing tasks.

Overall, the qualitative latent analysis and the oracle codec comparison provide supporting evidence for these design choices. The latent visualizations indicate that the codec representation carries source-dependent structure and that the learned masks exploit this structure primarily through channel-wise modulation. The oracle analysis shows that the codec provides a strong, though not lossless, reconstruction substrate, helping disentangle codec-limited distortion from separation-specific error. Taken together, these results support the central architectural premise of CodecSep: modern neural audio codec representations are structured enough that prompt-guided source extraction can be effectively realized as FiLM-conditioned masking in codec space, with the additional benefit that the same formulation aligns naturally with efficient codec-native deployment.

# C  Dataset Details

## C.1  Divide and Remaster v 2.0 (dnr-v2)

dnr-v2 Petermann et al. (2022) dataset consists of $60s$-duration artificial mixtures of speech, music, and SFX sampled from LibriSpeech Panayotov et al. (2015), Free Music Archive (FMA) Defferrard et al. (2017), and Freesound Dataset $50K$ (FSD50$K$) Fonseca et al. (2022), respectively. It includes $3,406$ ($56.7hrs$) training, $487$ ($8.13hrs$) validation, and $973$ ($16.22hrs$) test mixtures, each provided with its three individual source audios. The mixtures are generated by normalizing each source to fixed Loudness Units Full-Scale (LUFS) levels: $-17$ dB (speech), $-24$ dB (music), and $-21$ dB (SFX), with $\pm2$ dB random perturbations. Any source exceeding a peak threshold is normalized to 0.5 dB. The sources are mixed and normalized to $-27$ dB LUFS with additional random perturbations. The validation and test sets are trimmed for silence and split into $5s$ or $10s$ segments. Segments where sources are present for less than 50% of the duration are removed, resulting in $2,852$ ($\approx 3.96hrs$) validation and $1,840$ ($\approx 5.11hrs$) test mixtures.

While originally developed for 3-stem separation, we adapt dnr-v2 to the USS setting by replacing fixed source labels with natural language descriptions. For speech or music stem, we use broad, category-level prompts (e.g., "speech," "music"), reflecting realistic usage in production workflows. In contrast, SFX sources are more complex—often containing three or more overlapping events. We generate prompts to query the SFX stem using FSD50K's hierarchical annotations, combining fine-grained class labels with their parent categories. This results in long-form, compositional queries that reflect the structure of the mixture (e.g., "dog barking, Animal, engine rumbling, motor vehicle").

## C.2  Open-Domain Benchmarks

We benchmark on five open-domain datasets spanning captioned audio, environmental sounds, and large multi-event corpora: AudioCaps Kim et al. (2019), an AudioSet-derived collection of $> 46$k $10s$ YouTube clips paired with human-written captions describing the dominant sound events (used by us to synthesize training and test mixtures); ESC-50 Piczak, a curated environmental sound dataset of 2,000 clips ($5s$ each) organized into 50 classes with 40 examples per class across five meta-categories (animals, natural, human non-speech, domestic, exterior/urban); Clotho-v2 Drossos et al. (2020), 6,974 audio samples ($15$–$30s$) each annotated with five human captions ($8$–$20$ words) covering open-domain events; AudioSet Gemmeke et al. (2017), the evaluation split of AudioSet comprising human-labeled $10s$ YouTube clips over an ontology of 632 audio event classes in a multi-label setting; and VGGSound Chen et al. (2020a), an AudioSet-derived audio–visual corpus with $550+$ hours of $10s$ segments covering a wide variety of everyday sound categories. For AudioCaps we form both training and testing mixtures (same scale of test data as dnr-v2) by summing three clips (validation segmented into $5s$, test preserves clips up to $20s$), while for ESC-50, Clotho-v2, AudioSet-eval, and VGGSound we construct test-only mixtures using the same three-clip protocol.

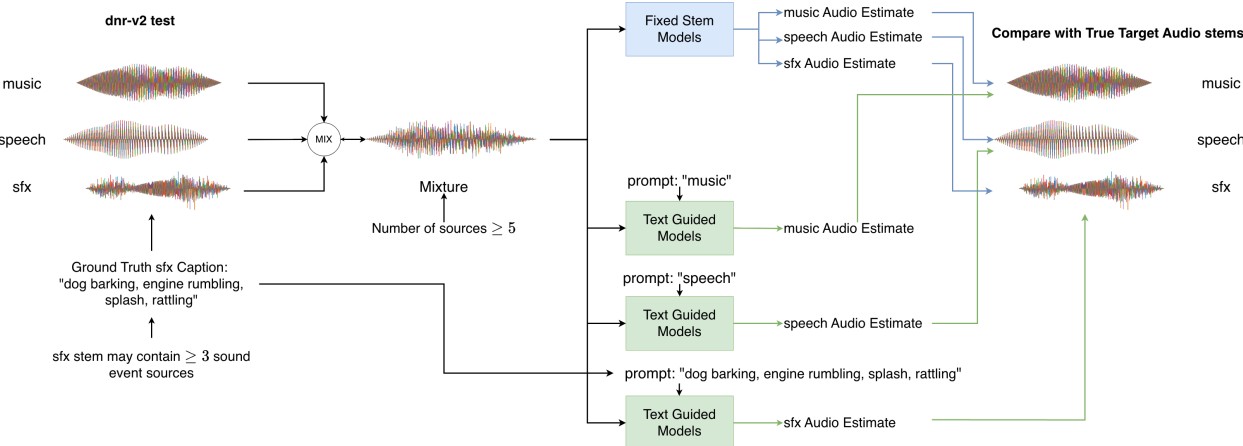

Figure 4: Evaluation workflow for dnr-v2. Each mixture contains multi-source stems: speech (often multi-speaker), music (multi-instrument), and SFX ($\geq 3$ overlapping events). Fixed-stem baselines predict a fixed set of outputs (e.g., 3 stems), whereas CodecSep and other text-guided models generate only the prompted source. Speech and music are evaluated using generic prompts, while SFX uses long-form compositional prompts listing all SFX events in each mixture. Extracted signals are compared with ground-truth category stems using SI-SDR and ViSQOL.

# D   Evaluation Details

## D.1   Divide and Remaster v 2.0 (dnr-v2)

The dnr-v2 benchmark presents a challenging open-domain separation setting: although the dataset provides three category labels—speech, music, and sound effects—each category represents a multi-source stem. A single mixture frequently contains five to ten underlying acoustic sources, including overlapping speakers, multiple musical instruments, and several sound-effect events occurring either concurrently or in sequence. The three reported stems are therefore semantic groupings that support interpretability and reproducibility, rather than an indication that the mixture contains only three sources. Any evaluation methodology must respect this structure. Figure 4 provides an overview of our dnr-v2 evaluation workflow and highlights how these multi-source stems are handled across fixed-stem un-guided and text-guided models.

For fixed-stem unguided architectures, evaluation is performed by mapping each predicted output stem to one of the three ground-truth stems and computing SI-SDR and ViSQOL on a per-category basis. Importantly, we do not employ PIT or MixIT training objectives for these baselines; instead, we train dedicated three-stem models that directly predict speech, music, and SFX stems.

Text-guided models, including CodecSep, follow a fundamentally different inference and evaluation paradigm. Speech and music stems are recovered using generic prompts ("speech," "music"), which reliably capture their multi-source content. In contrast, SFX stems require mixture-specific prompts because sound effects span a wide and open-domain label space. For each mixture, we use a long-form compositional prompt enumerating all SFX events present in the ground truth. This ensures that the model has sufficient semantic context to extract the full SFX stem. The number of sfx events present in a mixture varies considerably. We additionally perform an ambiguous-prompt evaluation, where deliberately underspecified prompts for speech and music are used to assess robustness to vague or incomplete semantic queries. After inference, the extracted waveform for each category is directly compared with the corresponding ground-truth stem using SI-SDR and ViSQOL. This evaluation design ensures fairness between fixed-stem and text-guided systems while faithfully reflecting the multi-source structure of dnr-v2 mixtures

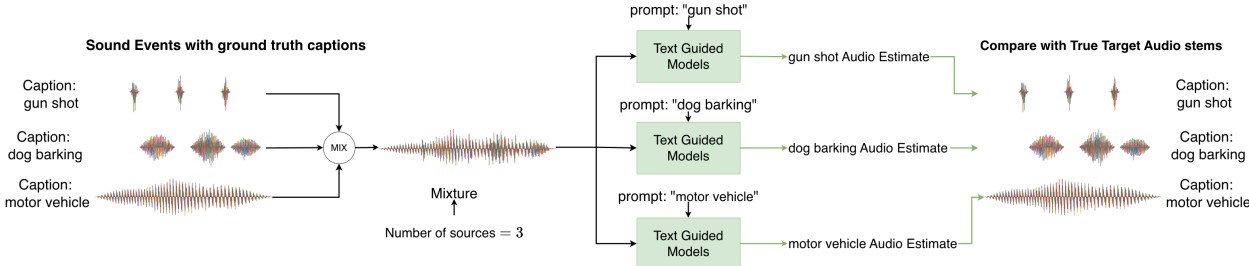

Figure 5: Evaluation workflow for the standardized three-source benchmarks (AudioCaps, ESC-50, Clotho, VGGSound, and AudioSet-eval). Following prior USS protocols, each mixture is constructed by combining three isolated events drawn from distinct classes. For each class, the corresponding textual prompt is supplied to the separator (e.g., "dog barking," "gun shot," "motor vehicle"), and the extracted signal is compared with the ground-truth isolated source using SI-SDR and ViSQOL.

## D.2 Open-Domain Benchmarks

For AudioCaps, ESC-50, Clotho, VGGSound, and AudioSet-eval, we adopt the standardized three-source mixture protocol. Following established practice, each mixture is constructed by combining three isolated events drawn from different classes. Each source is then extracted by the text-guided models using its corresponding textual caption as prompt, and evaluation metrics are computed against the ground-truth isolated audio. Figure 5 illustrates the evaluation workflow used for these benchmarks, highlighting how class-specific prompts are applied and how the resulting predictions are matched against the ground-truth isolated sources. Although these datasets do not reflect the complex multi-source structure of dnr-v2, the standardized 3-way protocol enables direct benchmarking against prior work (AudioSep) under consistent conditions.

# E  Training Details

The complete model, including the query module $query(.)$, is trained for $400K$ iterations with DAC Kumar et al. (2023) and CLAP Wu et al. (2023) modules frozen. Validation is conducted every $5K$ iterations and test every $10K$ iterations. We use ADAM Kingma & Ba (2017) as our optimizer and train with a batch size of 4 examples, each 2 seconds in duration, and a learning rate of $1.5e^{-4}$ on a single 24GB NVIDIA A-30 GPU. Training employs a $ReduceLRonPlateau$ Mukherjee et al. (2019) scheduler, which reduces the learning rate by a factor of 0.5 if the validation loss does not improve for two consecutive validation checks. We train two versions of CodecSep, one using the dnr-v2 dataset and the other using AudioCaps, to evaluate performance across different training distributions. We refer to these models using the suffixes +dnr-v2 and +AudioCaps, respectively, to indicate which dataset each model was trained on.

Since TDANet and CodecFormer were originally designed for speech separation, we re-train newly initialized 3-stem versions on the dnr-v2 training set using the same configuration as CodecSep. We also train a 3-stem Sudo rm-rf model to compare against compute-efficient separators. For AudioSep, we evaluate both the publicly available pretrained model—trained on diverse datasets—and versions re-trained on dnr-v2 and AudioCaps for consistency. We include SDCodec using the official pretrained checkpoints released by the authors. Finally, we incorporate the USS-pretrained variant of BiModalSS and re-train SudoRmRf+FiLM on dnr-v2 with CLAP text conditioning. To ensure a fair comparison, all inputs to TDANet, Sudo rm-rf, AudioSep, BiModalSS, and Sudo rm-rf+FiLM undergo codec processing with a full-band stereo-capable 48 kHz EnCodec during training and inference. This accounts for codec-induced distortions and artifacts, reflecting realistic deployment scenarios where audio is typically processed through compression pipelines in cloud-based systems.

Table 11: Generalization and transfer results for universal sound separation.

(a) Generalization on **AudioCaps-test**.

| Model | Separation | |
|---|---|---|
| | **SI-SDR (↑)** | **ViSQOL (↑)** |
| AudioSep | $\mathbf{-2.5^{\pm 12.14}}$ | $\mathbf{2.4^{\pm 1.08}}$ |
| AudioSep + dnr-v2 (zero-shot) | $-6.4^{\pm 11.48}$ | $2.3^{\pm 1.08}$ |
| **CodecSep + dnr-v2** (zero-shot) | $\mathbf{-6.1^{\pm 11.62}}$ | $2.2^{\pm 1.16}$ |
| AudioSep + AudioCaps | $-9.2^{\pm 18.71}$ | $2.3^{\pm 1.11}$ |
| **CodecSep + AudioCaps** | $\mathbf{-6.2^{\pm 10.58}}$ | $2.1^{\pm 1.00}$ |

(b) Transfer to **dnr-v2-test** when trained on AudioCaps (zero-shot on dnr-v2).

| Model | Metric (↑) | Music | Speech | Sfx |
|---|---|---|---|---|
| AudioSep + AudioCaps | SI-SDR | $-14.9^{\pm 23.08}$ | $-7.1^{\pm 25.80}$ | $-14.6^{\pm 23.26}$ |
| (zero-shot) | ViSQOL | $2.4^{\pm 0.71}$ | $2.4^{\pm 0.70}$ | $2.2^{\pm 0.79}$ |
| **CodecSep + AudioCaps** | SI-SDR | $-8.5^{\pm 2.78}$ | $2.5^{\pm 2.91}$ | $-5.9^{\pm 4.33}$ |
| (zero-shot) | ViSQOL | $2.3^{\pm 0.53}$ | $2.6^{\pm 0.47}$ | $2.1^{\pm 0.72}$ |

## F Cross-benchmark performance under two transfer directions: AudioCaps and dnr-v2.

**Why analyze AudioCaps separately as a cross-domain transfer case (cf. Table 11a)?** We include **AudioCaps** as a deliberate *stress test* for cross-domain generalization. Unlike the other external benchmarks, where the dnr-v2-trained CodecSep model remains favorable overall, AudioCaps exposes a more difficult transfer setting because it differs substantially from dnr-v2 in both source composition and prompt distribution, while the official pretrained **AudioSep** also benefits from broader upstream exposure to AudioSet-like data. We therefore use AudioCaps not mainly to showcase a success case, but to examine a *harder failure-case regime* and better understand the limits of transfer beyond the training distribution.

**Does codec-latent masking generalize competitively to AudioCaps when trained on dnr-v2 (cf. Table 11a)?** Under zero-shot transfer from **dnr-v2** to **AudioCaps-test**, **CodecSep+dnr-v2** is slightly better than **AudioSep+dnr-v2** in SI-SDR, while ViSQOL remains close. However, neither retrained model matches the official pretrained **AudioSep**, which remains strongest overall on AudioCaps. We therefore do *not* interpret AudioCaps as a strong positive transfer result for the dnr-v2-trained model. Rather, we view it as a useful stress test showing that, although codec-latent masking remains competitive relative to a matched retrained baseline, transfer to AudioCaps is substantially more challenging than transfer to the other external benchmarks considered in the paper.

**Under matched AudioCaps training, does codec-latent masking remain competitive with spectrogram-domain separation (cf. Table 11a)?** Yes. When both models are trained directly on **AudioCaps**, **CodecSep+AudioCaps** achieves better SI-SDR than **AudioSep+AudioCaps**, while **AudioSep+AudioCaps** retains a small advantage in ViSQOL. This shows that the weaker dnr-v2→AudioCaps transfer is not because CodecSep is fundamentally unsuitable for this benchmark; rather, it reflects the difficulty of cross-domain transfer into AudioCaps. Under matched AudioCaps training, codec-latent masking remains competitive and again shows the clearer advantage in signal-level separation quality.

**Does training on AudioCaps transfer effectively to the more structured and denser dnr-v2 mixtures (cf. Table 11b)?** Table 11b evaluates the reverse transfer direction: models trained on **AudioCaps** and tested on **dnr-v2**. Here, **CodecSep+AudioCaps** performs better than **AudioSep+AudioCaps** in SI-SDR across all three stems, with especially large gains for speech and SFX, while ViSQOL remains

broadly comparable. This indicates that the transfer picture is asymmetric: AudioCaps is a difficult target for dnr-v2-trained models, but AudioCaps-trained CodecSep transfers reasonably well to dnr-v2 at the signal level.

**Main takeaways.** Taken together, Tables 11a and 11b show that **AudioCaps serves as a useful stress test for cross-domain generalization**. In contrast to the other external benchmarks, the **dnr-v2-trained** model does not show a strong transfer win on AudioCaps in absolute terms, even though it remains competitive with a matched retrained AudioSep baseline. At the same time, under matched AudioCaps training, CodecSep again outperforms AudioSep in SI-SDR, and the AudioCaps-trained CodecSep model transfers favorably to dnr-v2. We therefore interpret this section not as a blanket claim of uniform cross-domain superiority, but as evidence that CodecSep remains competitive under substantial distribution shift while also revealing an informative failure case that helps define the boundary of its generalization.

Table 12: Relative gains (%) of **CodecSep** over **AudioSep** under matched training/prompt settings. Each sub-table reports percent improvements for a specific evaluation setup.

(a) DnR-v2 test set

| Metric | Relative Gain (%) | | |
|---|---|---|---|
| | **Speech** | **Music** | **SFX** |
| SI-SDR | **+29.8** | **+120.7** | **+119.1** |
| ViSQOL | **+26.1** | **+10.5** | **+0.5** |

(b) Ambiguous prompts (speech & music paraphrases)

| Metric | Relative Gain (%) | |
|---|---|---|
| | **Speech** | **Music** |
| SI-SDR | **+1.2** | **+13.0** |
| ViSQOL | **+3.8** | **+1.2** |

(c) Additional open-domain benchmarks

| Metric | AudioCaps | ESC-50 | Clotho-v2 | AudioSet | VGGSound |
|---|---|---|---|---|---|
| SI-SDR | **+5.5** | **+24.3** | **+30.0** | **+16.4** | **+13.0** |
| ViSQOL | **−4.3** | **+2.2** | **+2.4** | **+2.9** | **+2.9** |

(d) Training on AudioCaps

| Metric | AudioCaps-test | dnr-v2 | | |
|---|---|---|---|---|
| | | **Music** | **Speech** | **SFX** |
| SI-SDR | **+32.5** | **+43.1** | **+134.7** | **+59.5** |
| ViSQOL | **−6.5** | **−3.8** | **+5.4** | -4.2 |

## G   Discussion of Relative–Gain Summaries.

Tables 12a–12d summarize the relative gains of CodecSep over AudioSep under matched training data and prompt settings, and are intended to complement—not replace—the absolute results reported in the main text.

**Under matched training on dnr-v2, does codec-latent masking improve over spectrogram-domain text-guided separation (cf. Table 12a)?** Table 12a summarizes the relative gains of CodecSep over AudioSep on **dnr-v2** under matched training and prompt settings. CodecSep yields positive SI-SDR gains on all three stems, with particularly strong relative improvements for music and SFX. ViSQOL also improves for speech and music, while the SFX perceptual gain is essentially neutral. Overall, these results support a clear advantage for codec-latent masking on dnr-v2, especially in *signal-level separation quality*. Since relative percentages can look large when the corresponding baseline values are small, the appendix presents them as a compact summary alongside the absolute results in the main text; taken together, both views point to the same conclusion that CodecSep consistently improves over AudioSep on this benchmark.

**Does the method remain effective under prompt variation (cf. Table 12b)?** Table 12b evaluates relative gains under **paraphrased / ambiguous prompts**. Here, the gains remain positive but are clearly smaller than those on the standard dnr-v2 setting. This suggests that CodecSep retains some robustness to lexical variation, but also that prompt ambiguity reduces the margin between methods. We therefore interpret this table as evidence of *continued effectiveness under prompt variation*, rather than as showing strong invariance to paraphrase.

**Across additional open-domain benchmarks, are the advantages consistent across metrics (cf. Table 12c)?** Table 12c shows that SI-SDR relative gains are positive across all five additional benchmarks, suggesting that the signal-level advantage of CodecSep is fairly consistent in cross-domain evaluation. However, the ViSQOL gains are smaller and more mixed: they are positive on ESC-50, Clotho-v2, AudioSet, and VGGSound, but negative on AudioCaps. We therefore interpret these results as showing a more consistent benefit in separation quality than in perceptual quality. In other words, the cross-benchmark trend is favorable overall, but not uniformly strong across all metrics and datasets.

**When trained on AudioCaps, does codec-latent masking transfer effectively across datasets (cf. Table 12d)?** Table 12d summarizes the relative gains of CodecSep over AudioSep when both models are trained on **AudioCaps**. In this setting, CodecSep shows positive SI-SDR gains both on **AudioCaps-test**

and on all three **dnr-v2** stems, with the largest relative improvement on speech. The ViSQOL results are more mixed, with a positive gain only for speech and small negative differences on AudioCaps-test, music, and SFX. We therefore interpret this result more specifically as showing that, when trained on AudioCaps, CodecSep retains a clear advantage in *signal-level transfer* across datasets, even though the perceptual gains are less uniform. More broadly, the appendix shows that cross-benchmark generalization is strongest for the **dnr-v2-trained** CodecSep model, which transfers competitively across multiple external benchmarks.

**Main takeaways.** Taken together, Tables 12a–12d suggest that CodecSep generally provides positive relative gains over AudioSep under matched training and prompt protocols, especially in SI-SDR. The trend is strongest on dnr-v2 and remains visible under cross-benchmark transfer, while becoming smaller under prompt ambiguity. At the same time, the perceptual gains are more modest and sometimes mixed, and the relative percentages can overstate practical impact when the baseline values are small. For this reason, we view these summaries as a compact complement to the absolute tables: they highlight the overall direction of improvement, but should be interpreted alongside the underlying absolute results and variance estimates.

Table 13: Results: Reconstruction Performance, Universal Sound Separation (**dnr-v2-test**)

| Model | Metric (↑) | Reconstruction | | | |
|---|---|---|---|---|---|
| | | **Mixture** | **Music** | **Speech** | **Sfx** |
| **3-Stem: Fixed Stem, Non Text-guided** | | | | | |
| TDANet | SI-SDR | $-3.3^{\pm 7.78}$ | $\mathbf{8.0^{\pm 5.29}}$ | $\mathbf{11.1^{\pm 3.32}}$ | $\mathbf{4.7^{\pm 5.11}}$ |
| | ViSQOL | $3.9^{\pm 0.35}$ | $\mathbf{4.2^{\pm 0.46}}$ | $\mathbf{4.5^{\pm 0.32}}$ | $\mathbf{4.1^{\pm 0.39}}$ |
| CodecFormer | SI-SDR | $-47.6^{\pm 9.51}$ | $-47.1^{\pm 10.97}$ | $-47.8^{\pm 9.65}$ | $-48.2^{\pm 9.87}$ |
| | ViSQOL | $1.0^{\pm 0.07}$ | $1.0^{\pm 0.12}$ | $1.0^{\pm 0.06}$ | $1.2^{\pm 0.47}$ |
| **CodecSep + dnr-v2** | SI-SDR | $3.4^{\pm 1.85}$ | $4.1^{\pm 3.97}$ | $6.2^{\pm 2.87}$ | $0.8^{\pm 5.16}$ |
| **(unguided, 3-stem)** | ViSQOL | $3.2^{\pm 0.20}$ | $3.0^{\pm 0.33}$ | $3.5^{\pm 0.24}$ | $3.2^{\pm 0.46}$ |
| SDCodec | SI-SDR | $\mathbf{7.0^{\pm 2.49}}$ | $7.7^{\pm 4.60}$ | $8.3^{\pm 3.26}$ | $2.5^{\pm 5.65}$ |
| | ViSQOL | $\mathbf{4.3^{\pm 0.15}}$ | $4.0^{\pm 0.28}$ | $4.4^{\pm 0.15}$ | $4.0^{\pm 0.34}$ |
| **Text-guided** | | | | | |
| AudioSep | SI-SDR | $5.5^{\pm 1.96}$ | $4.7^{\pm 5.36}$ | $11.0^{\pm 2.99}$ | $-2.0^{\pm 5.68}$ |
| (zero-shot) | ViSQOL | $4.1^{\pm 0.38}$ | $3.8^{\pm 0.65}$ | $4.6^{\pm 0.13}$ | $3.2^{\pm 0.77}$ |
| AudioSep + dnr-v2 | SI-SDR | $6.5^{\pm 2.26}$ | $8.0^{\pm 4.55}$ | $8.1^{\pm 3.35}$ | $2.3^{\pm 5.95}$ |
| | ViSQOL | $4.2^{\pm 0.18}$ | $4.1^{\pm 0.21}$ | $3.0^{\pm 0.29}$ | $3.8^{\pm 0.47}$ |
| **CodecSep + dnr-v2** | SI-SDR | $4.1^{\pm 2.06}$ | $3.9^{\pm 3.93}$ | $6.1^{\pm 2.86}$ | $0.7^{\pm 5.29}$ |
| | ViSQOL | $3.7^{\pm 0.22}$ | $3.4^{\pm 0.33}$ | $3.8^{\pm 0.24}$ | $3.5^{\pm 0.44}$ |
| **CodecSep + dnr-v2** | SI-SDR | $\mathbf{12.2^{\pm 2.42}}$ | $\mathbf{12.6^{\pm 3.81}}$ | $\mathbf{13.6^{\pm 2.59}}$ | $\mathbf{8.7^{\pm 4.17}}$ |
| (ablate **Masker**) | ViSQOL | $\mathbf{4.4^{\pm 0.14}}$ | $\mathbf{4.1^{\pm 0.31}}$ | $\mathbf{3.9^{\pm 0.34}}$ | $3.8^{\pm 0.54}$ |
| AudioSep + AudioCaps | SI-SDR | $6.7^{\pm 2.52}$ | $8.1^{\pm 4.63}$ | $8.4^{\pm 3.21}$ | $2.4^{\pm 6.12}$ |
| (zero-shot) | ViSQOL | $4.2^{\pm 0.19}$ | $4.1^{\pm 0.21}$ | $4.2^{\pm 0.21}$ | $\mathbf{3.8^{\pm 0.46}}$ |
| **CodecSep + AudioCaps** | SI-SDR | $0.6^{\pm 1.89}$ | $-0.2^{\pm 5.15}$ | $-11.^{\pm 5.21}$ | $1.2^{\pm 4.84}$ |
| (zero-shot) | ViSQOL | $3.3^{\pm 0.23}$ | $2.9^{\pm 0.64}$ | $1.7^{\pm 0.48}$ | $3.4^{\pm 0.41}$ |

## H Further Studies: Reconstruction Performance.

**Is masking preferable to generation in codec latent space for reconstruction (cf. Table 13, fixed-stem non-text-guided block)?** Table 13 evaluates reconstruction in a **single-source reconstruction** setting, where each model—although trained for *source separation*—is given an isolated source and asked to reproduce it; we also report **mixture reconstruction** by summing the separated outputs and comparing the result to the original mixture. Within the non-text-guided codec-latent models, the results suggest that *masking-based separation* is more stable than decoder-style latent generation. In particular, **CodecSep+dnr-v2 (unguided, 3-stem)** substantially outperforms **CodecFormer** on both per-source and mixture reconstruction, indicating that reweighting information already present in the codec latent space is more effective than attempting to regenerate source latents through a decoder. At the same time, this should not be interpreted as saying that codec-latent masking is best overall for reconstruction. **TDANet** remains strongest on single-source reconstruction, while **SDCodec** gives the best mixture reconstruction; importantly, SDCodec is architecturally advantaged for this setting because it uses *separate source-specific codebooks* to reconstruct sources from mixture audio. We therefore interpret this comparison more specifically as showing that, among codec-latent separation models trained for source separation, *explicit masking is a more reliable mechanism than direct latent generation.*

**Under text guidance, does CodecSep reconstruct as faithfully as spectrogram-domain AudioSep (cf. Table 13, text-guided block)?** Under text guidance, the two model families generate outputs in different ways. **AudioSep** predicts a text-conditioned mask in the STFT domain and reconstructs the waveform from the masked time–frequency representation, whereas **CodecSep** applies text-conditioned modulation / masking in the NAC latent space and then decodes the resulting latent representation back to audio through the frozen codec decoder. In the reconstruction setting of Table 13, **AudioSep** and **AudioSep+dnr-v2** generally achieve stronger scores than **CodecSep+dnr-v2**, especially for mixture reconstruction and for music / speech SI-SDR, and ViSQOL follows a similar overall pattern. We therefore do not interpret reconstruction as a regime in which CodecSep is uniformly stronger than spectrogram-domain masking. Instead, these results suggest that AudioSep's STFT-domain masking and waveform resynthesis pipeline is better suited to high-fidelity source-preserving reconstruction, while CodecSep is primarily optimized for *source-selective separation* in compressed latent space rather than exact waveform-faithful reconstruction. CodecSep nevertheless remains competitive on some SFX reconstruction metrics.

**What does the masker ablation reveal about reconstruction versus separation (cf. Table 13 and Table 6)?** The **masker-ablated CodecSep** variant achieves the strongest reconstruction scores in Table 13, but performs poorly as a separator in Table 6. We interpret this as showing that *source-consistent reconstruction* and *source extraction from mixtures* require different mechanisms. In the ablated variant, FiLM conditioning is applied directly to the intermediate layers of the NAC encoder, and the decoder reconstructs audio from this conditioned encoder representation. In the single-source reconstruction setting, where the input already contains only the target source and the prompt matches that source, this direct FiLM-based affine transformation can preserve or enhance source-relevant structure, leading to very strong reconstruction. However, in mixtures, the same affine modulation tends to *collapse the latent space* toward a prompt-conditioned representation rather than preserving the separable source structure needed for disentanglement. As a result, direct FiLM conditioning of the encoder is not sufficient for mixture separation, since it does not provide the explicit source-selection behavior needed to isolate one source from competing content. These results therefore suggest that, while encoder-side FiLM conditioning can support strong source-consistent reconstruction, an *explicit masker* is needed to preserve usable source structure and extract a target source from a mixture.

**Does cross-dataset transfer preserve reconstruction quality (cf. Table 13, AudioCaps-trained variants)?** The AudioCaps-trained variants show a mixed picture. **AudioSep+AudioCaps** remains fairly strong when transferred zero-shot to dnr-v2, whereas **CodecSep+AudioCaps** degrades substantially, especially for speech. We therefore do not view reconstruction as a robust cross-dataset strength of CodecSep under this training setup. This is consistent with the broader interpretation that CodecSep is better understood as a codec-latent separator than as a model optimized for faithful source reconstruction under distribution shift.

**Main takeaways.** Taken together, Table 13 supports four conclusions. First, among codec-latent source separation models, **explicit masking is clearly more effective than decoder-style latent generation**, as shown by the strong gap between CodecSep and CodecFormer. Second, this does *not* imply that codec-latent masking is the strongest reconstruction strategy overall: **TDANet** and especially **SDCodec** remain very strong baselines for reconstruction, with SDCodec benefiting from its use of separate source-specific codebooks. Third, under text guidance, **AudioSep** generally remains stronger for faithful source-preserving reconstruction, which is consistent with its STFT-domain masking and waveform resynthesis pipeline being better aligned with this objective. Finally, the masker ablation clarifies the functional role of the full CodecSep design: **direct FiLM-based affine conditioning inside the NAC encoder can reconstruct or enhance source-consistent content when the input already matches the prompt, but an explicit masker is needed to extract a target source from a mixture**. We therefore interpret this section not as evidence that CodecSep is optimized for reconstruction, but as a diagnostic analysis showing that its main strength lies in **source-selective separation through explicit masking in codec latent space**.

