# OpenReview forum: "CodecSep:  Prompt-Driven Universal Sound Separation on Neural Audio Codec Latents"
_TMLR — Accepted by TMLR_

### Review · Reviewer_RArg · 2026-02-28

**Summary Of Contributions:**

This paper introduces CodecSep, which proposes using neural codec latents as the working space for text-driven universal sound separation. Instead of operating directly on waveforms or spectrograms, the method performs masking in the continuous latent space of a frozen DAC encoder, with CLAP text embeddings injected through FiLM layers into a lightweight transformer masker. The main idea is that modern codec representations are already structured enough that separation can be done through masking alone, without explicitly regenerating audio.

The authors evaluate the system on dnr-v2 and several open-domain benchmarks, comparing against AudioSep, CodecFormer, SDCodec, TDANet, Sudo rm-rf, and others. CodecSep reports better SI-SDR than AudioSep under matched prompting and training protocols, similar perceptual quality in many cases, and substantially lower compute when operating directly on codec streams. The deployment motivation is clearly a central part of the paper: if audio already exists as codec bitstreams, avoiding decode–process–reencode cycles can significantly reduce compute.

**Audience:**

Yes

**Audience Explanation:**

The intersection between neural audio codecs and downstream audio tasks is timely and important. As codec-based pipelines become more common in real-world systems, performing processing directly in codec representations is a natural and practically meaningful direction. The deployment framing is intuitive. The text-conditioned, open-vocabulary aspect further broadens relevance to multimodal and language-audio communities. Even with the current limitations, the overall research direction is worthwhile and worth discussion by the community.

**Broader Impact Concerns:**

The paper currently lacks a broader impact discussion. A short discussion acknowledging risks and possible mitigations would strengthen the submission.

**Claims And Evidence:**

No

**Claims Explanation:**

The paper includes a large amount of experimentation, but some core claims are not fully supported by the evidence as currently presented.

(1) A major concern is comparison fairness. CodecSep operates using a 16 kHz DAC backbone, while AudioSep runs at 32 kHz. In addition, the paper routes non-codec baselines through a 48 kHz EnCodec during both training and inference to simulate “realistic deployment.” While the motivation is understandable, this introduces a strong confound: these baseline systems were not designed to operate under codec compression artifacts. As a result, it becomes difficult to tell whether CodecSep’s gains come from the proposed latent masking approach or simply from baselines being disadvantaged by an additional codec pass. Reporting baseline results both with and without this processing would make the comparison much clearer.

(2) The central scientific claim, that NAC latents are already sufficiently structured for masking-based separation, is also only indirectly supported. The paper demonstrates that masking works empirically, but does not actually analyze the latent space itself. There are no visualizations of mask behavior, no probing experiments showing how latent dimensions relate to sources, and no disentanglement analysis. The masking-versus-generation ablation is useful, but it does not explain why masking succeeds. A simple control experiment, such as performing masking on latents from an untrained encoder, would help isolate whether structure truly comes from the codec representation.

(3) The compute efficiency claim needs clearer framing. The reported ~54× reduction applies specifically to the codec-stream setting where encoding is already present. This is a somewhat reasonable deployment assumption, but it blends an architectural contribution with a systems-level one. In the audio-stream comparison, CodecSep is actually more expensive than AudioSep, which should be stated more explicitly.

(4) The MOS evaluation feels uneven. CodecSep retrained on dnr-v2 is compared against a pretrained AudioSep checkpoint rather than a retrained counterpart, which breaks the otherwise consistent matched-training philosophy of the paper. The study size is also relatively small and lacks statistical testing.

**Requested Changes:**

The most important issue to address is the codec-processing confound. Baseline systems should be evaluated both with and without the EnCodec pass, or the authors should provide stronger justification for why this setup represents the only fair comparison. As it stands, the main performance advantage is difficult to interpret.

The paper also needs direct evidence supporting the “structured latent” hypothesis. Some form of latent-space analysis would significantly strengthen the scientific contribution, whether through mask visualization, probing experiments, disentanglement metrics, or controls using untrained encoders. I suggest at least look into mechanistic interpretability literature on this topic and cite relevant articles as argument basis.

Given the high variance in several tables, statistical significance testing should be added for key comparisons. Claims of consistent improvement are otherwise difficult to verify.

The MOS study should compare against a retrained AudioSep baseline under matched conditions rather than only a pretrained checkpoint.

Beyond these critical points, several smaller clarifications would improve the paper. The sampling-rate mismatch should be discussed explicitly, especially regarding whether lower bandwidth simplifies separation. The bandwidth scaling results would benefit from deeper analysis explaining why performance degrades at higher rates. Reporting per-dataset or per-source breakdowns instead of aggregated averages would help interpretation. A discussion of failure cases would also make the work more complete, particularly where latent masking may fundamentally struggle. Some notation inconsistencies could be cleaned up, and the claim about being the first on-device codec-based separation model should be slightly softened since deployment is inferred from compute analysis rather than demonstrated directly (and I'm not sure if this is actually the first on-device model.)

---

> ### Author Response · Authors · 2026-03-27
> **[Part 1] Response to Reviewer RArg**
>
> We sincerely thank the reviewer  for the constructive suggestions and encouraging assessment. We agree that the main issues were comparison fairness, direct support for the structured-latent hypothesis, framing of the efficiency claim, and the MOS protocol. We revised the manuscript substantially to address each point.
>
> **1. Codec-processing confound / comparison fairness.**
> We thank the reviewer for raising this important point. As the reviewer requested, we have made the comparison setup and its intended scope much more explicit. Our primary claim was always about the **codec-mediated deployment regime**, where audio is already exchanged as compressed bitstreams. To address this directly, we now:
> (i) explicitly frame **Tables 1 and 2** as **compressed-input / deployment-oriented comparisons**;
> (ii) have added a dedicated discussion, **“Why evaluate non-codec baselines on codec-processed audio?”**, explaining that this setup is meant to reflect the deployment regime motivating CodecSep—namely, that in realistic edge--server workflows, non-codec baselines must decode codec streams, separate in the waveform/spectrogram domain, and re-encode if codec-compatible output is needed, whereas CodecSep avoids this decode–separate–re-encode overhead by operating directly on codec representations; and
>
> (iii) have added a new completeness table, **Table 3**, reporting the non-codec baselines in their **native raw-audio setting**, again as the reviewer requested.
>
> These added raw-audio results confirm the reviewer’s concern that AudioSep and related baselines improve in their native operating regime. However, they also clarify that this does not change the paper’s main conclusion in the **deployment regime of interest**. Concretely, on **dnr-v2-test**, **CodecSep+dnr-v2** in the codec-processed setting (Table 1) still exceeds the **raw-audio retrained AudioSep baselines** in SI-SDR on all three stems, while remaining broadly competitive in perceptual quality. We also note that the **zero-shot AudioSep checkpoint** remains perceptually strong in the raw-audio setting, which is unsurprising given that it was trained on a much larger and more diverse corpus (roughly **15K hours**); even so, **CodecSep+dnr-v2** still surpasses this stronger zero-shot AudioSep model in **SI-SDR** on all three dnr-v2 stems, despite AudioSep retaining an advantage on some perceptual scores. Likewise, on the five additional benchmarks, **AudioSep+dnr-v2** on raw audio is modestly stronger than its codec-processed counterpart, but **CodecSep+dnr-v2** in the codec-processed setting (Table 2) still maintains consistently stronger SI-SDR across all five datasets, with broadly comparable ViSQOL and only a small deficit on AudioCaps.
>
> We therefore now present the comparison explicitly as a **deployment-regime comparison**, not as a claim that non-codec baselines are universally weaker in their native raw-audio setting.
>
>
>
> **2. Qualitative  support for the “structured latent” hypothesis.**
> We agree that the earlier submission supported this claim too indirectly. We therefore added **Sec. 4.1.4: “Qualitative Evidence for Structured Source Organization in Codec Latent Space”** and **Fig. 3**. The figure shows the mixture latent, the reference speech/music/SFX latents, the estimated source-conditioned masks, and a residual analysis. The heatmaps show that the frozen codec latent space already preserves clear source-dependent structure, while the estimated masks are predominantly **channel-wise**, with little temporal variation. This indicates that CodecSep separates sources mainly through source-conditioned reweighting of latent channels rather than localized temporal gating. To further probe source-compositional structure, we compare the mixture latent with the average of the corresponding source latents and visualize the absolute residual. The residual remains sparse over broad regions, suggesting that the mixture latent remains closely related to the organization of its constituent sources rather than collapsing into an unstructured representation. While this does not imply exact linear additivity, it provides qualitative evidence of structured source organization and partial source disentanglement in codec latent space. We also clarify that the slight residual leakage heard in some supplementary examples is consistent with this mechanism, since shared latent channels may carry partially overlapping information and remain weakly active after masking.

---

> ### Author Response · Authors · 2026-03-27
> **[Part 2] Response to Reviewer RArg**
>
> **3. Sampling-rate mismatch and bandwidth-scaling experiments.**
> We thank the reviewer for raising this important concern. We agree that both the **sampling-rate mismatch** and the role of **bandwidth scaling** should be discussed explicitly.
>
> **Why AudioSep remains at 32 kHz.**
> In our study, **AudioSep** is evaluated in the configuration for which it is actually available and established: the official pretrained checkpoint is a **32 kHz** model trained on a much larger corpus (roughly **15K hours**), and for fairness we also retrain the same **32 kHz AudioSep variant** under our matched training protocol. We did not change AudioSep to a different sampling rate because doing so would require nontrivial redesign and careful hyperparameter re-optimization, which would introduce a new confound by evaluating a materially altered version of the baseline rather than the standard model itself.
>
> **Why we added bandwidth-scaling experiments.**
> At the same time, we agree that lower bandwidth can simplify separation, so this mismatch should not simply be ignored. To address this directly, we retained the standard **32 kHz AudioSep** baselines and added a dedicated **bandwidth-scaling study for CodecSep** (**Sec. 4.1.8 / Table 10**), where we evaluate the same masking framework with **16, 24, 44.1, and 48 kHz** codec backbones.
>
> **What the bandwidth-scaling results show.**
> This analysis shows that, for a fixed masker capacity, performance generally **degrades as sampling rate increases**, which supports the reviewer’s concern that lower-bandwidth settings are indeed easier. Concretely, relative to **CodecSep at 16 kHz** (\(1.2/10.0/0.9\) SI-SDR for music/speech/SFX; \(2.9/3.1/2.3\) ViSQOL), the **24 kHz** version remains strong but drops modestly (\(0.2/8.8/0.6\) SI-SDR; \(2.7/3.0/2.3\) ViSQOL), while the **44.1 kHz** and **48 kHz** versions degrade further (\(-2.3/5.9/-0.3\) and \(-2.8/5.4/-0.5\) SI-SDR, respectively). Thus, the 16 kHz operating point is indeed favorable and should not be interpreted as bandwidth-independent evidence.
>
> **Why the main CodecSep model uses 16 kHz DAC.**
> This also explains why our **main CodecSep model uses the 16 kHz DAC backbone**. The paper’s core objective is to study **low-MAC, codec-mediated separation**, and the 16 kHz DAC variant provides the most favorable tradeoff between separation quality and deployment efficiency. Higher-bandwidth backbones remain important to test the generality of the masking interface, but they are progressively less aligned with the low-compute regime emphasized in this work because they increase latent sequence length, compute, and modeling difficulty. We therefore use the **16 kHz DAC model as the primary system** because it best reflects the practical low-MAC setting that motivates CodecSep, and we include the higher-bandwidth results to show how the same approach behaves as bandwidth increases.
>
> **Why higher-bandwidth separation is harder.**
> As the sampling rate \(F_s\) increases, the representation must account for richer high-frequency structure, finer temporal transients, and more densely packed spectral detail. In codec latent space, these additional details are typically less cleanly isolated than lower-frequency coarse structure and become more tightly entangled across sources. As a result, a masker-based separator must make more delicate source-selection decisions over a denser and less separable latent organization. At the same time, the latent sequence length typically increases, which raises both compute and modeling difficulty. Together, these effects make full-band separation intrinsically harder than narrow-band or mid-band separation, even though the underlying masking formulation remains unchanged.
>
> **How bandwidth-scaled CodecSep compares with AudioSep.**
> Importantly, however, the same study also shows that the **masking formulation remains effective beyond 16 kHz**, rather than collapsing outside the narrowband case. Against the **retrained 32 kHz AudioSep+dnr-v2** baseline (\(-5.6/7.7/-4.5\) SI-SDR; \(2.6/2.5/2.3\) ViSQOL), **CodecSep at 24 kHz** still performs better on all three stems in SI-SDR and also improves ViSQOL for music and speech while matching SFX ViSQOL. Even the **44.1 kHz** and **48 kHz** CodecSep variants remain substantially stronger than retrained AudioSep on music and SFX SI-SDR. Relative to the much stronger **pretrained 32 kHz AudioSep** checkpoint (\(-2.5/4.9/-0.3\) SI-SDR; \(2.9/3.1/2.6\) ViSQOL), the comparison is more mixed: **CodecSep at 24 kHz** still exceeds it in SI-SDR on all three stems, while **44.1/48 kHz CodecSep** continue to outperform it on speech SI-SDR and remain competitive on music/SFX SI-SDR. In ViSQOL, the pretrained AudioSep checkpoint remains especially strong perceptually, particularly for SFX, which is unsurprising given its much larger training corpus.
>
>
> [*Continuation of this rebuttal point is provided in the next comment.*]

---

> > ### Author Response · Authors · 2026-03-27
> > **[Part 3] Response to Reviewer RArg**
> >
> > **3. Sampling-rate mismatch and bandwidth-scaling experiments (continued).**
> >
> > **Why we do not report a 32 kHz codec-domain counterpart.**
> > There is also no publicly available neural audio codec in our setup operating at **32 kHz**, so an exactly matched codec-domain 32 kHz counterpart was not available for a cleaner apples-to-apples comparison. We therefore used the best available higher-bandwidth codec backbones (24, 44.1, and 48 kHz) to test whether the core method still functions beyond the 16 kHz case. All of these codec-based variants also preserve the main deployment advantage of CodecSep: they can operate directly on code streams, including a codes-in / codes-out path, with minimal loss relative to the continuous-latent path.
> >
> > **How the revised paper frames this comparison.**
> > We have revised the manuscript  to make this interpretation explicit. The comparison to AudioSep should therefore be understood as follows: **AudioSep is benchmarked in its standard and strongest available 32 kHz form**, while **CodecSep is benchmarked in the codec-native settings supported by available neural codec backbones**, with **16 kHz used as the primary low-MAC operating point** and higher-bandwidth experiments included specifically to contextualize the sampling-rate mismatch and test bandwidth scaling. Our claim is therefore not that 16 kHz and 32 kHz are perfectly matched operating points, but that (i) CodecSep is effective in the codec-mediated deployment regime, (ii) its masking interface remains valid as bandwidth increases, and (iii) higher-bandwidth operation becomes progressively harder, which we now state clearly and treat as an important limitation / future direction.
> >
> > We have added all the discussion to Section 4.1.8  in the revised version of manuscript.
> >
> >
> > **4. Efficiency claim / systems vs. architectural contribution.**
> > We clarify that the \(\sim54\times\) efficiency gain was always intended specifically for the **code-stream deployment regime**, not as a universal end-to-end cost claim. We now make this explicit throughout the paper, especially in **Sec. 4.1.7: “Efficiency in Code-Stream Deployment”** and the paragraph **“How should the audio-stream results be interpreted?”** There we now state directly that in the **audio-stream setting**, where CodecSep must also run the codec encoder/decoder, it is **not cheaper than AudioSep** end-to-end. The main efficiency claim is therefore now framed strictly as a **deployment-specific systems advantage**, while the **architecture-only** comparison is reported separately.
> >
> > **5. MOS protocol and statistical testing.**
> > We now clarify that we compared against the **official pretrained AudioSep checkpoint** because on dnr-v2 it performed substantially better than our retrained AudioSep baseline; using the stronger AudioSep system therefore makes the subjective comparison more conservative. We also note that this pretrained AudioSep checkpoint was trained on a much larger corpus (roughly **15K hours** of audio), which further strengthens the conservativeness of this comparison. In addition, we added **paired statistical testing on per-clip MOS-LQS means** (paired \(t\)-tests, Wilcoxon tests, and 95% confidence intervals) in Table 4. The gains of CodecSep are statistically significant overall and for music, speech, and SFX individually.
> >
> > **6. Statistical significance for objective results.**
> > We added paired statistical analysis for the additional benchmark results in **Table 5**, including **mean paired gains, 95% confidence intervals, paired \(t\)-tests, and Wilcoxon tests**. CodecSep+dnr-v2 improves SI-SDR over AudioSep+dnr-v2 on all five additional benchmarks, and the 95% confidence intervals exclude zero in every case. For ViSQOL, the gains are smaller, and we now interpret them more cautiously, especially on AudioCaps.

---

> ### Author Response · Authors · 2026-03-27
> **[Part 4] Response to Reviewer RArg**
>
> **7. Additional clarifications requested by the reviewer.** We also revised the manuscript to address the smaller but important points raised:
>
>
> - We clarified the reporting of **dataset-level vs. per-source results** in Section 4 para- Dataset , page-8. For **dnr-v2**, we report **per-source scores** for speech, music, and SFX. For the additional open-domain benchmarks, however, evaluation is **event-conditioned rather than stem-conditioned**, and many labels do not map cleanly onto the speech/music/SFX taxonomy. We therefore report **dataset-level averages**, which better reflect the open-domain extraction setting; a finer semantic grouping could be explored in future work.
>
> - We added an **oracle codec reconstruction vs. separated output** analysis (**Sec. 4.1.5 / Table 7**) to separate codec-induced distortion from separation-specific error.
>  In the oracle setting, the mixture and each clean reference stem are independently passed through the frozen DAC encoder-decoder and evaluated against their references, isolating codec-only distortion. The oracle analysis shows that DAC is already lossy, with SI-SDR / ViSQOL of \(6.3/4.1\) for mixture, \(6.8/3.8\) for music, \(7.4/4.2\) for speech, and \(1.8/3.8\) for SFX. Comparing these values with CodecSep (\(4.1/3.7\), \(1.2/2.9\), \(10.0/3.2\), \(0.9/2.3\)) shows that the degradation cannot be attributed to the codec alone. For mixture, music, and SFX, CodecSep remains below the oracle in both metrics, indicating an additional separation gap. We also clarify that this gap is not solely due to separator imperfections, since the oracle reconstructs clean stems directly whereas CodecSep must recover each target from a mixture, which is intrinsically harder in dnr-v2. Speech shows a different pattern, with higher SI-SDR but lower ViSQOL than the oracle; we now explain this as a metric distinction, where stronger interference suppression can improve signal-level recovery even when perceptual fidelity remains limited by codec and masking artifacts.
>
> - We also expanded the discussion of **limitations / failure cases**. In the previous submission, these limitations were deferred to the Appendix; in the revision, we have **moved them into the main paper Section 6** and also added a new limitation motivated directly by our qualitative latent-space analysis. Specifically, we now discuss: (i) limited **training data scale and prompt diversity** relative to open-domain audio, (ii) the fact that we do not yet evaluate **temporal/relational prompts**, (iii) cases where **SFX perceptual quality** can still lag despite stronger SI-SDR, and (iv) a newly added limitation on **channel-wise masking and residual overlap**—namely, that because the learned masks are predominantly channel-wise rather than time-localized, sources sharing partially overlapping latent channels may not be perfectly disentangled, leaving some non-target content weakly active in dense mixtures with overlapping events.
>
> - We reorganized the experiments around **explicit question-style headings**, so each paragraph now answers a concrete hypothesis directly from the corresponding results.
>
> - We clarified the notation and made the pipeline description substantially more explicit by reorganizing **Section 3** into **(a) task formulation, (b) model architecture, (c) training objective, and (d) deployment discussion**. We now explicitly specify the DAC latent \(Z \in \mathbb{R}^{d\times T}\), the transformer width \(d_t\), the pointwise convolutional projection from codec width to transformer width, the FiLM injection points (layers \(2,\dots,L-1\)), the query network design, and the pointwise convolutional mask head that maps the masker output back to codec width. We also clarify the FiLM formulation and stability in **Sec. 3.2.3**, including that CodecSep uses additive Post-LN FiLM only (\(\gamma^l=\mathbf{1}\)) with **Xavier uniform** weight initialization and **zero** bias initialization.
>
> - We added a **Broader Impact** section (Section 8) discussing both positive applications and misuse risks, together with possible mitigations.
>
> - We softened potentially overstrong claims, including the deployment wording, so that the paper now presents CodecSep as **useful for on-device or edge applications based on its favorable compute and memory footprint**, rather than as a system whose on-device capability has been directly demonstrated experimentally. Our evidence is limited to benchmarking on a small GPU and code-stream efficiency analysis, so we now phrase this point more carefully throughout the manuscript.
>
> We thank the reviewer again for these valuable suggestions. We believe the revised manuscript now addresses the major concerns directly and is substantially clearer and stronger scientifically.

---

> ### Author Response · Authors · 2026-03-30
> **[Part 5] Response to Reviewer RArg**
>
> **8. Failure cases and where latent masking can struggle.**
> We agree that a clearer discussion of failure cases makes the paper more complete.
>
> In the revised manuscript, we now address failure modes in three complementary ways.
>
>  First, we use **AudioCaps** as a deliberate **cross-domain stress test / failure-case analysis** (cf. appendix F): unlike the other external benchmarks, this is the one setting where the **dnr-v2-trained CodecSep** does not retain the same favorable perceptual behavior, making it an informative boundary case for generalization under substantial distribution shift.
>
> Second, we added the **oracle codec reconstruction vs. separated-output analysis** (**Sec. 4.1.5 / Table 7**), which functions as a separation-specific failure-case study: it shows that for mixture, music, and SFX, CodecSep remains below the codec upper bound, so the remaining degradation cannot be explained by codec lossiness alone and reflects an additional **separation gap**.
>
>  Third, we expanded the limitations discussion in the main paper to make the mechanism-level limits of latent masking explicit. In particular, our qualitative analysis shows that the learned masks are predominantly **channel-wise** rather than strongly time-localized, which helps explain why dense mixtures with overlapping events may retain **residual overlap** and why latent masking may struggle when multiple sources share partially overlapping latent channels. The **masker ablation** clarifies that **FiLM-only conditioning inside the NAC encoder is sufficient for source-consistent reconstruction but not for source extraction from mixtures**. In the ablated variant, the affine FiLM modulation can reconstruct an already isolated, prompt-matched source very well, but in mixtures it tends to **collapse the latent space toward a prompt-biased representation** rather than preserving the separable structure needed for disentanglement. As a result, separation degrades sharply, showing that an **explicit masker** is necessary to preserve usable source structure and perform source selection.
>
> We believe these additions make the paper more complete by identifying not only where CodecSep works well, but also where codec-latent masking becomes more challenging.
>
> We thank the reviewer again for these valuable suggestions.

---

### Review · Reviewer_CYZH · 2026-03-11

**Summary Of Contributions:**

The paper proposes CodecSep, a text-guided universal sound separation method operating directly on neural audio codec latents via a FiLM-conditioned transformer masker with a frozen DAC backbone and CLAP text embeddings. The main motivation is to enable prompt-driven separation with substantially lower compute and better deployment compatibility than spectrogram-domain baselines, especially in code-stream settings. The paper reports experiments on dnr-v2 and five additional open-domain benchmarks, along with ablations, paraphrase tests, and compute analysis.

Strength:
- The paper addresses a practically relevant problem: bringing text-guided sound separation closer to efficient, deployment-friendly settings, especially code-stream or edge/server pipelines. The compute-oriented motivation is clear and important.
- The empirical study is broad in scope, covering in-domain evaluation, multiple external benchmarks, prompt granularity, paraphrase robustness, ablations, and compute benchmarking. This breadth suggests substantial engineering effort.
- The codec-domain design is practically appealing, and the compute advantages in code-stream settings are potentially significant.

Weaknesses:
- The paper’s primary contribution is not yet sufficiently strong or clearly distinguished from existing combinations of known components.
- Section 3 is difficult to follow and not well organized. The paper mixes design rationale, deployment arguments, architectural description, and method details in a way that makes the actual model specification harder to parse than necessary. Several statements are intuitive but not presented with enough precision to fully reconstruct the method from the main text alone. For example, the exact structure of the query network / FiLM parameterization is under-specified.
- The experimental section is extensive but not logically organized around a small number of testable hypotheses. Many tables are presented, but the evaluation narrative is somewhat diffuse. It is not always clear which experiment is intended to validate which central claim.

**Audience:**

Yes

**Audience Explanation:**

Researchers working on audio modeling, source separation, multimodal conditioning, and efficient deployment would likely find the paper relevant.

**Broader Impact Concerns:**

No Broader impact concerns.

**Claims And Evidence:**

Yes

**Claims Explanation:**

The paper does provide a relatively large amount of empirical evidence, and some claims are reasonably supported. In particular, the efficiency/deployment claim is one of the stronger parts of the paper: the code-stream setting is clearly motivated, and the reported GMACs/time/parameter comparisons suggest a substantial systems advantage for the codec-domain approach.

However, the evidence is not always presented clearly enough. The methodology and evaluation protocol require substantial effort to piece together across the main text and appendix. Besides, the claim that the work demonstrates a deeper property of NAC latent spaces—namely, that they are already sufficiently source-structured for universal masking—is interesting, but the paper does not really isolate this claim with strong mechanistic evidence. More direct evidence, such as qualitative latent-space visualizations or targeted representation analysis, would strengthen this part of the paper.

**Requested Changes:**

1. Reorganize Section 3 for clarity.
  The method section should separate:

    (a) task formulation,

    (b) actual model architecture,

    (c) training objective, and

    (d) deployment discussion.

    Right now, rationale, architecture, and deployment arguments are interleaved, which makes the paper hard to follow. The authors should present the core pipeline more concretely, including the dimensionalities, the exact query network design, where FiLM is applied, and how the DAC latent dimension interfaces with the transformer width.

2. Restructure the experiments around explicit hypotheses.
The paper would be much stronger if each experimental block answered one clear question, for example:
- Does codec-latent masking outperform spectrogram-domain text-guided separation under matched training?
- Is masking better than generation in codec latent space?
- Does the method remain effective under prompt variation?
- What exactly is gained in code-stream deployment?
This would make the evaluation much easier to interpret.

3. Strengthen the analysis of statistical significance and practical significance.
Some gains, especially on the additional datasets, are small relative to the reported variance. The paper would benefit from confidence intervals, significance testing, or at least a more cautious interpretation of these results.

---

> ### Author Response · Authors · 2026-03-27
> **[Part 1] Response to reviewer CYZH**
>
> We sincerely thank the reviewer for the constructive suggestions and encouraging assessment. We are grateful that the reviewer found the paper practically relevant, the code-stream deployment motivation important, and the empirical scope substantial. In response, we revised the manuscript to incorporate all requested changes and clarifications.
>
> **1. Section 3 has been reorganized for clarity, as requested by the reviewer.**
> Following the reviewer’s suggestion, we now separate the method into **(a) task formulation, (b) model architecture, (c) training objective, and (d) deployment discussion**. We also made the pipeline more concrete by explicitly specifying the DAC latent \(Z \in \mathbb{R}^{d\times T}\), the transformer width \(d_t\), the pointwise convolutional projection from codec width to transformer width, the FiLM injection points (layers \(2,\dots,L-1\)), the query network design, and the pointwise convolutional mask head that maps the masker output back to codec width. We further clarified the FiLM design and stability in **Sec. 3.2.3**, including that CodecSep uses additive Post-LN FiLM only (\(\gamma^l=\mathbf{1}\)) with **Xavier uniform** weight initialization and **zero** bias initialization, so the conditioned masker starts as an identity-like unmodulated masker.
>
> **2. The experiments have been restructured around explicit questions / hypotheses, as requested by the reviewer.**
> We agree that the earlier presentation was too diffuse. In the revision, each experimental paragraph now begins with a **question-style heading**, and the paragraph that follows answers that specific question directly from the corresponding results. This structure now guides **Section 4**, so each experimental block supports a specific central claim and makes the role of each table easier to interpret.
>
> **3. We strengthened statistical analysis and clarified practical significance.**
> To address the reviewer’s concern that some gains are small relative to variance, we added paired statistical analysis for the additional benchmark results in **Table 2**, including **mean paired gains, 95% confidence intervals, paired \(t\)-tests, and Wilcoxon signed-rank tests**. CodecSep+dnr-v2 improves SI-SDR over AudioSep+dnr-v2 on all five additional benchmarks: **+1.88 dB (ESC-50), +2.4 dB (Clotho-v2), +1.25 dB (AudioSet), +0.92 dB (VGGSound), and +0.30 dB (AudioCaps)**. The **95% confidence intervals exclude zero in every case**, and both paired tests indicate significance throughout. For ViSQOL, the gains are smaller; we now interpret them more cautiously and explicitly note that AudioCaps is mixed/slightly favors AudioSep. We also added statistical significance analysis for the **MOS-LQS subjective evaluation** in **Table 4**. Using paired tests on per-clip mean ratings, the gains of CodecSep over AudioSep are statistically significant for the **overall MOS-LQS** as well as for **music, speech, and SFX individually**. We therefore now support the subjective results not only with mean scores, but also with confidence intervals and paired significance tests, showing that the perceptual gains are consistent across clips rather than driven by a small subset of examples.
>
> **4. We added direct qualitative evidence for source structure in codec latent space.**
> To better support the claim that NAC latents are already source-structured, we added **Sec. 4.1.4** and **Fig. 3**, which visualize the mixture latent, the reference source latents, the estimated source-conditioned masks, and a residual analysis. The residual is generated by comparing the mixture latent with the average of the corresponding speech, music, and SFX reference latents and visualizing the absolute difference. The heatmaps show clear source-dependent structure, while the learned masks are predominantly **channel-wise** rather than strongly time-localized. The residual remains sparse over broad regions, suggesting that the mixture latent remains closely related to the organization of its constituent sources rather than collapsing into an unstructured representation. We present this as **qualitative support** for structured source organization / partial source disentanglement.
>
> **5. We also added an oracle analysis to clarify what limits output quality.**
> We added **Sec. 4.1.5** and **Table 7** to separate codec-induced distortion from separation-specific error. The oracle analysis shows that DAC itself is already lossy, but also that the observed degradation in CodecSep cannot be attributed to the codec alone. For mixture, music, and SFX, CodecSep remains below the oracle, indicating an additional separation gap beyond codec distortion; for speech, CodecSep exceeds the oracle in SI-SDR but not in ViSQOL, which we now explain as a metric distinction between signal-level recovery and perceptual quality.
>
> We thank the reviewer again for these valuable suggestions.

---

> ### Author Response · Authors · 2026-03-30
> **[Part 2] Response to reviewer CYZH**
>
> **6. We revised the prompt-granularity analysis in Sec. 4.1.2 and softened comparative claims under coarse prompting.**
> In the revision, we substantially rewrote the discussion in **Sec. 4.1.2 Effect of Prompt Granularity on Source Separation Performance** to make the role of prompt granularity more precise and better aligned with the reported evidence. In particular, we now explicitly distinguish between **generic prompting**, which uses only fixed stem labels such as \textquote{music}, \textquote{speech}, and \textquote{sfx}, and **fine-grained prompting**, which replaces the coarse \textquote{sfx} label with more detailed compositional SFX descriptions. This distinction is now stated directly in the text so that the reader can clearly understand what is being varied in the experiment.
>
> We also revised the interpretation to avoid overstating CodecSep’s advantage under the **generic-prompt** setting. We now make clear that, under **matched training and evaluation with only coarse stem labels**, CodecSep should be interpreted as **competitive** with AudioSep rather than uniformly better on every stem and metric. The revised text highlights the specific aspects where CodecSep remains strong in this regime—most notably **speech perceptual quality** and **SFX separation quality**—while also acknowledging that the gains are **not uniform across all stems**, particularly for music.
>
> Finally, we clarified the intended takeaway of this section. The main conclusion is **not** that CodecSep consistently dominates fixed-stem baselines or generic-prompt text-guided baselines in every setting. Rather, the key finding is that **finer-grained semantic supervision improves the consistency and strength of universal prompt-conditioned separation**. In other words, the analysis is meant to show that prompt specificity is itself an important design variable: CodecSep remains effective even under coarse prompting, but richer semantic descriptions yield a clearer and more consistent benefit across the mixture. We believe this revised framing makes the claim more accurate, better calibrated to the results, and more useful to readers interested in how prompt design affects universal source separation.
>
>
> We thank the reviewer again for these valuable suggestions.

---

> ### Author Response · Authors · 2026-03-30
> **[Part 3] Response to reviewer CYZH**
>
> **7. We also reorganized and revised the appendix experiments to follow the same question-driven structure.**
> In response to the reviewer’s suggestion, we revised the **appendix F-H** so that each experimental block now addresses a **specific hypothesis** rather than appearing as a loosely connected extension of the main text. This includes explicit question-led discussions for **cross-benchmark transfer between AudioCaps and dnr-v2** (appendix F), **relative-gain summaries under matched training and prompt settings** (appendix G), and **reconstruction studies** (appendix H).
>
> We also revised the interpretation of these appendix experiments to make the takeaways clearer and more precise.
>
>  The **cross-benchmark transfer** analysis now makes clear that the **AudioCaps** experiment serves not only as a transfer study, but also as a deliberate **stress test / failure-case analysis**. In the broader benchmark suite, CodecSep remains favorable overall, but **AudioCaps is the one dataset where the dnr-v2-trained CodecSep model does not show the same perceptual advantage**, making it an informative boundary case for cross-domain generalization under substantial distribution shift. The revised text therefore presents AudioCaps more explicitly as a harder transfer target, while still showing that CodecSep remains **competitive in SI-SDR** relative to a matched retrained AudioSep baseline and that, under **matched AudioCaps training**, CodecSep again shows a signal-level advantage.
>
> The **relative-gain summaries** are now framed as compact support for the main matched-setting results, with the text explicitly clarifying that these percentages are intended to complement the absolute tables and are strongest on **dnr-v2**, still visible under transfer, and smaller under prompt ambiguity.
>
>  For the **reconstruction study**, we now state the protocol explicitly: although the models are trained for **source separation**, each model is evaluated in a **single-source reconstruction** setting by giving it an isolated source together with the matching prompt and asking it to reproduce that source; we also report **mixture reconstruction** by summing the separated outputs and comparing the result to the original mixture.
>
> The revised reconstruction discussion also makes the architectural interpretation clearer. It now shows that, within codec-latent source separation models, **explicit masking is substantially more effective than decoder-style latent generation**, while also clarifying that reconstruction and separation probe different behaviors. In particular, the **masker-ablated CodecSep** variant achieves very strong reconstruction because FiLM is applied directly inside the NAC encoder, allowing source-consistent reconstruction when the input already matches the prompt. However, this same direct FiLM-based conditioning is not sufficient for source extraction from mixtures, since affine modulation at the intermediate layers tends to collapse the latent space toward a prompt-biased representation; the full \textbf{explicit masker} is therefore needed to preserve separable structure and perform source selection.
>
>  Overall, these appendix revisions make the extended empirical evidence easier to interpret and more tightly aligned with the central hypotheses of the paper.
>
> We also reflect the same revision in section 4.1.9 Additional Experiments in main paper.
>
> We thank the reviewer again for these valuable suggestions.

---

> > ### Author Response · Authors · 2026-04-01
> > **[Part 4] Response to Reviewer  CYZH**
> >
> > **8. We also revised Appendix B to mirror the requested method structure and to make its relation to the main paper explicit.**
> >
> >
> > In addition to reorganizing **Sec. 3** in the main paper, we also substantially revised **Appendix B: “Extended Design Rationale: FiLM-Conditioned Masking in NAC Latent Space”** so that it follows the same reviewer-requested presentation order: **(i) task formulation and pipeline contrast, (ii) model architecture and rationale, (iii) training / representation considerations, and (iv) deployment discussion**. The revised appendix is now explicitly written as an \emph{extension} of the main method section rather than as a loosely connected set of additional arguments. In particular, Appendix B now opens by stating its direct relation to **Sec. 3**, clarifying that its purpose is to expand and technically motivate the design choices already introduced in the main paper, not to redefine the method independently.
> >
> > We also aligned the appendix discussion closely with the revised main-paper content. The updated Appendix B now explains, in the same organizational logic as Sec. 3, why CodecSep operates on NAC latents rather than spectrograms, why masking is preferred over decoder-style latent generation, why FiLM is injected inside the masker, why the main experiments are reported on continuous latents (Z), and why the same formulation extends naturally to code-stream deployment. It further connects these architectural arguments directly to the newly added qualitative latent analysis (**Sec. 4.1.4**) and oracle codec analysis (**Sec. 4.1.5**), so that the appendix now serves as a coherent bridge between the formal method description in the main text and the mechanistic evidence added in the experiments. We believe this revision addresses the reviewer’s concern that the design rationale and method details were previously too interleaved and difficult to reconstruct across the main paper and appendix.
> >
> >
> > We thank the reviewer again for these valuable suggestions.

---

### Review · Reviewer_sfaK · 2026-03-15

**Summary Of Contributions:**

This paper presents CodecSep, a highly efficient, prompt-driven framework for universal sound separation. The primary technical contribution lies in performing separation directly within the latent space of a pre-trained Neural Audio Codec (specifically DAC). By leveraging a lightweight Transformer Masker conditioned via FiLM (Feature-wise Linear Modulation) using CLAP embeddings, the authors achieve a 54x reduction in computational complexity (1.35 GMACs) compared to state-of-the-art spectrogram-based models like AudioSep, while maintaining competitive separation performance.

Key Strengths:
1. Computational Efficiency: The paradigm shift from waveform/spectrogram domains to the compressed latent domain provides a significant breakthrough for on-device deployment.
2. Generalized Modality: Successfully bridges the gap between fixed-class codec-domain separators and open-vocabulary, text-prompted universal separation.
3. Methodological Clarity: The discovery that simple masking in the NAC latent space is sufficient for high-fidelity separation is an insightful observation for the representation learning community.

**Audience:**

Yes

**Audience Explanation:**

The work is highly relevant to researchers in efficient machine learning, audio signal processing, and multimodal representation learning. The strategy of "compressed-domain processing" offers a blueprint for other tasks such as audio enhancement, editing, and synthesis.

**Broader Impact Concerns:**

No significant ethical concerns are identified beyond the standard considerations.

**Claims And Evidence:**

Yes

**Claims Explanation:**

The authors provide a robust empirical evaluation across multiple benchmark datasets, including FSD50K and AudioCaps. The performance gains are measured using both objective signal-level metrics (SI-SDR) and perceptual metrics (ViSQOL), which validates the practical utility of the separated audio. The ablation studies effectively isolate the impact of the prompt-driven mechanism.

**Requested Changes:**

1. Post-LN FiLM over AdaLN:The current architecture employs a Post-LayerNorm FiLM modulation. While recent generative architectures (e.g., DiT) favor AdaLN (Adaptive Layer Norm), separation tasks often require more aggressive feature suppression to isolate target sources.Requirement: The authors must provide a technical justification for this design choice. Specifically, discuss whether Post-LN provides a more "precise" masking capability compared to AdaLN, which tends to distribute modulation across the normalization statistics and might result in "softer" masks that fail to suppress interference effectively in low-GMAC regimes.

2. Ablation and Upperbound Analysis (Oracle vs. Model):The separation quality is inherently bottlenecked by the DAC reconstruction. Please include a comparison between the "Codec Oracle" (direct reconstruction of the mixture) and the "Separated Output" to clarify the source of artifacts—distinguishing between separation errors and codec-induced distortion.

Strengthening Changes：
1. Stability in Training:Post-LN structures can be prone to training instabilities. It would strengthen the paper to detail the initialization strategy used for the FiLM linear layers (e.g., zero-initialization of $\gamma$ and $\beta$ to ensure the identity mapping at start-up).

2. Visual Analysis of Latent Masks:Include a visualization (e.g., a heatmap) of the predicted masks compared to the ground truth latent activations to demonstrate how the FiLM-guided attention selectively preserves target-relevant channels.

---

> ### Author Response · Authors · 2026-03-27
> **[Part 1] Response to Reviewer sfaK**
>
> We sincerely thank the reviewer for the constructive suggestions and encouraging assessment. We are grateful that the reviewer found the paper practically relevant and of interest to the TMLR audience. In response, we have revised the manuscript to incorporate all requested changes and clarifications raised by the reviewer.
>
> **1. Post-LN FiLM vs. AdaLN.**
> We added a task-specific justification for using **Post-LN FiLM** rather than **AdaLN** (**Sec. 3.2.3, page.5; “Why FiLM inside the masker,” “Why Post-LN FiLM instead of AdaLN”**). Our argument is that CodecSep performs **masking-based source selection**, not broad conditional generation, so conditioning should act as directly as possible on the latent features that must be preserved or suppressed. We therefore apply FiLM **after** the transformer sublayer, where it modulates hidden activations through channel-wise affine conditioning and behaves as an explicit feature gate: for a given query, it can directly attenuate channels carrying interfering content and emphasize channels carrying target-relevant structure, while leaving the backbone normalization statistics unchanged. By contrast, **AdaLN** injects conditioning into the normalization transform itself. While this is highly effective in generative architectures such as DiT, it is less directly aligned with sharp source-selective suppression in our lightweight low-GMAC separation regime, where more diffuse conditioning can reduce source selectivity. We also note that this interpretation is consistent with our new qualitative analysis (**Sec. 4.1.4**), where the learned masks appear predominantly channel-wise rather than strongly time-localized, supporting the view that separation is achieved mainly through FiLM-conditioned feature reweighting within the masker.
>
> **2. FiLM parameterization and training stability.**
> We now clarify the FiLM parameterization and initialization (**Sec. 3.2.3, page.7, para- FiLM parameterization and training stability. **). Following prior text-guided separation work, we use fixed scale and learned bias only, i.e., additive conditioning with \(\gamma^l=\mathbf{1}\). The FiLM layers follow AudioSep and are implemented as linear projections of the text embedding, with **Xavier uniform** weight initialization and **zero** bias initialization. Thus, the FiLM term is initially zero, so the conditioned masker starts as an identity-like unmodulated masker. This improves stability by avoiding uncontrolled multiplicative scaling, which can be especially problematic in Post-LN architectures.
>
> **3. Oracle codec reconstruction vs. separated output.**
> To distinguish codec-induced distortion from separation-specific error, we added **Sec. 4.1.5: “Oracle Codec Reconstruction vs. Separated Output”** and **Table 7**. In the oracle setting, the mixture and each clean reference stem are independently passed through the frozen DAC encoder-decoder and evaluated against their references, isolating codec-only distortion. The oracle analysis shows that DAC is already lossy, with SI-SDR / ViSQOL of \(6.3/4.1\) for mixture, \(6.8/3.8\) for music, \(7.4/4.2\) for speech, and \(1.8/3.8\) for SFX. Comparing these values with CodecSep (\(4.1/3.7\), \(1.2/2.9\), \(10.0/3.2\), \(0.9/2.3\)) shows that the degradation cannot be attributed to the codec alone. For mixture, music, and SFX, CodecSep remains below the oracle in both metrics, indicating an additional separation gap. We also clarify that this gap is not solely due to separator imperfections, since the oracle reconstructs clean stems directly whereas CodecSep must recover each target from a mixture, which is intrinsically harder in dnr-v2. Speech shows a different pattern, with higher SI-SDR but lower ViSQOL than the oracle; we now explain this as a metric distinction, where stronger interference suppression can improve signal-level recovery even when perceptual fidelity remains limited by codec and masking artifacts.

---

> ### Author Response · Authors · 2026-03-27
> **[Part 2] Response to Reviewer sfaK**
>
> **4. Visual analysis of latent masks.**
> We added **Sec. 4.1.4: “Qualitative Evidence for Structured Source Organization in Codec Latent Space”** and **Fig. 3**. The figure shows the mixture latent, the reference speech/music/SFX latents, the estimated source-conditioned masks, and a residual analysis. The heatmaps show that the frozen codec latent space already preserves clear source-dependent structure, while the estimated masks are predominantly **channel-wise**, with little temporal variation. This indicates that CodecSep separates sources mainly through source-conditioned reweighting of latent channels rather than localized temporal gating. To further probe source-compositional structure, we compare the mixture latent with the average of the corresponding source latents and visualize the absolute residual. The residual remains sparse over broad regions, suggesting that the mixture latent remains closely related to the organization of its constituent sources rather than collapsing into an unstructured representation. While this does not imply exact linear additivity, it provides qualitative evidence of structured source organization and partial source disentanglement in codec latent space. We also clarify that the slight residual leakage heard in some supplementary examples is consistent with this mechanism, since shared latent channels may carry partially overlapping information and remain weakly active after masking.
>
> We thank the reviewer again for these valuable suggestions.

---

> ### Author Response · Authors · 2026-04-01
> **[Part 3] Response to Reviewer sfaK**
>
> **5. Deployment discussion and broader codec-native applicability.**
>
> We also expanded **Sec. 3.4, Deployment Discussion** by adding a new concluding paragraph (the **last paragraph** of Sec. 3.4) to make the broader systems significance of the codec-domain formulation more explicit. Specifically, we now clarify that the proposed **codes-in / codes-out** pathway is not only an efficiency advantage for CodecSep itself, but also a more general **codec-native deployment pattern** for audio processing systems. The revised text explains that, once audio is already represented in a neural codec space, downstream models need not repeatedly follow the conventional \emph{decode $\rightarrow$ process in waveform/STFT space $\rightarrow$ re-encode} pipeline. Instead, they can operate directly on codec representations, which preserves representational continuity across compression, transmission, and inference while reducing redundant latency, memory traffic, and energy cost.
>
> We further state in the revised manuscript that this design principle may extend beyond text-guided source separation to other deployment-oriented audio tasks such as **target speaker extraction, speech enhancement, denoising, dereverberation, and prompt-guided audio editing**, where the goal is likewise to conditionally modulate or refine an already compressed representation without leaving codec space. The claim is not that CodecSep solves all such tasks, but that it provides a concrete example of how neural audio codecs can serve as a **shared representational backbone** for a broader family of codec-aware audio processing pipelines.
>
> To reflect the broader-impact side of this point as well, we additionally incorporated the same discussion into **Sec. 8, Broader Impact**, where we note that the positive systems impact of the work is not only the efficiency of one separator, but also the suggestion of a reusable codec-native inference template for practical audio processing under deployment constraints. We believe this addition strengthens the practical significance of the paper by making clear that the deployment contribution is both immediate for CodecSep and potentially relevant to a wider class of real-world audio systems.
>
>
>
> We thank the reviewer again for these valuable suggestions.

---

### Author Response · Authors · 2026-03-30
**Final Revision Uploaded**

We thank the reviewers for their constructive feedback. We have now completed the rebuttal process and uploaded the final revised manuscript.

In the updated version, we strengthened the **Abstract**, **Section 1 (Introduction)**, and **Section 5 (Conclusion)** so that the scope of the paper, the supporting evidence, and the main takeaways are stated more clearly and consistently.

In the **Abstract**, we revised the summary to better reflect the final paper’s contribution structure and overall positioning. It now highlights not only the core methodological contribution—**CodecSep** as a text-guided universal sound separation framework operating directly in neural audio codec latent space—but also the main empirical findings, the architectural and qualitative evidence for codec-latent masking, and the practical deployment significance of the codec-native formulation.

In the **Introduction**, we now summarize the evaluation explicitly across seven complementary axes: in-domain evaluation, cross-domain generalization, prompt granularity, paraphrase robustness, architectural analysis, qualitative and diagnostic latent-space analysis, and deployment-oriented efficiency benchmarking.

We also reorganized the contributions into a clearer **four-part structure**. These now emphasize:
1. the introduction of **CodecSep** as a text-guided universal sound separation framework operating directly in neural audio codec latent space;
2. the empirical finding that codec-latent masking is an effective formulation for universal sound separation under matched comparisons with **AudioSep**;
3. the architectural, qualitative, and diagnostic evidence supporting this codec-latent masking mechanism, including source-dependent latent structure, predominantly channel-wise mask behavior, and oracle/reconstruction analyses; and
4. the substantial deployment advantage of the codec-native formulation in code-stream settings, including a concrete **codes in: codes out** deployment path.

We also revised **Section 5 (Conclusion)** so that it better reflects the final paper as a whole. It now summarizes not only the main empirical findings, but also their broader interpretation: **CodecSep** consistently improves **SI-SDR** over **AudioSep** under matched protocols, remains competitive in **ViSQOL**, shows gains in **MOS--LQS**, benefits from finer-grained semantic supervision, and supports a deployment-ready **codes in: codes out** pathway. We also clarified the systems perspective more explicitly: in codec-mediated edge--server workflows, when the transmitted representation already consists of NAC codes from the same codec on which the separator is built, the server can map those codes to codec embeddings by codebook lookup and perform separation directly in codec space, thereby avoiding a separate decode--separate--re-encode cycle. More broadly, this provides a concrete blueprint not only for codec-mediated separation, but also for efficient downstream audio processing built directly on NAC representations.

The revised conclusion now also states more clearly that the qualitative and diagnostic analyses support the central design premise that modern neural audio codec latents preserve meaningful source-dependent structure, while the oracle and reconstruction diagnostics clarify what portion of the remaining error comes from codec reconstruction, mixture difficulty, and separation-specific limitations.

More broadly, the revised manuscript incorporates the requested clarifications on architecture, qualitative interpretation, statistical significance, deployment realism, bandwidth scaling, failure cases, and limitations. We believe the paper is now clearer in its positioning, better supported in its empirical claims, and more precise about both the strengths and the boundaries of the proposed approach.

We sincerely thank the reviewers and the Action Editor for their time, care, and feedback.

---

> ### Author Response · Authors · 2026-04-01
> **Couple more Revisions Done**
>
> We substantially revised the methodological exposition around deployment and design rationale in both the main paper and appendix. In **Sec. 3.4 (Deployment Discussion)**, we added a new concluding paragraph clarifying that the proposed **codes-in / codes-out** inference path is not only an efficiency advantage for CodecSep, but also a more general **codec-native deployment pattern** for audio processing systems. We now explicitly state that, once audio is already represented in a neural codec space, downstream models need not repeatedly follow the conventional \emph{decode $\rightarrow$ process in waveform/STFT space $\rightarrow$ re-encode} pipeline, but can instead operate directly on codec representations, thereby preserving representational continuity across compression, transmission, and inference while reducing redundant latency, memory traffic, and energy cost. We also note that this design principle may extend beyond source separation to other deployment-oriented tasks such as **target speaker extraction, speech enhancement, denoising, dereverberation, and prompt-guided audio editing**, where the goal is likewise to conditionally refine or modulate an already compressed representation without leaving codec space. To reflect the broader-impact side of this point, we incorporated the same discussion into **Sec. 8 (Broader Impact)**.
>
> We also revised **Appendix B: “Extended Design Rationale: FiLM-Conditioned Masking in NAC Latent Space”** so that it now mirrors the same reviewer-requested structure adopted in the main paper: **(i) task formulation and pipeline contrast, (ii) model architecture and rationale, (iii) training / representation considerations, and (iv) deployment discussion**. The appendix is now explicitly framed as an \emph{extension} of **Sec. 3**, rather than as a loosely connected set of additional arguments. In the revised version, Appendix B makes its relation to the main method section explicit at the outset, and then expands, in the same organizational order, the motivations for operating on NAC latents rather than spectrograms, using masking rather than decoder-style generation, injecting FiLM inside the masker, training on continuous latents (Z), and extending the same formulation to code-stream deployment. It also now connects these design arguments directly to the newly added qualitative latent analysis and oracle codec analysis, so that the appendix serves as a clearer bridge between the formal method description in the main text and the mechanistic evidence provided in the experiments.
>
> Taken together, these revisions strengthen the paper’s overall positioning in two ways: first, they make the method and its rationale easier to reconstruct from the manuscript in a coherent, top-down fashion; and second, they clarify that the contribution is not only an efficient codec-domain separator, but also a more general example of how **neural audio codecs can serve as a shared representational backbone for broader codec-aware audio processing pipelines** under realistic deployment constraints.

---

> > ### Author Response · Authors · 2026-04-01
> > **Final Touch Up**
> >
> > **We strengthened the abstract to make the systems contribution and broader codec-native deployment perspective explicit.**
> > In the revised manuscript, we updated the **Abstract** to better reflect not only the separation results, but also the broader *systems perspective* of the work. In particular, we now state more explicitly that CodecSep provides a concrete *codes in: codes out* deployment path for codec-mediated processing: when the edge device transmits NAC codes generated by the same codec backbone used by the separator, the server can map those codes to codec embeddings through codebook lookup and perform separation directly in codec space, thereby avoiding a separate decode--separate--re-encode cycle.
> >
> > We also added a broader claim to clarify why this is important beyond the immediate CodecSep setting. The revised abstract now explains that this formulation is not only an efficiency advantage for one source separator, but also a concrete *blueprint for codec-native downstream audio processing*. Specifically, we note that the same systems principle may be relevant to other deployment-oriented audio tasks---such as enhancement, denoising, dereverberation, and prompt-guided audio editing---where it may be preferable to process an already compressed NAC representation directly, rather than repeatedly decoding to waveform and re-encoding after each downstream stage.
> >
> > We made this change so that the abstract better matches the final scope of the paper: the contribution is not only that CodecSep is an efficient text-guided separator, but also that it exemplifies a more general codec-native design pattern for practical audio inference under realistic deployment constraints.

---

### Decision · Action_Editor_7Bcp · 2026-05-03

**Recommendation:** Accept as is

**Audience:**

Yes

**Audience Explanation:**

This paper explores text-guided sound separation in the context of neural audio codecs, a direction that is highly relevant and valuable to a broad segment of the TMLR community.

**Claims And Evidence:**

Yes

**Claims Explanation:**

This submission presents CodecSep, a framework for text-guided universal sound separation operating in neural audio codec latent space. During the review process, the authors have revised the manuscript, addressing key concerns related to experimental rigor, claim scoping, and empirical validation.

The paper’s claims are supported by empirical evidence. It demonstrates consistent improvements over baselines under the same conditions, with added statistical significance testing for objective metrics. These are complemented by qualitative analyses (e.g., structured latent masks and residuals) and additional comparisons (e.g., raw-audio baselines).

---

> ### Author Response · Authors · 2026-05-21
>
> We sincerely thank the Action Editor and reviewers for the constructive and thoughtful reviews. Their feedback substantially improved the clarity and presentation of the paper.
>
> We have uploaded the final camera-ready version.

---

> > ### Author Response · Authors · 2026-05-31
> >
> > We have uploaded a video presentation as well.